# The impact of convection-permitting model rainfall on the dryland water balance

**George Blake[1], Katerina Michaelides[1], Elizabeth Kendon[1,2], Mark Cuthbert[3], and Michael Bliss Singer[3,4]**

[1]School of Geographical Sciences, University of Bristol, Bristol, BS8 BSS, United Kingdom
[2]Met Office Hadley Centre, Fitzroy Road, Exeter, EX1 3PB, United Kingdom
[3]School of Earth and Environmental Sciences, Cardiff University, Cardiff, CF10 3AT, United Kingdom
[4]Earth Research Institute, University of California, Santa Barbara, CA 93106, USA

**Correspondence:** George Blake (george.blake@bristol.ac.uk)

Received: 16 March 2025 – Revised: 22 October 2025 – Accepted: 7 November 2025 – Published:

**Abstract.** In drylands, rainfall is typically delivered during short-lived, localised convective storms, whose characteristics strongly influence how water is partitioned into different terrestrial stores. However, the rainfall data often used in modelling future projections of water resources is typically derived from climate models that are too coarse to represent convective processes occurring at scales smaller than the model grid. In this paper we quantify the impact of climate model representation of convection on the simulated water balance at four locations in the Horn of Africa: a humid site in the Ethiopian Highlands, a semi-arid site in southern Kenya, an arid site in eastern Ethiopia, and a hyper-arid site in northern Somalia. We benchmark a novel pan-Africa convection permitting climate model (CP4A) and its parameterised counterpart (P25) against high-resolution satellite-derived gridded datasets of rainfall (IMERG) and PET[TS1] (hPET). The comparison shows that explicitly resolving convection improves the representation of dryland rainfall characteristics, such as rainfall frequency, intensity, and the relative contribution of low vs. high-intensity rainfall to annual totals. We also demonstrate that convective representation can impact model PET, but differences are more muted relative to rainfall, and both CP4A/P25 can capture seasonal and diurnal PET dynamics. To establish how the impact of convective representation on rainfall characteristics can control hydrology, we used Hydrus 1-D to run one-dimensional vadose zone hydrological simulations at our four study sites, where Hydrus is driven by rainfall and PET from CP4A and P25 (and hPET). We find that the "drizzle" bias in P25 means that when rainfall is propagated through Hydrus, wetting fronts are confined to upper soil layers, resulting in higher evaporative losses, lower soil moisture, and lower bottom drainage in drylands. The improved representation of dryland rainfall characteristics in CP4A means that cumulative (in mm) surface runoff is up to ten times higher (over the ten-year simulation), bottom drainage (indicative of potential recharge) is up to 25 times higher, and soil moisture remains above the wilting point for longer compared to P25 Hydrus runs (despite simulating lower total rainfall and infiltration). Whereas at our humid site, water partitioning is less sensitive to rainfall characteristics and hydrological fluxes more closely follow annual rainfall totals. Our results demonstrate that dryland vadose zone hydrology is highly sensitive to the impact of convective representation on rainfall characteristics, and that studies focused on modelling future water resources using climate models that parameterise the average effects of convection risk misrepresenting societally relevant fluxes such as soil moisture, groundwater availability, and surface runoff.

## 1 Introduction

Drylands are characterised by limited and highly variable rainfall (both in space and time), where high temperatures, minimal humidity, and high solar radiation means the atmospheric water demand exceeds the available moisture supply (Reynolds et al., 2007a). When rainfall does occur in drylands, it typically falls during intense, localised, short duration rainstorms that are convective in nature (Nicholson, 2011; Singer and Michaelides, 2017; Singer et al., 2018; Hill et al., 2023). These storms are a critical source

of moisture and represent a key control on dryland sub-surface water availability (Singer et al., 2018; Quichimbo et al., ~~2021~~2021b, 2023). Therefore, the representation of convection and convective storms in climate models is critical for capturing dryland rainfall characteristics and subsequent water partitioning when rainfall is propagated through hydrological models. However, the grid resolution of general (global) or regional climate models (GCM/RCMs), which can range from 10–250 km, means they are unable to explicitly represent convective processes, which are occurring at the sub-grid scale (Prein et al., 2015; Clark et al., 2016).

The inability of GCM/RCMs to explicitly resolve convection means they may fail to adequately represent dryland rainfall characteristics, which is a major limitation in understanding future global climate resilience, as drylands cover ∼ 45 % of the Earth's land surface, are home to around 3 billion people, and represent the largest biome on Earth (Schimel, 2010; Mirzabaev et al., 2019). This is particularly important for people living in drylands, where livelihoods are often tightly linked to the regional expression of climatic variability, especially where populations primarily rely on intermittent surface water resources and rainfed pasture and crops (Stringer et al., 2009; Davenport et al., 2017, 2018; Hoffmann et al., 2022).

One such region is the Horn of Africa (HOA), where inconsistent seasonal rainfall during the two rainy seasons of March–May and October–December, and increasingly severe and frequent drought (Lyon and DeWitt, 2012; Lyon, 2014; Funk et al., 2019; Wainwright et al; 2019) are increasing the risk of regional food and water insecurity (Cheechi and Robinson, 2013; Nicholson, 2014; Funk et al., 2019). With research suggesting climate change will exacerbate water stress, desertification, and land degradation (Cook et al., ~~2020~~2022; Hoffmann et al., 2022; Kimutai et al., ~~2023~~2025), there is a pressing need to understand how future rainfall will impact surface and subsurface water availability in the HOA.

However, hydrological processes are particularly complex in drylands as the partitioning between infiltration, runoff, evaporation, transpiration, and recharge is highly dependent on the spatial and temporal characteristics of rainfall, rather than annual or seasonal totals (Taylor et al., 2013; ~~Arpuv~~ Apurv et al., 2017; Singer and Michaelides, 2017; Cuthbert et al., 2019; Adloff et al., 2022; Kipkemoi et al., 2021; Quichimbo et al., ~~2021~~2021a, 2023). In drylands high-intensity, short-lived, and localised convective rainfall events (or longer duration rainfall events of sufficient intensity) are a vital source of moisture, as they can produce sufficient rainfall to overcome the high evaporative demand, resulting in high infiltration rates, enhanced soil moisture, and groundwater recharge (Taylor et al., 2013; Batalha et al., 2018; Kipkemoi et al., 2021; Adloff et al., 2022; Boas and Mallants, 2022). These intense convective events can also yield additional focused recharge when rainfall is heavy enough to generate surface runoff, where a certain proportion of runoff can enter dry ephemeral river channels and generate substan-

tial flows that can lead to localised transmission losses (Osborn, 1983; Scanlon et al., 2006; Cuthbert et al., 2016, 2019; Singer and Michaelides, 2017; Seddon et al., 2021; Zarate et al., ~~2022~~2021; Quichimbo et al., ~~2021~~2021b, 2023). On the contrary, if rainfall is low-intensity and long-duration, evaporative losses will be higher and rainfall will be quickly returned to the atmosphere (Batalha et al., 2018; Kipkemoi et al., 2021). Furthermore, precipitation dry spell length can impact evaporative losses and influence the antecedent soil moisture conditions that can govern hydrological responses to rainfall (Zhang and ~~Shilling~~Schilling, 2006; Nazarieh et al., 2018; Schoener and Stone, 2019; Schoener, 2021; Boas and Mallants, 2022).

The non-linear relationship between rainfall and hydrology in drylands makes it challenging to understand how anthropogenically-driven changes in rainfall and PET will impact dryland water security in the HOA, particularly as the GCMs and RCMs often used to make future assessments of dryland water resources (Crosbie et al., 2010; McKenna and Sala, 2017; Razack et al., 2019; Cook et al., 2022) poorly represent convective processes. Climate models with horizontal grid spacings larger than 10 km must rely on parameterisation schemes to estimate the average effects of convection (Prein et al., 2015; Kendon et al., 2017), which results in "parameterised" climate models systematically overestimating the frequency of low intensity rainfall events, simulating rainfall too early in the day, and underestimating the magnitude of extreme rainfall (Stephens et al., 2010; Stratton and Stirling, 2012; Prein et al., 2013; Ban et al., 2014; Kendon et al., 2019; Finney et al., 2019). As water partitioning in drylands is highly sensitive to rainfall characteristics, such biases could have significant impacts on future water resource projections.

While the primary focus of this study is how model representation of convection influences rainfall characteristics and subsequent water partitioning, we also consider the role of ~~potential evapotranspiration (PET)~~PET. In drylands, actual evapotranspiration (AET) is primarily constrained by soil moisture rather than atmospheric demand (i.e., water rather than energy limited), meaning that the direct hydrological effects of PET are typically small relative to those of precipitation (Vicente-Serrano et al., ~~2019~~2020). However, PET can still exert an important influence on vegetation and soil moisture dynamics, and plays a key role in land–atmosphere feedbacks (Seneviratne et al., 2010). Moreover, the strong temporal variability of rainfall means that drylands are not always water-limited – for example, during or following high-rainfall periods when PET can shape antecedent soil moisture conditions before the next rainfall event, as soils dry rapidly between events (Zhang and ~~Shilling~~Schilling, 2006; Nazarieh et al., 2018; Cuthbert et al., 2019; Schoener and Stone, 2019; Schoener, 2021; Boas and Mallants, 2022).

One approach to improving the representation of dryland rainfall characteristics is using convection-permitting models (CPMs), which are run at sufficiently high resolution

($< 5$ km) to explicitly resolve deep convection and represent a step-change in climate modelling capabilities (Clark et al., 2016). Here, we assess whether CPMs can better capture dryland rainfall characteristics relative to traditional parameterised climate models, if there are differences in PET dynamics between CPMs and parameterised climate models, and critically if dryland water partitioning is sensitive to climate model representation of convection (via its impact on rainfall and PET).

We compare two climate models that share the same underlying model physics and driving GCM, but one is a high-resolution ($\sim 4.5$ km) convection-permitting model that explicitly represents convection (CP4A), while the other is a regional climate model ($\sim 25$ km) that parameterises the average effects of convection (P25). While both models simulate comparable annual and seasonal rainfall totals (Kendon et al., 2019; Wainwright et al., 2019), the impact of convective representation on rainfall characteristics could have implications for how moisture propagates through a dryland hydrological system. Furthermore, to our knowledge no studies to date have assessed how model representation of convection affects PET when it is externally calculated from model atmospheric variables (using the Penman-Monteith equation for reference crop evapotranspiration – see Sect. 2.2 and Eq. 1). Accordingly, we consider three key questions:

1. Does explicitly resolving convection in climate models improve the representation of dryland rainfall characteristics and hydrologically relevant rainfall metrics in the Horn of Africa?

2. Does convective representation affect simulated PET dynamics in the Horn of Africa?

3. Does the impact of convective representation on rainfall characteristics and PET influence how water is partitioned between different stores in drylands?

To do this we compare our CPM (CP4A) and a conventional RCM (P25) to high-resolution hourly gridded satellite derived datasets of rainfall (IMERG) and PET (hPET), focusing on the ability of each model to capture key hydrologically relevant dryland rainfall and PET metrics. Then to assess hydrological sensitivity to rainfall characteristics and PET, we force a 1-D vadose zone hydrological model (Hydrus 1-D) with climate model rainfall and PET at four locations along an aridity gradient across the HOA.

## 2 Methods

### 2.1 Climate Model Description

This study utilises data from two pan-African climate models run under the Future Climate for Africa (FCFA) Improving Model Processes for African cLimAte (IMPALA) project (Stratton et al., 2018; Kendon et al., 2019). Both the Convection-Permitting Model for Africa (CP4A) and the 25 km regional model (P25) are configurations of the Met Office Unified Model (Walters et al., 2017). CP4A is run at a horizontal resolution of 4.5 km × 4.5 km in CP4A and 26 × 39 km in P25 and is available for a "historical" (1997–2007) and "future" (2095–2105) time slice. In this study we are only focusing on the "historical" time slice, which is statistically consistent with the observed climate between 1997–2007. So while it simulates a plausible sequence of weather consistent with the observed large-scale climate of 1997–2007, it does not reproduce the actual day-to-day or event-level sequence of observed weather over this period. However, ~~tt~~ it realistically captures annual totals, rainfall seasonality, and the distribution of rainfall characteristics, but the precise timing, duration, spatial pattern, and intensity of individual storms do not correspond to those seen in IMERG. Both CP4A and P25 utilise the same domain, aerosol and land surface forcing, and are forced by the 25 km Unified Model GCM at their lateral boundaries and Reynolds sea-surface temperature observations (Reynolds et al., 2007b; Walters et al., 2017; Stratton et al., 2018). While CP4A and P25 use different cloud and blended boundary layer schemes, and CP4A also considers moisture conservation, the biggest difference is that the deep convection parameterisation scheme has been "switched off" in CP4A (Stratton et al., 2018).

While CP4A is clearly a potential step forward, it remains subject to certain limitations, such as the use of a uniform sandy soil across the entire domain. For water limited regions such as the HOA, soil moisture (partly a function of soil properties) directly regulates evapotranspiration and can contribute to precipitation via moisture recycling (Seneviratne et al., 2010), so a uniform soil map risks poor representation of the soil moisture–precipitation feedbacks critical to inducing convective rainfall and a realistic spatial pattern of rainfall (Taylor et al., 2011, 2012; Hsu et al., 2017; Zhou et al., 2021). There are also clearly limitations with results based upon the use of a single climate model, although there is a consistent pattern among CPMs in being able to better capture rainfall characteristics (compared to non-convection-permitting models) which increases confidence in CP4A (Kouadio et al., ~~2018~~2020; Luu et al., 2022).

### 2.2 Climate Data

To establish whether CP4A can better capture dryland rainfall characteristics and PET dynamics (relative to P25), we compared the "historical" runs of both climate models (statistically consistent with the observed 1997–2007 climate) to the gridded Integrated Multi-satellitE Retrievals for GPM (IMERG; Huffman et al., ~~2012~~2020) rainfall product and an hourly potential evapotranspiration dataset (hPET; Singer et al., 2021).

IMERG provides rainfall estimates at 0.1° (spatial) and 30 min (temporal) resolution using space-based radar, passive microwave, infrared, and rain gauge data from the Global Monthly Precipitation Climatology Centre (Huffman et al., 2020), although in the HOA there are few rain gauges and there are persistent gaps in any available data (Cocking et al., 2024). IMERG compares well to other satellite-derived rainfall datasets and gauge data in the HOA where such data is available (Dezfuli et al., 2017; Cocking et al., 2024). Critically it is the high spatial and temporal resolution of IMERG that means it is the most appropriate for evaluating dryland rainfall metrics at a regional scale (Ageet et al., 2022), however, it is only available from June 2000, so we can only compare CP4A/P25 to 6.5 years of rainfall data.

The hPET global hourly PET dataset (Singer et al., 2021) is derived from ERA5-Land variables (Muñoz Sabater ~~,~~ ~~2024~~et al., 2021) using the Food and Agriculture Organisations' (FAO) Penman-Monteith equation (Eq. 1) for reference crop evapotranspiration (Allen et al., 1998). hPET is available at a high spatial (0.1°) and temporal (hourly) resolution and can capture both diurnal and seasonal variability in atmospheric evaporative demand (Singer et al., 2021). We computed PET from CP4A and P25 outputs using the same FAO equation (Eq. 1; Blake, 2025 TS2). Note while most variables could be directly outputted from the model, this did require converting 10 m wind speeds to 2 m (Eq. 2) and estimating relative humidity from specific humidity, surface pressure, and air temperature. Full details of the PET calculation (Eqs. 5–14 TS3) and rationale behind this approach is provided in Appendix A.

Climate model PET at an hourly resolution ($t$) at each pixel ($x$) in our domain was calculated using Eq. (1).

$$\text{hPET}_{x,t} = \frac{0.408\Delta\left(R_n - G + \gamma\left(\frac{37}{T_a+273}\right)u_2\left(e_s - e_a\right)\right)}{\Delta + \gamma\left(1 + 0.34u_2\right)} \quad (1)$$

Where $R_n$ is hourly net radiation ($\text{MJ m}^{-2}$), $G$ is the soil heat flux ($\text{MJ m}^{-2}$), $\gamma$ is the psychometric constant ($\text{kPA °C}^{-1}$), $\Delta$ is the slope of saturation vapour pressure ($\text{kPA °C}^{-1}$), $T_a$ is hourly air temperature (°C), $e_s$ is saturation vapour pressure (kPa), $e_a$ is the actual vapour pressure, and $u_2$ is wind speed ($\text{m s}^{-1}$) at 2 m above the land surface.

Surface net solar radiation ($\text{J m}^{-2}$), surface net thermal radiation ($\text{J m}^{-2}$), atmospheric surface pressure (Pa), and 2 m air temperature (K) can all be directly outputted from CP4A/P25. Wind speed is also directly outputted from CP4A/P25 but is only available at 10 m above the surface, so zonal and meridional wind speed was converted to the required 2 m value using the logarithmic velocity profile above a short grass surface (Eq. 2):

$$u_2 = u_z\left(\frac{4.87}{\ln\left(67.8z - 5.42\right)}\right) \quad (2)$$

Where $u_z$ is the wind speed at height $z$ above the land surface (10 m in this case) computed as (Eq. 3):

$$u_z = \sqrt{u^2 + v^2} \quad (3)$$

Since the 2 m dew point temperature is not directly outputted from CP4A/P25, it was calculated using the 2 m air temperature and relative humidity (also not directly outputted by CP4A/P25), with relative humidity calculated using 2 m air temperature, surface pressure, and specific humidity (see Appendix A Eqs. A1–A4). For a breakdown of how the saturation vapour pressure ($e_s$), actual vapour pressure ($e_a$), slope of saturation vapour pressure ($\Delta$), net radiation ($R_n$), and the soil heat flux ($G$) used in Eq. (1) were calculated please refer to Eqs. (A5)–(A10) given in Appendix A.

## 2.3 Dryland Rainfall Metrics

Several studies have demonstrated that CP4A better captures rainfall frequency, extremes, the diurnal cycle, and the spatial structure of rainfall events (Ban et al., 2014; Prein et al., 2015; Kendon et al., 2017, 2019; Finney et al., 2019, 2020). Here we benchmark CP4A and P25 against a series of metrics that are explicitly indicative of the key dryland rainfall characteristics that are important in hydrological partitioning:

1. Rainfall mode – Percentage of annual rainfall from ~~high intensity~~ high-intensity ($> 95$th percentile IMERG rainfall) versus low-intensity drizzle ($\leq 1\,\text{mm h}^{-1}$).

2. Maximum dry spell length – average longest run of dry days in any given year.

3. Extreme rainfall magnitude – 99th percentile of wet hour rainfall in the dominant rainy season.

For all the above metrics we define a wet hour as any hour with $\geq 0.1\,\text{mm}$ of rainfall. For the extreme rainfall metric we define the dominant ~~rain~~ rainy season at each grid cell based on the percentage of annual rainfall delivered in each of the following seasons: January–February, March–May, June–September, and October–December. For dryland regions which have a bimodal regime the wettest season will either be MAM or OND, and for the more humid Ethiopian Highlands it is more likely to be JJAS (Fig. 1 and Appendix B – Fig. B1)

To capture the high-intensity, low-duration nature of rainfall in drylands, our analysis was conducted at hourly rather than three-hourly resolution as has been done elsewhere (~~Bethou~~ Berthou et al., 2019; Kendon et al., 2019; Finney et al., 2019, 2020), as resampling to a lower temporal resolution dampens the intensity of rainfall and impacts water partitioning (Batalha et al., ~~2021~~2018; Kipkemoi et al., 2021). To ensure consistency, all datasets were re-gridded using first order conservative regridding to the P25 grid ($26 \times 39\,\text{km}$). We conducted all analysis using the "historical" CP4A/P25

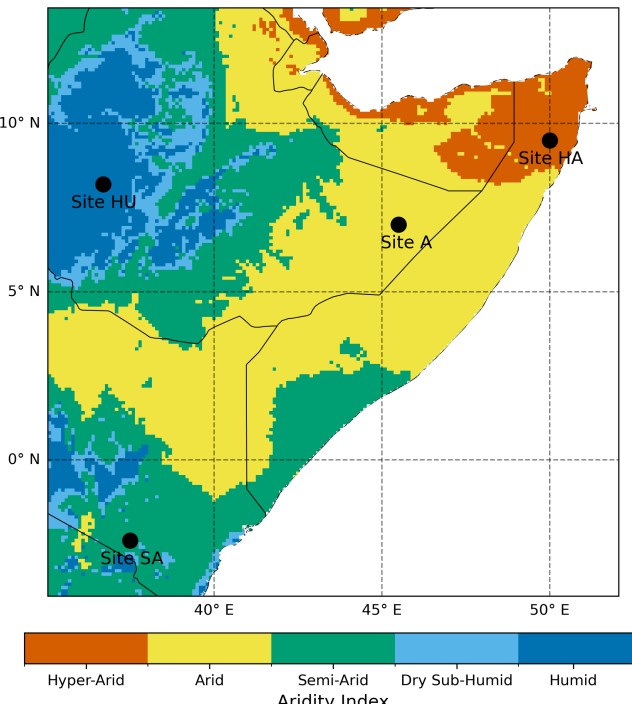

**Figure 1.** Horn of Africa Aridity Index and Study Site Locations. Locations of the four Hydrus locations plotted against aridity index values taken from CGIAR-CSI. Sites are found in the humid Ethiopian Highlands (Site HU), semi-arid southern Kenya (Site SA), arid eastern Ethiopia (Site A), and hyper-arid northern Somalia (Site HA).

runs, so all results and figures presented refer to "historical" rainfall data which is statistically consistent with observed climate between 1997–2007.

## 2.4 Hydrological Modelling

### 2.4.1 Hydrus 1-D

We used Hydrus 1-D v4.17 (Šimunek et al., 2012) to simulate dynamic changes in infiltration, surface runoff, evaporation, transpiration, soil moisture, and bottom drainage when forced with each climate model rainfall and PET. Hydrus 1-D uses time series of rainfall, PET, and Leaf Area Index (LAI) to numerically solve a version of the Richards equation (Richards, 1931). It can simulate vertical water redistribution in the soil subsurface under a wide range of different climatic, soil, and vegetation conditions, and has been widely used to understand soil moisture, evapotranspiration, and groundwater recharge in both ~~arid~~ dryland and humid landscapes (Leterme et al., 2012; McKenna and Sala, 2017; Batalha et al., 2018; Rodriguez et al., 2020; Boas and Mallants, 2022; Corona and Ge, 2022). However, it does not consider lateral flows, and larger scale moisture redistribution processes.

**Table 1.** Climate classifications based on aridity index thresholds taken from Mirzabaev et al. (2019).

| Climate Type | Aridity Index |
|---|---|
| Hyper-Arid | AI < 0.05 |
| Arid | $0.05 \leq AI < 0.2$ |
| Semi-Arid | $0.2 \leq AI < 0.5$ |
| Dry Sub-Humid | $0.5 \leq AI < 0.65$ |
| Humid | $AI \geq 0.65$ |

Hydrus 1-D requires parameterisation of soil hydraulic properties, which are typically obtained from in-situ measurements or derived using soil textures filtered through a pedotransfer function (Van Genuchten, 1980). It also includes a sink term to account for root water uptake (also referred to as transpiration), which is estimated using the water stress response function detailed by Feddes (1978). For further details on Hydrus 1-D please refer to (Šimunek et al., 2012).

### 2.4.2 Hydrological Study Sites

We ran four 1-D vadose-zone hydrological simulations along an aridity gradient across the HOA, ranging from the humid Ethiopian Highlands to hyper-arid Northern Somalia (Fig. 1). We classify aridity based on aridity index (AI = P / PET) values taken from the CGIAR-CSI (Consortium of International Agricultural Research Centres' Consortium for Spatial Information) (Zomer et al., 2007) using the classification of Mirzabaev et al. (2019).

The Aridity Index (AI) is a numerical indicator of climatic aridity based on long-term precipitation deficits relative to atmospheric water demand (Eq. 4):

$$AI \, (AridityIndex) = MAP/MAE \qquad (4)$$

Where MAP is mean annual precipitation and MAE is mean annual potential evaporation, CGIAR-CSI calculate both MAP and MAE using data obtained from WorldClim Global Climate Data (Hijmans et al., 2005). CGIAR-CSI outputs AI values at 1 km resolution, which can be used to define the climate type based on the climate classification of Mirzabaev et al. (2019) as shown in Table 1.

Figure 1 shows the climatic zones of the HOA along with the locations of our four sites, with one site in each major aridity zone: humid (9.7 % of land mass), semi-arid (31.8 %), arid (43.6 %), and hyper-arid (7.7 %). For this study we define drylands as any grid cell with an AI ≤ 0.5 rather than 0.65, so our "dryland" region refers to any grid cell identified as Hyper-Arid, Arid, or Semi-Arid (not Dry Sub-Humid).

For each hydrological site, we tried to choose locations within each aridity classification where mean annual rainfall and PET were comparable, but in lieu of perfect matches, we selected locations where P25 simulated higher mean annual rainfall (Table 2). At each site (represented by a single climate model grid cell) P25 simulated 15 % (humid),

Track changes document – Do not use for proofreading

**Table 2.** Mean annual rainfall, PET, and vegetation type at our four study sites. Rainfall and PET simulated by CP4A are in bold, and P25 in non-bold. Vegetation type is taken from the iSDAsoil dataset (iSDA, 2024). All values are calculated using the "historical" CP4A/P25 simulations. These "historical" simulations were designed to be statistically consistent with the observed climate between March 1997 and February 2007.

| Site | Rainfall (CP4A) | PET (CP4A) | Vegetation |
|------|-----------------|------------|------------|
| Site HU (humid) – Ethiopian Highlands | 2000 mm (**1730 mm**) | 1290 mm (**1400 mm**) | Maize |
| Site SA (semi-arid) – Southern Kenya | 670 mm (**600 mm**) | 1620 mm (**1580 mm**) | Shrubs |
| Site A (arid) – Eastern Ethiopia | 430 mm (**350 mm**) | 2180 mm (**1920 mm**) | Shrubs |
| Site HA (hyper-arid) – Northern Somalia | 240 mm (**180 mm**) | 1970 mm (**1720 mm**) | Bare Soil |

12 % (semi-arid), 23 % (arid), and 33 % (hyper-arid) higher total annual rainfall. Choosing locations where P25 simulates higher total rainfall ensures that if fluxes such as soil moisture or bottom drainage are higher when forcing Hydrus with CP4A rainfall, it is reflective of differences in rainfall characteristics rather than simply higher annual rainfall totals. P25 also simulated 3 %, 14 %, 15 % higher total annual PET at our dryland sites (SA, A, and HA respectively), while PET is 9 % higher using CP4A at our humid site (HU). However, this still means that at all sites the ratio of P/PET is higher using P25. Please refer to Appendix B for mean monthly rainfall at each site (Fig. B1), mean monthly PET (Fig. B2), and the raw hourly time series of rainfall at each site (Fig. B3).

It is worth noting that we did not drive Hydrus with IMERG rainfall at each study site. The aim of this study was not to evaluate which model best reproduces observed hydrological fluxes, but rather to isolate how differences in convective representation between CP4A and P25 influences rainfall characteristics and subsequent water partitioning.

### 2.4.3 Hydrological Model Set Up, Data, and Sensitivity

~~This~~ The aim of this study is to use a series of experimental, one-dimensional Hydrus simulations to examine the impact of convective representation on rainfall characteristics and subsequent water partitioning when rainfall is propagated vertically through the vadose zone at a 1-D scale. Although one-dimensional point-based hydrological modelling is a clear simplification of the hydrological system and does not represent watershed-scale surface and sub-surface lateral flows, the focus on the 1-D water balance and processes was a deliberate decision to ensure we could most effectively isolate the impact of rainfall characteristics – specifically the representation of convection on rainfall intensity-duration – on vertical vadose zone hydrological partitioning, without the complexity that would be introduced by lateral and non-local processes if modelling was carried out at a basin or regional scale. We are not dismissing the importance of overland flow generation and runoff as significant hydrological processes in dryland catchments, we exclude their consideration to simply understand the balance between evapotranspiration, soil moisture and deeper drainage in response to differing rainfall characteristics. And while our simulations are experimental, they use plausible soil hydraulic and vegetation parameters where available.

All Hydrus simulations utilised a 3 m soil profile (preliminary simulations suggested minimal water fluxes below this depth at some locations) with a free draining bottom boundary (no interactions between water table and soil profile above). Although a simplification, this setup is not unreasonable as the water table is commonly deep across the Horn of Africa (Bonsor and MacDonald, 2011; Fan et al., 2013) and in drylands generally. However, this does mean that the simulated bottom drainage is not necessarily indicative of groundwater recharge, as it is unlikely the water table would be so shallow (Bonsor and MacDonald, 2011; Fan et al., 2013), and there is robust evidence that dryland acacia shrubs (dominant shrub species in the HOA) are particularly deep rooted and are capable of extracting water directly from the water table (Stone and Kalisz, 1991; Maeght et al., 2013; Shadwell and February, 2017).

For the top boundary layer, we used atmospheric boundary conditions with surface runoff, meaning where model rainfall exceeds the infiltration capacity of the topsoil, water is ~~re-routed~~ rerouted as runoff and cannot enter the soil profile (Fig. 2). For all simulations the soil profile was discretized in 2 cm intervals, with a minimum and maximum time step interval of 5 s to 20 d (high max time step needed as long periods with no rainfall in drylands). This ensured the model converged (using default Hydrus water content and pressure head tolerances) within 10 timesteps (Hydrus default) and relative mass balance errors remained below 1 %. All model runs at each site were initially forced using the mean hourly IMERG and hPET climatology (over 2000–2007) until steady state conditions were reached and relative mass balance errors were below 0.1 % (6–15 years depending on study site).

Soil hydraulic properties were calculated using Genuchten-Mualem (Van Genuchten, 1980) equations based on soil texture values taken from the iSDAsoil database (iSDA, 2024), which applies a multiscale ensemble machine learning approach to a range of fine and coarse-scale satellite observations to predict soil properties at 30 m resolution at two depth intervals (0–20 and 20–50 cm) (Hengl et al., 2021). While robust estimates of water movement would

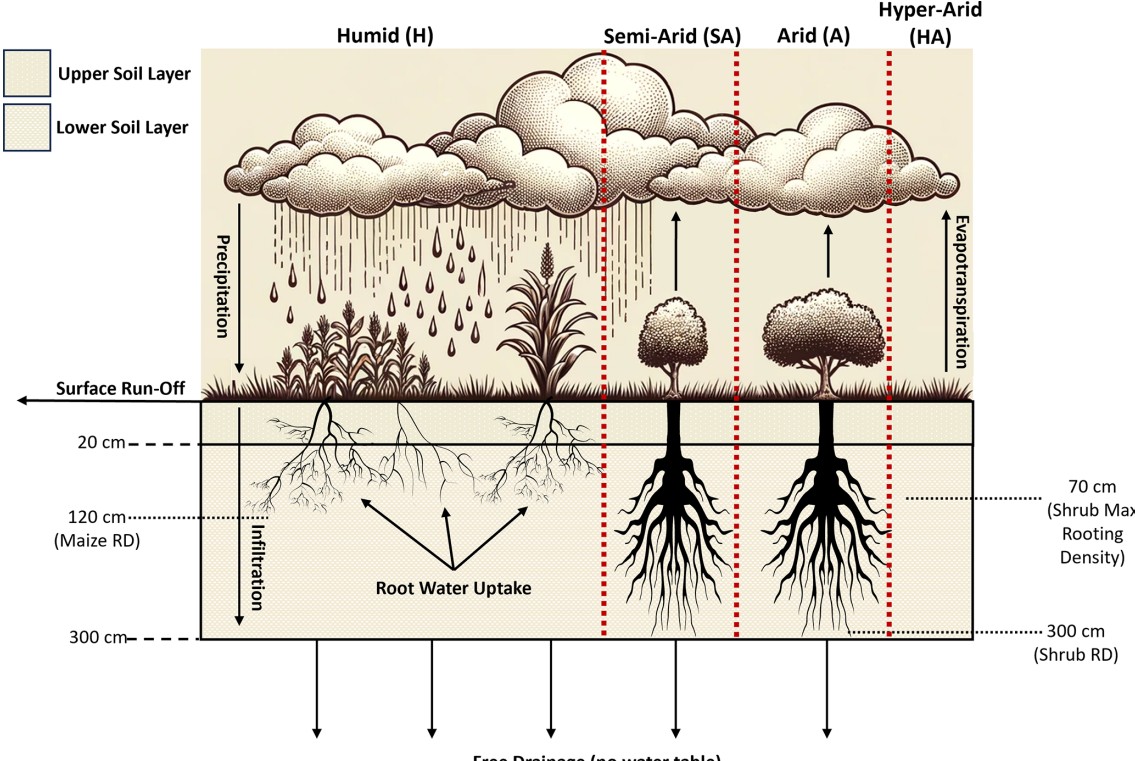

**Figure 2.** Hydrus Simulations Schematic. Conceptual schematic of how water is partitioned within Hydrus 1-D at any given timestep within the simulation. Please note that in the figure shrubs, maize, and bare soil are represented, whereas only one vegetation type can be modelled in any one simulation. Here the schematic is divided to represent the model set up for each site-specific simulation. For all sites the soil profile is divided into two layers (boundary is marked at 20 cm) and is discretised at a resolution of 2 cm. At Site HU, maize roots to a depth of 120 cm, while at Sites SA and A shrubs utilise moisture through the entire 300 cm profile. At Site HA we assume grass cover is extremely sparse, so here we only consider bare soil evaporation and no transpiration.

require soil textures for the entire profile, here we divided our 3 m profile into two layers following the iSDAsoil data depth intervals (which uses 0–20 and 20–50 cm). This means we divide our profile into an upper (0–20 cm) and lower layer (20–300 cm), a significant simplification as it assumes that soil properties from 50–300 cm ~~follows~~ follow those at shallower depths (20–50 cm).

To calculate transpiration Hydrus needs land cover (Table 2) and leaf area index (LAI) data, which were taken from iSDA (as of 2019) and the National Centers for Environmental Information AVHRR LAI dataset (Vermote, 2019) respectively. Hydrus uses the Feddes' (1978) approach to estimate transpiration (referred to as root water uptake within Hydrus) under various pressure heads (water stress) and root densities. For Site HU (humid) we utilised Feddes' parameters for maize (from the internal Hydrus database – Wesseling ~~et al. (1991) )~~ and Brandyk, 1985) and set the maximum rooting depth to 1.2 m b.g.l. (Zinyengere et al., 2011). Shrubs are not included in the internal Hydrus vegetation database, and there is limited information on Feddes' parameters for dryland (acacia) shrubs, so we combined data from available literature to estimate a reasonable parameter set (Appendix C

– Table C1) (Xia and Shao, 2008; Sela et al., 2015; Watson, 2015). As dryland shrubs tend to be deep-rooted (Stone and Kalisz, 1991; Maeght et al., 2013; Shadwell and February, 2017), we assumed they can utilise moisture from the entire soil profile and specified maximum root density at ~ 0.7 m b.g.l., which tends to be the depth at which shrub root water uptake is greatest (Geißler et al., 2019).

Given the uncertainty with Feddes' acacia shrub parameters, at Sites SA (southern Kenya) and A (eastern Ethiopia) (site HA is bare soil), we ran additional Hydrus simulations using a range of shrub Feddes' parameters (Appendix C – Table C1) as given by Sela et al. (2015). Within the range given by Sela et al. (2015) we used the upper parameter set as our "default" run, as the wilting point corresponded better with other published data (Xia and Shao, 2008; Watson, 2015).

At all sites, we also ran Hydrus simulations using low (lowK) and high (highK) hydraulic conductivity soil parameters (Appendix C – Table C2), and to assess dryland hydrological sensitivity to PET, we also forced Hydrus with climate model rainfall but replaced climate model-derived PET with gridded hPET values (see Sect. 2.2). However, unless stated otherwise, all results refer to simulations forced using

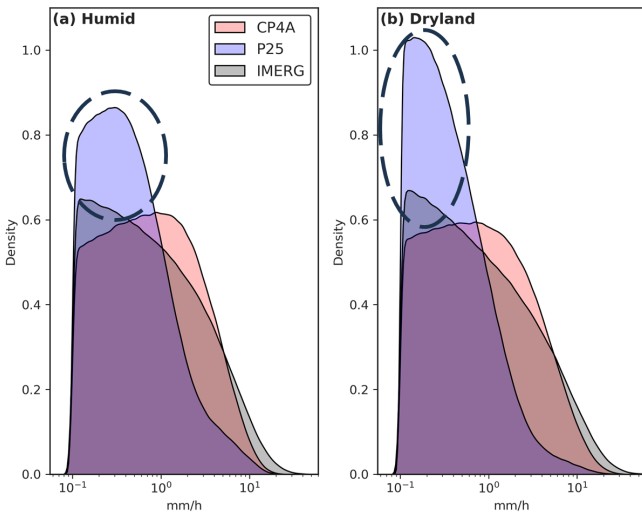

**Figure 3.** Rainfall KDE Plots. Kernel density estimate (KDE) plots of CP4A, P25, and IMERG hourly rainfall ($\geq 0.1\,\mathrm{mm\,h^{-1}}$) in humid **(a)** (AI $\geq 0.65$) and dryland **(b)** (AI $< 0.5$) regions of the Horn of Africa. Dashed circles highlight the "drizzle" effect seen in P25. Plots cover period of June 2000 to February 2007, which is the period where CP4A/P25 and IMERG overlap. These "historical" CP4A/P25 simulations were designed to be statistically consistent with the observed climate between March 1997 and February 2007.

"historical" climate model rainfall/PET and the default soil hydraulic and Feddes' parameters given in Appendix C – Tables C1 and C2.

## 3 Results

Please note that all results discussed refer to the "historical" CP4A/P25 runs, we do not use the "future" runs at any point in this study.

### 3.1 Rainfall

Figure 3a–b shows rainfall intensity distributions for humid (AI $\geq 0.65$) and dryland regions (AI $< 0.50$) in the HOA, based on all rainfall hours $\geq 0.1\,\mathrm{mm\,h^{-1}}$. The plots highlight the "drizzle" effect associated with parameterised climate models (shown by the large frequency peaks in rainfall intensities between $10^{-1}$ and $10^{1}\,\mathrm{mm\,h^{-1}}$ – dashed black circles), with P25 overestimating the frequency of rainfall events $\leq 1\,\mathrm{mm\,h^{-1}}$ in both regions, but particularly drylands (Fig. 3b). CP4A does not simulate the same "drizzle effect" in drylands, although (in both humid and dryland regions) CP4A simulates fewer rainfall events with an intensity of $> 10\,\mathrm{mm\,h^{-1}}$ compared to IMERG.

We used the Kolmogorov–Smirnov (KS) test with a null hypothesis that the modelled distribution of hourly rainfall intensities (based on all hours with rainfall $\geq 0.1\,\mathrm{mm\,h^{-1}}$) is drawn from the same distribution as IMERG. Both P25 and

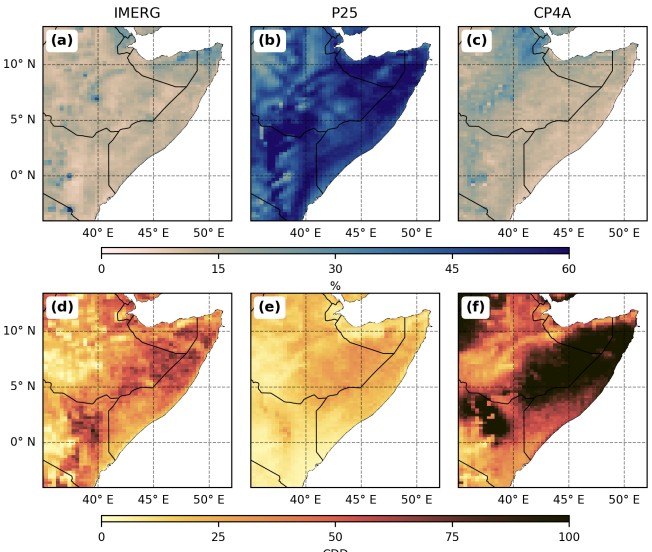

**Figure 4.** Percentage of Annual Rainfall delivered as drizzle and maximum number of consecutive dry days. Percentage of mean annual rainfall that falls as "drizzle", where "drizzle" is any rain hour with an intensity $\leq 1\,\mathrm{mm\,h^{-1}}$ **(a–c)** for IMERG **(a)**, P25 **(b)**, and CP4A **(c)**. **(d–f)** Mean annual maximum dry spell length, where a dry hour is any hour that receives less than 0.1 mm of rain, for IMERG **(d)**, P25 **(e)**, and CP4A **(f)**. Plots cover rainfall recorded/simulated between June 2000 and February 2007, which is the period where CP4A/P25 and IMERG overlap. These "historical" CP4A/P25 simulations are statistically consistent with the observed climate between March 1997 and February 2007.

CP4A show statistically significant differences from IMERG ($p < 0.05$), but the KS statistic is markedly lower for CP4A (0.03) than for P25 (0.24), indicating CP4A better matches the observed distribution (IMERG). Also, while Fig. 3 aggregates data across the entire study period, the "drizzle" effect seen in P25 simulations remains consistent between years and across each of the four seasons in the region (Appendix D – Fig. D1).

The tendency for P25 to overestimate the frequency of light rainfall events means most annual rainfall is delivered via events with a magnitude of $\leq 1\,\mathrm{mm\,h^{-1}}$ (Fig. 4a–c) across the HOA. In humid regions, on average P25 simulates 39.0 % of annual rainfall falling as "drizzle" versus 17.1 % and 13.6 % in CP4A and IMERG respectively. This bias is even more pronounced in dryland areas, where the median proportion of annual rainfall falling as drizzle is 51.5 % in P25 versus 14.1 % and 13.0 % for CP4A and IMERG. Apart from a few isolated locations, no areas in the drylands receive more than 20 % of rainfall via "drizzle" in CP4A and IMERG (Table 3).

The "drizzle" bias also means P25 underestimates maximum dry spell length (CDD – consecutive dry days) compared to IMERG and CP4A (Fig. 4d–f). Both climate models replicate the spatial pattern of CDD observed in IMERG,

**https://doi.org/10.5194/**

**Table 3.** Interquartile ranges of the percentage of rainfall delivered as "drizzle" and "heavy" rain, extreme rainfall (99th wet season percentiles), and the maximum number of consecutive dry days (CDD) in humid (AI ≥ 0.65) and dryland (AI < 0.5) regions of the HOA (see Fig. 1). Where "drizzle" is any rain hour with an intensity ≤ 1 mm h$^{-1}$, and the "heavy" rain threshold is spatially variable and is based on the 95th percentile of hourly wet season IMERG rainfall (see Fig. 5d). All values are calculated based on hourly rainfall recorded/simulated between June 2000 and February 2007, which is the period where CP4A/P25 and IMERG overlap. These "historical" CP4A/P25 simulations were designed to be statistically consistent with the observed climate between March 1997 and February 2007. n/a = not applicable.

| Humid | Drizzle (% of annual rainfall) | Heavy Rainfall (% of annual rainfall) | 99th Percentiles (mm h$^{-1}$) | Max CDD |
|---|---|---|---|---|
| IMERG | 12.0–16.2 | ~~Nn/Aa~~ | 14.2–19.3 | 7–19 d |
| P25 | 30.6–46.8 | 5.1–13.4 | 5.4–9.2 mm h$^{-1}$ | 7–14 d |
| CP4A | 14.7–20.0 | 8.8–13.3 | 8.4–11.1 mm h$^{-1}$ | 24–73 d |
| Drylands | | | | |
| IMERG | 11.6–14.8 | ~~Nn/Aa~~ | 12.4–16.3 mm h$^{-1}$ | 27–51 d |
| P25 | 45.2–56.6 | 3.8–11.0 | 4.6–7.1 mm h$^{-1}$ | 11–24 d |
| CP4A | 12.5–16.6 | 13.5–25.4 | 10.9–14.4 mm h$^{-1}$ | 48–95 d |

where CDD is higher in drylands (compared to the humid Ethiopian Highlands), but P25 underestimates CDD length compared to IMERG and CP4A overestimates CDD length (Table 3). In IMERG, median CDD in drylands is ~ 38 d, in P25 it is ~ 18 d and only increases to ~ 20 d in arid regions (AI < 0.2). Whereas in CP4A there are dry spells of over 100 d across large parts of eastern Ethiopia, northern Somalia, and northern Kenya (Fig. 4f). In humid regions CP4A also overestimates CDD length relative to P25 and IMERG.

Figure 5a–c compares the "extreme" rainfall (99th percentile of wet hours in the wettest rainy season) and shows both climate models underestimate the magnitude of wet extremes relative to IMERG. However, the bias is reduced in CP4A, which is in line with other studies using three-hourly data (~~Bethou~~ Berthou et al., 2019; Kendon et al., 2019; Finney et al., 2019, 2020). The improvement in CP4A is more pronounced in drylands (Table 3), whereas differences between CP4A and P25 are more muted in humid regions, where the median 99th percentile value is 36 % higher in CP4A vs. P25 (compared to over 110 % in drylands) and IQR ranges overlap (Table 3). Also, unlike CP4A, P25 simulates higher median 99th percentiles in humid (7.1 mm h$^{-1}$) rather than dryland (5.8 mm h$^{-1}$) regions (see Table 3 for IQR), although the 99th percentile may appear lower in drylands (in P25) because the large number of light rain events (most pronounced in drylands) dilutes the distribution of wet-hour rainfall (not necessarily because absolute rainfall extremes are lower).

A greater proportion of annual rainfall is also delivered via "heavy" rainfall events in CP4A relative to P25 (Fig. 5e–f), most notably in drylands. The percentage of annual rainfall that falls as "heavy" rainfall is based on the 95th percentile of hourly IMERG rainfall rather than using a consistent mm h$^{-1}$ intensity across the entire domain. This means our threshold varies (Fig. 5d) grid cell by grid cell (2.3–10.9 mm h$^{-1}$ in

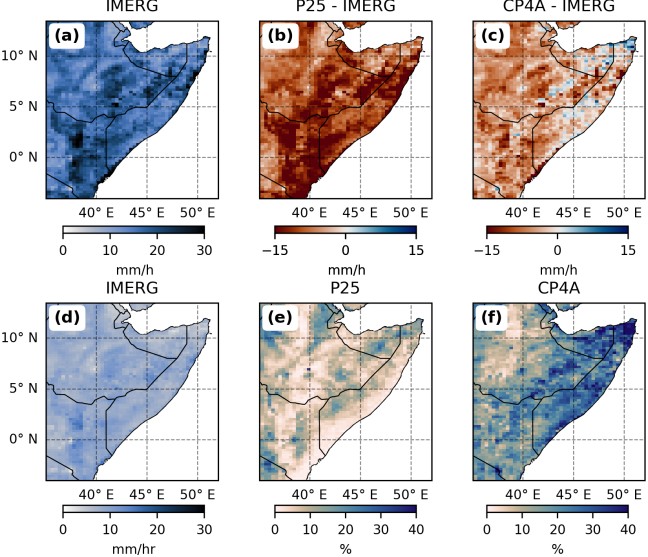

**Figure 5.** 99th percentiles and percentage of rainfall delivered via "heavy" events. Wet season "extreme" precipitation (**a–c**) for IMERG observations (**a**), and anomalies with respect to IMERG for P25 (**b**) and CP4A (**c**). Extreme precipitation is defined as the 99th percentile of all wet hours, where wet hours are any hours that receive ≥ 0.1 mm of rain. (**d–f**) 95th percentile of all wet hours in IMERG observations (**d**), this is used as the "heavy" rain threshold for panels (**e**) and (**f**), which show the percentage of annual rainfall that falls as "heavy" rainfall in P25 (**e**) and CP4A (**f**) respectively. Plots cover rainfall recorded/simulated between June 2000 and February 2007, which is the period where CP4A/P25 and IMERG overlap. These "historical" CP4A/P25 simulations were designed to be statistically consistent with the observed climate between March 1997 and February 2007.

 

humid regions and 2.9–16.2 mm h$^{-1}$ in dryland regions) and ensures we reflect the tendency for a greater percentage of rainfall to fall as "heavy" events in drylands.

In western humid regions both models simulate comparable contributions from "heavy" rainfall, with overlapping IQRs (Table 3), whereas in drylands the IQR of the percentage of annual rainfall falling during "heavy" events is 13.5 %–25.4 % in CP4A versus 3.8 %–11.0 % in P25 (Table 3). In arid regions (eastern Ethiopia and Somalia) the difference is more pronounced, with the median contributions of heavy rainfall being 21.5 % and 7.8 % in CP4A and P25 respectively. CP4A also better replicates the spatial pattern one would expect, where dryland regions in the east tend to receive more rainfall via "heavy" events relative to humid regions in the west (median values of 19.2 % vs. 13.8 %), whereas the opposite is true in P25 (median values of 6.8 % vs. 8.4 %).

## 3.2 Potential Evapotranspiration (PET)

PET derived from climate model atmospheric variables captures the seasonal (Fig. 6e–g) and diurnal cycle (Fig. 6d–f) seen in hPET, although both models simulate earlier peaks in diurnal PET (12:00 vs. 13:00 EAT). Both models also correctly simulate higher evaporative demand in eastern drylands compared to the more humid Ethiopian Highlands (Fig. 6a–c) and produce comparable annual magnitudes (hPET – 1715 mm, CP4A – 1787 mm, P25 – 1883 mm). CP4A simulates marginally higher mean annual PET in humid regions (1416 mm vs. 1387 mm) but lower in arid regions (1901 mm vs. 2027 mm). The most pronounced differences are in arid Somalia and eastern Ethiopia (Fig. 6g), where CP4A exceeds 2000 mm yr$^{-1}$ in 18 % of cells versus 53 % and 43 % in P25 and hPET, respectively.

During dry seasons (JF, JJAS) in arid regions, P25 PET (median = 1098 mm yr$^{-1}$) exceeds CP4A (1000 mm yr$^{-1}$) and hPET (1035 mm yr$^{-1}$). In rainy seasons (MAM, OND), P25 PET remains higher (905 mm yr$^{-1}$) relative to CP4A (860 mm), but lower compared to hPET (912 mm). Multilinear regression (MLR) attributes CP4A-P25 PET differences mainly to temperature and dew point temperature (in both humid and dryland regions), with meridional wind speed also important in drylands (Appendix D – Table D1), with higher P25 wind speeds during JJAS partly driving the higher P25 PET during JJAS (Appendix D – Fig. D2).

## 3.3 Water Partitioning

At all study sites, CP4A and P25 reproduce the seasonal cycle of rainfall and produced broadly comparable seasonal totals (Appendix B – Fig. B1), though P25 delivers higher annual rainfall (Table 4). Both models also produced comparable PET totals and simulated the same seasonal cycle, although P25 simulates substantially higher PET during JJAS at Site HA (Appendix B – Fig. B2). Critically, differences in

rainfall characteristics follow the results discussed above in Sect. 3.1 (Table 4).

Figure 7a–d shows histograms of depth-integrated soil moisture (% soil saturation in entire 300 cm of the soil profile), where soil moisture is expressed as a saturation percentage (given the symbol $\theta s$). Soil saturation reflects the proportion of pore spaces filled with water relative to if all pore space is saturated (e.g. 100 % means all pore space is filled with water, 0 % means all pores are filled with air). Despite simulating lower total rainfall, forcing Hydrus with CP4A rainfall results in higher $\theta s$ at all dryland locations (Table 5), although the IQR of $\theta s$ still overlap at all sites other than Site HA. Focusing on the depth-integrated $\theta s$ masks differences relative to $\theta s$ at specific depths. For example, the relative difference in $\theta s$ between driving Hydrus with CP4A (vs. P25) at Site SA increases from +7.5 % at a depth of 90 cm, to 16.1 % at 2.1 m b.g.l., while at Site HA the difference ranges from 17.0 % (0.2 m b.g.l.) to 23.1 % (1.2 m b.g.l.). There are statistically significant differences ($p$-values are statistically significant to 95 % confidence) in medians (Mann-Whitney) and distributions (Kolmogorov-Smirnov) at all depths (see Appendix E – Table E1 for depth intervals) and all sites (including Site HU), with the KS Test Statistic increasing with depth (Appendix E – Table E1). Calculating the KS Test Statistic using the entire depth-integrated $\theta s$ values shows differences in distributions are more pronounced in drylands (KS Test Statistics: HU– 0.08, SA – 0.29, A – 0.37, HA – 0.92).

Figure 8a–h visualises the deeper wetting fronts in drylands when forcing Hydrus with CP4A, most notably at Sites SA and HA. Shallower wetting fronts in P25 could have ecological implications, for example, at Site A $\theta s$ is above the wilting point (WP) (for shrubs) at a depth of 1.2 m b.g.l. for 44 % of the Hydrus simulation compared to 83 % using CP4A (Fig. 9b). However, it is important to note that this is using the "upper" default Feddes' shrub parameters (Appendix C – Table C1), if one uses the "lower" Feddes' parameters soil moisture never exceeds the wilting point in P25 ("upper" WP = ∼ 25 %, "mid" WP = ∼ 55 %). It is a similar story at Site SA, where $\theta s$ is above WP for 27 % of the time using P25 versus 50 % using CP4A based on the "upper" Feddes' parameter set (Fig. 9a). The large WP range depending on choice of Feddes' parameters highlights the importance of accurately quantifying the Feddes' parameters used for any given vegetation (Fig. 9). However the key finding is that regardless of the Feddes' parameters used, $\theta s$ is either above the wilting point for longer when forcing Hydrus with CP4A, or where both models simulate $\theta s$ as always above the WP, using CP4A means $\theta s$ is high enough for vegetation to transpire at the maximum rate for longer (Site SA – 78 % vs. 57 %, Site A – 13 % vs. 0 %). Median soil moisture is higher when driving Hydrus with CP4A regardless of the Feddes' or soil hydraulic parameters ~~uses~~ used (Appendix E – Tables E2 & E3).

We also forced Hydrus with climate model rainfall and hPET (rather than model PET), where once again at all

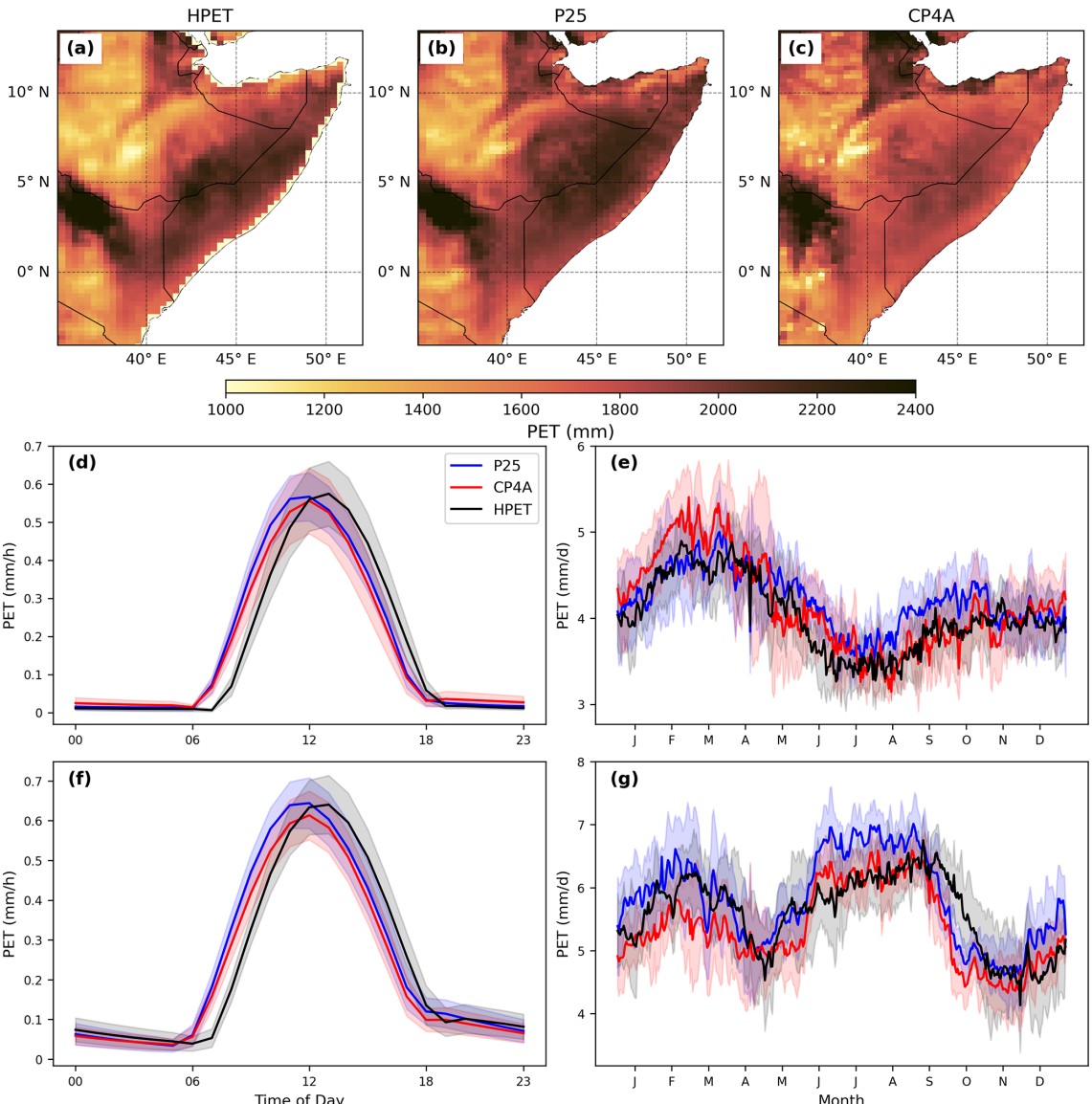

**Figure 6.** Mean annual PET, diurnal PET cycle, and daily climatology of PET. Spatial plots of mean annual PET (top panel) for HPET observations **(a)**, P25 **(b)**, and CP4A **(c)**. Mean diurnal cycle and daily climatology of PET in humid (AI $\geq 0.65$) **(d, e)** and arid to hyper-arid (AI $< 0.2$) **(f, g)** regions of the Horn of Africa for HPET, P25, and CP4A. Plots cover hourly PET recorded/simulated between March 1997 and February 2007. These "historical" CP4A/P25 simulations were designed to be statistically consistent with the observed climate between March 1997 and February 2007.

depths and at every site, there are statistically significant differences in $\theta s$ between Hydrus simulations, although KS test statistics are marginally lower (Appendix E – Table E4). However, PET alone can exert an influence; the percentage of time $\theta s$ is above the WP at Site SA is lower when forcing Hydrus with CP4A rainfall and hPET (41 % vs. 50 %), although $\theta s$ is still above the WP for longer using CP4A compared to P25 (41 % vs. 24 %). The reduction is especially pronounced when considering the WP at Site A, where the percentage of time $\theta s$ is above the WP (using the "upper" Feddes' parameter set) drops from 83 %/44 % to 44 %/21 % for CP4A and

P25 respectively. That there is a reduction in soil moisture when using hPET despite PET being lower than P25 PET highlights how offsets offsets between rainfall and PET can influence the antecedent conditions that govern hydrological responses to rainfall.

Figure 10 details how water is partitioned between surface runoff, evaporation, transpiration, and bottom drainage at our dryland sites when CP4A/P25 rainfall and PET is propagated through Hydrus (for clarity infiltration and Site HU data is not included in Fig. 10 but can be seen in Table 6). Given that $\theta s$ tends to be above the WP for longer in the CP4A runs,

**Table 4.** Mean annual rainfall, "drizzle", "heavy" rain, 99th wet season percentiles, and maximum number of consecutive dry days (CDD) at each of our four sites. The percentage of annual rainfall that falls as "drizzle" or "heavy" rain (based on the 95th percentile of hourly wet season IMERG rainfall) is given in brackets. All values are calculated using the "historical" CP4A/P25 simulations which is designed to be statistically consistent with the observed climate rainfall between March 1997 and February 2007.

|  | Site HU | Site SA | Site A | Site HA |
|---|---|---|---|---|
| P25 Total Rainfall (mm) | 2000 | 670 | 430 | 240 |
| CP4A Total Rainfall (mm) | 1730 | 600 | 350 | 180 |
| P25 Total "Drizzle" (mm) (% of Annual Total) | 650 (33 %) | 358 (53 %) | 234 (54 %) | 133 (55 %) |
| CP4A Total "Drizzle" (mm) (% of Annual Total) | 329 (19 %) | 80 (13 %) | 40 (11 %) | 22 (12 %) |
| P25 Total "Heavy" Rain (mm) (% of Annual Total) | 449 (22 %) | 75 (11 %) | 53 (12 %) | 71 (30 %) |
| CP4A Total "Heavy" Rain (mm) (% of Annual Total) | 378 (22 %) | 149 (25 %) | 170 (49 %) | 148 (82 %) |
| P25 99th Percentile (mm h$^{-1}$) | 8.5 | 5.8 | 5.5 | 4.3 |
| CP4A 99th Percentile (mm h$^{-1}$) | 9.1 | 15.6 | 11.2 | 17.9 |
| P25 Max CDD (days) | 13 | 12 | 8 | 20 |
| CP4A Max CDD (days) | 41 | 43 | 105 | 107 |

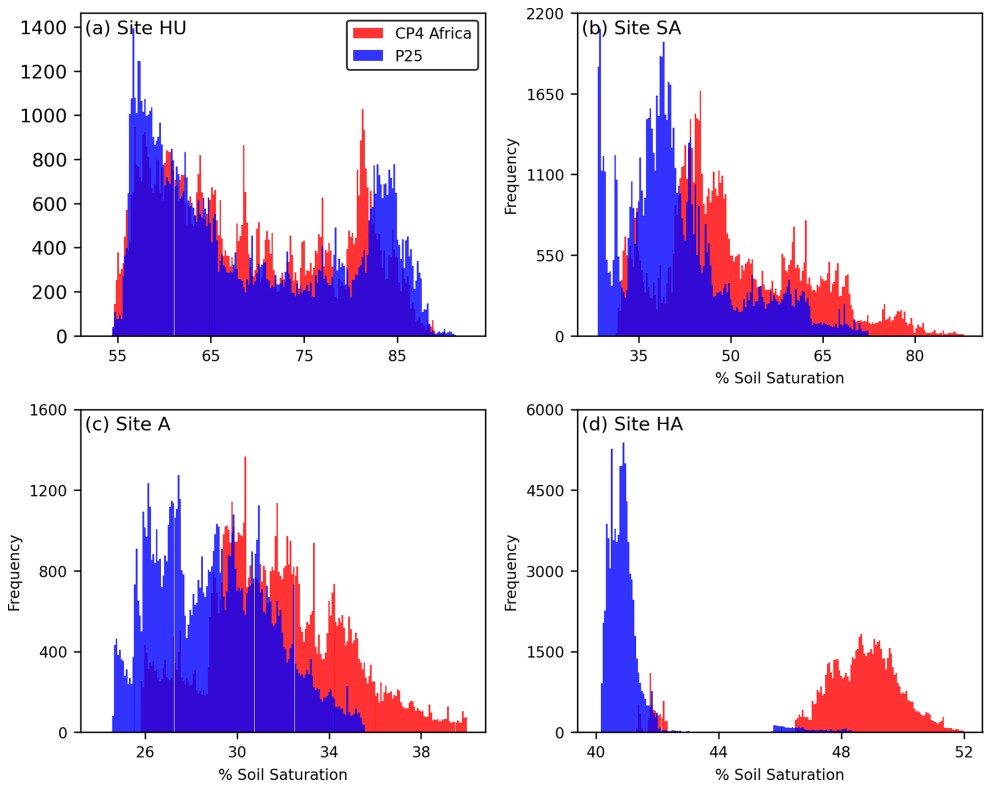

**Figure 7.** Depth Integrated Soil Moisture. Modelled distribution of depth integrated soil moisture in the 3 m soil profile when using P25 (blue) and CP4A (red) rainfall/PET at our humid **(a)**, semi-arid **(b)**, arid **(c)**, and hyper-arid sites **(d)**. Plots computed using hourly Hydrus simulated soil moisture across the "historical" CP4A/P25 runs. These "historical" CP4A/P25 runs were designed to be statistically consistent with the observed climate between March 1997 and February 2007.

and shrubs can transpire at the maximum rate for longer, it is unsurprising that Fig. 10a–f shows substantially higher transpiration at our semi-arid and arid sites when using CP4A (Table 6) despite total infiltration being lower. The shallower wetting fronts when using P25 means evaporative losses are higher (Table 6), with 79 %–83 % of infiltration returned to the atmosphere at sites SA & A respectively, versus 67 %–72 % using CP4A rainfall. Whereas at Site HU evaporative losses are near identical (23.3 % vs. 23.6 %) and transpiration is higher in the P25 Hydrus simulations (Table 6).

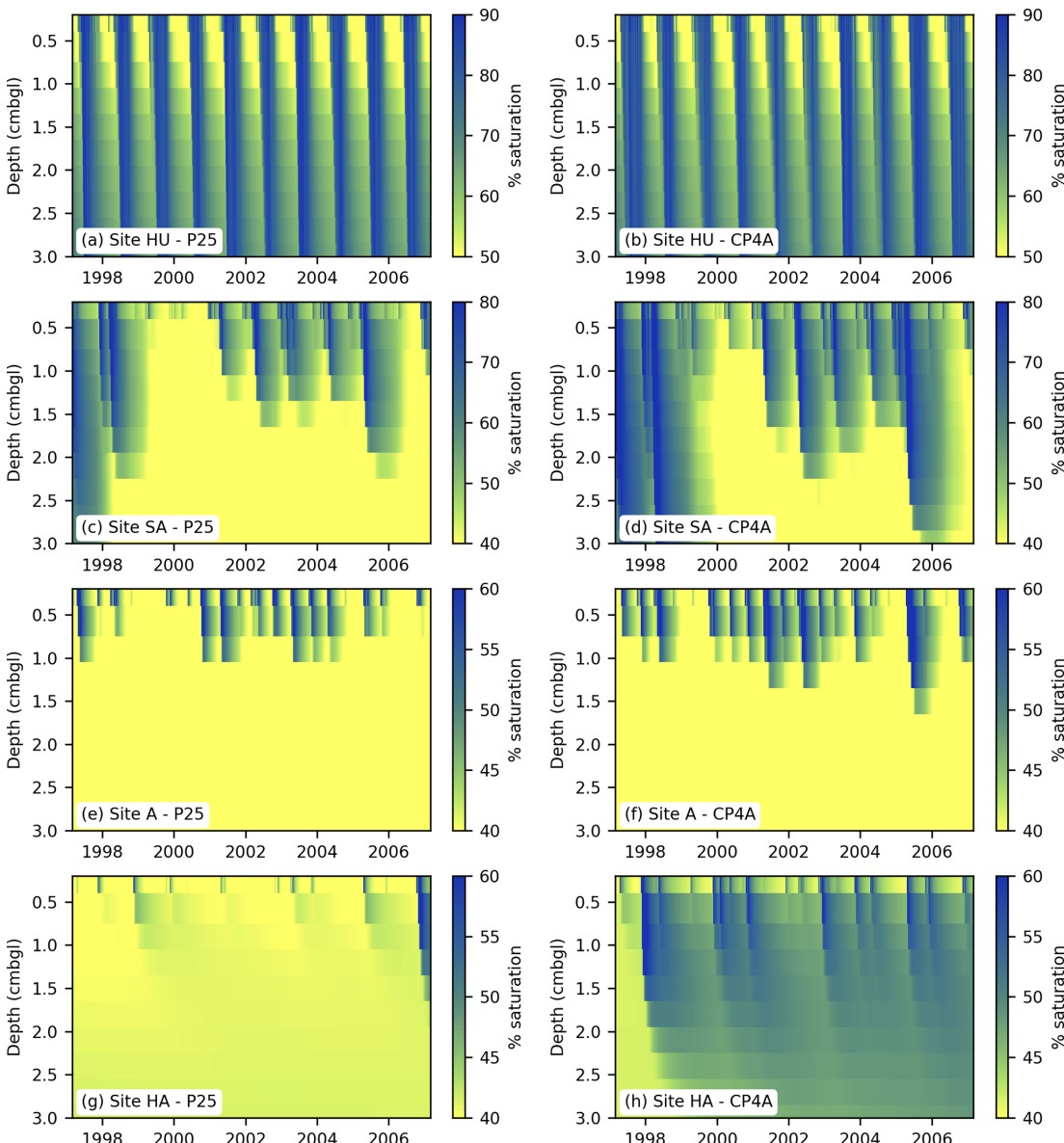

**Figure 8.** Modelled soil moisture profiles using P25 (left) and CP4A (right) rainfall and PET to drive Hydrus at our humid **(a–b)**, semi-arid **(c–d)**, arid **(e–f)**, and hyper-arid **(g–h)** sites across the HOA. It is worth noting that while the *x*-axis covers 1997–2007, the "historical" CP4A/P25 simulations are only designed to be statistically consistent with the observed climate rainfall between March 1997 and February 2007. They cannot replicate individual rainfall events observed over this period.

Even when forcing Hydrus with hPET (rather than model PET) (Appendix E – Table E5) evaporative losses are higher when using P25, the percentage of rainfall lost to evaporation in P25 (CP4A) simulations are 24 % (23 %), 81 % (66 %), 88 % (75 %), and 96 % (90 %) at Sites HU, SA, A, and HA respectively. In the CP4A runs, using hPET reduces transpiration totals, but they remain significantly higher than P25 runs in drylands (30 %–36 % higher in absolute terms) (Appendix E – Table E5). Transpiration totals are also obviously sensitive to the Feddes' parameters used, but it doesn't im-

pact the relative bias in totals between the CP4A and P25 Hydrus runs (Appendix 6 – Table E6).

Surface runoff is also greater when using CP4A rainfall despite lower annual totals (Table 6), in dryland locations, between 6 % and 10 % of rainfall is lost to runoff in the CP4 Hydrus runs versus 0.3 %–2 % with P25 rainfall. There are also large differences in bottom drainage from the soil profile, 286 vs. 23 mm at Site SA and 52 vs. 2 mm at Site HA (there was no drainage at Site A). Bottom drainage is lower in the hPET runs for both climate models, but the relative bias remains (Appendix E – Table E5). Drainage is also sen-

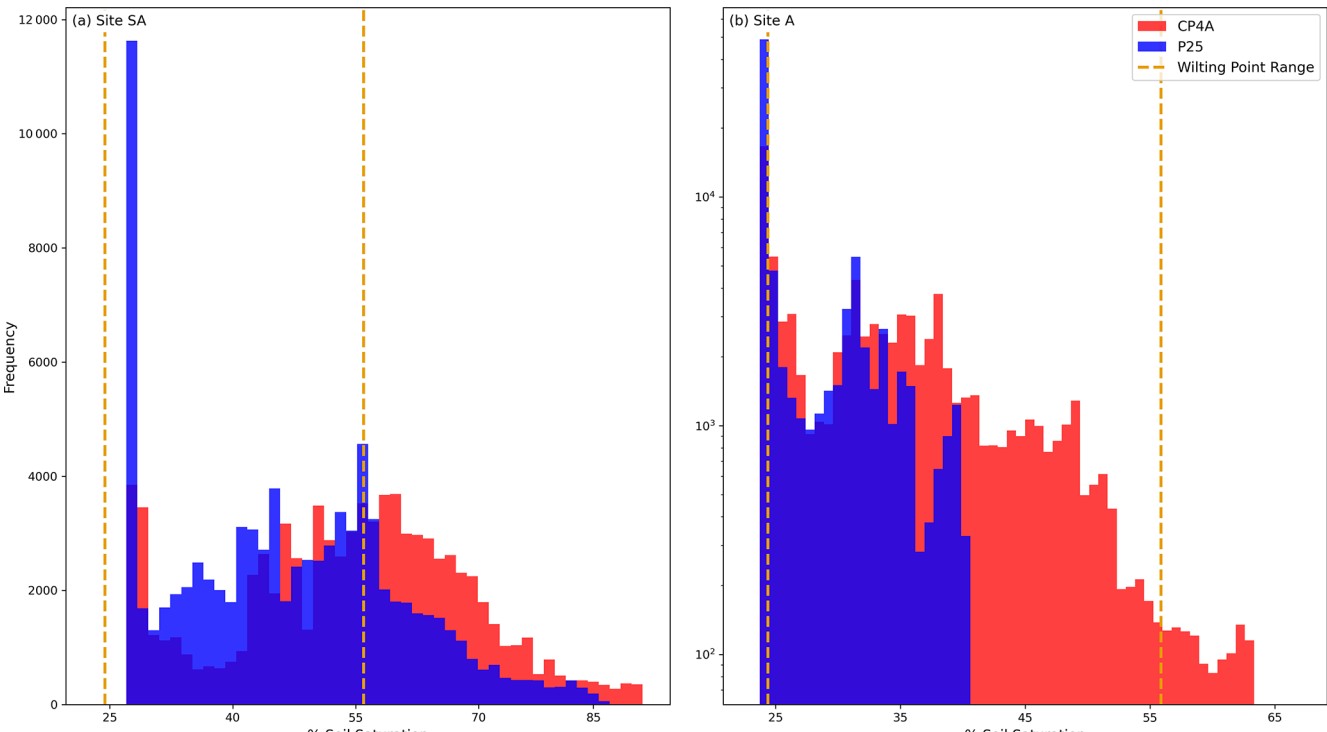

**Figure 9.** Soil Moisture Distributions with Wilting Points at Sites SA & A. Modelled distribution of soil moisture at 1.2 m b.g.l. at Site SA **(a)** and A **(b)** using P25 (blue) and CP4A (red) rainfall and PET. The dashed orange lines represent the wilting point range for Acacia shrubs, based on taking the upper and lower Feddes' parameters given in Appendix C – Table C1 (wilting point = P2H) (Sela et al., 2015). Plots computed using hourly Hydrus simulated soil moisture across the "historical" CP4A/P25 runs. These "historical" CP4A/P25 runs were designed to be statistically consistent with the observed climate between March 1997 and February 2007.

**Table 5.** Interquartile ranges of depth integrated soil moisture in the upper 1.2 m of the soil profile (top), and the entire 3 m profile (bottom) in our CP4A and P25 default (soil and Feddes' parameters) runs. All values refer to the % saturation and cover the CP4A/P25 "historical" runs.

| | Site SA | Site A | Site HA | Site HU |
|---|---|---|---|---|
| **1.2 m Below Ground Level** | | | | |
| CP4A | 49.7–63.6 | 37.7–46.5 | 47.2–51.3 | 53.0–78.9 |
| P25 | 43.4–58.2 | 31.4–41.0 | 39.7–41.1 | 51.6–81.3 |
| **3.0 m Below Ground Level** | | | | |
| CP4A | 42.7–59.0 | 29.7–33.7 | 47.8–49.5 | 60.6–77.7 |
| P25 | 35.4–45.1 | 27.0–30.8 | 40.5–41.1 | 59.6–79.4 |

sitive to the soil and Feddes' parameters used, but again in all cases CP4A runs simulate higher bottom drainage at our dryland sites (Appendix E – Tables E6 and E7). Whereas at our humid location (Site HU) P25 simulated 21 % higher bottom drainage, closely following the difference in the volume of cumulative rainfall delivered (15 %) (Table 6).

## 4   Discussion

In this paper we evaluated how climate model representation of convection influences rainfall characteristics and PET dynamics across the Horn of Africa (HOA), and what impact this has on the 1-D water balance at four sites along an aridity gradient in the HOA. In line with other studies, we find that climate models that explicitly resolve convection (CPMs) capture key dryland rainfall characteristics more effectively than those that parameterise the average effects of convection (when compared against satellite-derived rainfall observations). Both CP4A and P25 simulate comparable annual/seasonal totals, but they deliver rainfall in fundamentally different ways (light/frequent vs. heavy/infrequent), resulting in distinct hydrological outcomes at a point-based scale when their output is propagated through a vadose-zone hydrological model. This study also verifies that while dryland vadose-zone hydrology can be impacted by PET, ~~but~~ water partitioning is far more sensitive to rainfall characteristics (not rainfall totals or PET), and convective representation exerts a greater control on rainfall dynamics compared to PET.

Our modelling supports other work demonstrating that water partitioning in drylands is sensitive to the magnitude-

**https://doi.org/10.5194/**

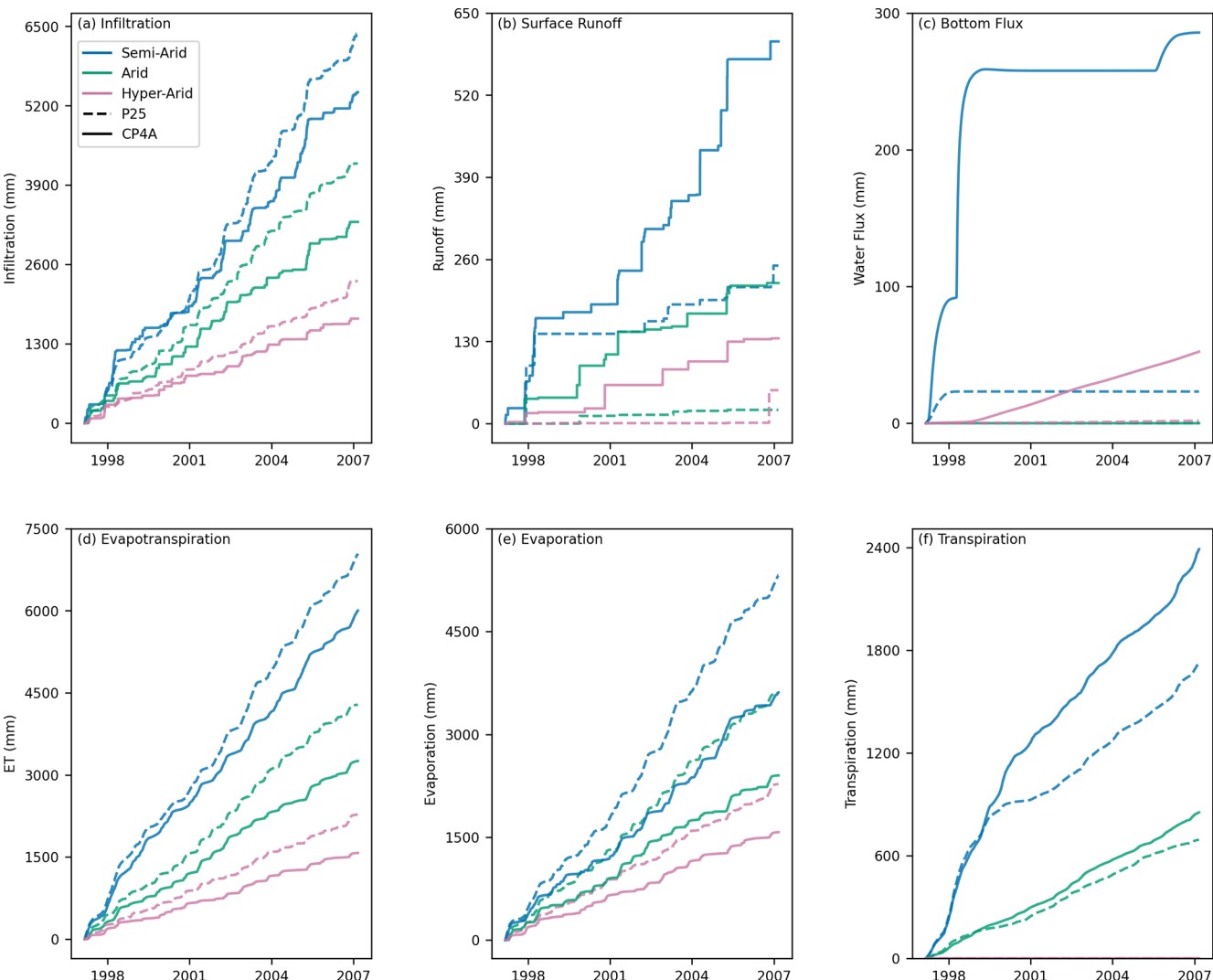

**Figure 10.** Cumulative Infiltration Runoff, Surface Runoff, Bottom Drainage, Evapotranspiration, Evaporation, and Transpiration. Modelled components of the water balance using CP4A (solid lines) and P25 (dashed lines) rainfall/PET as input for Hydrus 1-D at each of our dryland sites – Site SA (blue), Site A (green), and Site HA (pink). Plots show cumulative infiltration **(a)**, surface run-off **(b)**, bottom drainage **(c)**, evapotranspiration **(d)**, evaporation **(e)**, and transpiration **(f)**. So, for example, the solid blue line shows the CP4A semi-arid simulations, and the pink dashed line is the P25 hyper-arid simulation. To avoid using multiple axis, Site HU is not included, but the raw values are given in Table 6. It is worth noting that while the x-axis covers 1997–2007, the "historical" CP4A/P25 simulations are only designed to be statistically consistent with the observed climate rainfall between March 1997 and February 2007. They cannot replicate individual rainfall events observed over this period.

duration spectrum of rainfall (Taylor et al., 2013; ~~Arpuv~~ Apurv et al., 2017; Singer and Michaelides, 2017; Cuthbert et al., 2019; Kipkemoi et al., 2021; Adloff et al., 2022, Quichimbo et al., 2023), whereas in humid regions hydrological fluxes are more closely tied to seasonal rainfall totals. Our direct comparison between CPM and a traditional parameterised model (P25) provides another stark example of the importance of rainfall characteristics, where despite both models using the same global model configuration (Walters et al., 2017), simulating comparable annual totals, and seasonal cycles (Wainwright et al., 2021), hydrological out-

comes differ significantly in the dryland vadose-zone when climate model rainfall is propagated through a simple 1-D model. This highlights that while any hydrological study must carefully select the driving datasets, the importance of this choice increases in drylands as one must explicitly consider both rainfall totals and characteristics.

The ability of CPMs to better represent rainfall frequency, intensity, and the magnitude of extremes is well documented (Prein et al., 2015; Kouadio et al., ~~2018~~2020; Berthou et al., 2019; Luu et al., 2022), including in the context of the HOA (~~Bethou~~ Berthou et al., 2019; Kendon et al., 2019; Finney et

**Table 6.** Cumulative rainfall, potential evapotranspiration (PET), runoff, infiltration, evapotranspiration (ET), evaporation, transpiration, and drainage from the bottom of the soil profile. Values are given in mm and are the totals over the entire ten-year Hydrus simulations. All results are taken from the default (soil and Feddes' parameters) Hydrus runs forced with CP4A/P25 rainfall/PET. Those values given in brackets are the P25 Hydrus runs, non-brackets are CP4A. Values are the cumulative Hydrus fluxes recorded when driving Hydrus using the "historical" CP4A/P25 runs. These "historical" CP4A/P25 simulations were designed to be statistically consistent with the observed climate between March 1997 and February 2007. n/a = not applicable.

| Default Hydrus Run | Rainfall (mm) | PET (mm) | Runoff (mm) | Infiltration (mm) | ET (mm) | Evaporation (mm) | Transpiration (mm) | Drainage (mm) |
|---|---|---|---|---|---|---|---|---|
| Site HU | 17 333 (20 003) | 13 982 (12 894) | 0 (30) | 17 300 (19 863) | 10 474 (11 418) | 4030 (4693) | 6445 (6725) | 7154 (8638) |
| Site SA | 5952 (6669) | 15750 (16 230) | 605 (250) | 5423 (6398) | 6003 (7044) | 3611 (5320) | 2392 (1724) | 286 (23) |
| Site A | 3521 (4289) | 19 233 (21 817) | 223 (22) | 3297 (4254) | 3255 (4283) | 2402 (3598) | 853 (694) | 0 (0) |
| Site HA | 1849 (2394) | 17181 (19 714) | 135 (53) | 1713 (2328) | n/a | 1574 (2277) | n/a | 52 (2) |

al., 2019; 2020). This study demonstrates this improvement is most marked in dryland regions of the HOA (AI ≤ 0.5) compared to more humid (AI > 0.65) regions (Ethiopian Highlands). This is problematic, as despite their reduced skill in drylands, conventional climate models (those that parameterise the average effects of convection) are widely used to make future projections of dryland rainfall (Huang et al., 2017) and used as driving datasets for hydrological modelling (Crosbie et al., 2010; McKenna and Sala, 2017; Razack et al., 2019; Cook et al., 2022).

~~Our modelling demonstrates water partitioning in drylands is sensitive to the magnitude-duration spectrum of rainfall delivery (Taylor et al., 2013; Arpuv et al., 2017; Singer and Michaelides, 2017; Cuthbert et al., 2019; Kipkemoi et al., 2021; Adloff et al., 2022, Quichimbo et al., 2023). That two climate model products based on the same global model configuration (Walters et al., 2017), with comparable annual totals, and seasonal cycles (Wainwright et al., 2021) produce such differing hydrological outcomes when propagated through a simple 1-D model highlights the importance of carefully selecting driving datasets in~~ A key limitation with any one-dimensional modelling study is that lateral and non-local processes are explicitly excluded, however were such processes to be considered, it is unlikely to contradict our findings that using parameterised climate model output in dryland hydrological studies will lead to higher evaporative losses, lower soil moisture, transpiration, and potential groundwater recharge (bottom drainage) relative to CPMs. In fact, one could make the argument that differences would only become more pronounced were such analysis conducted at a basin/regional scale.

The low infiltration rates and high rainfall intensities typical of drylands partition rainfall into some combination of surface runoff and infiltration (Zhu et al., 2018; Aryal et al., 2020), where runoff is predominately generated via the infiltration-excess overland flow mechanism (Hortonian overland flow) (Horton, 1933). Whether runoff is generated is a function of storm characteristics and land surface properties, with only the most intense rainfall events able to gen-

erate enough surface runoff to form the ephemeral channels/pools that lead to large recharge events (Taylor et al., 2013; Schreiner-McGraw et al., 2019). If one was to use a basin/regional scale process-based dryland hydrological model that accounts for these processes (Quichimbo et al., ~~2021~~2021b), it is reasonable to assume that the more intense rainfall (Prein et al., 2015; Kouadio et al., ~~2018~~2020; Berthou et al., 2019; Kendon et al., 2019; Finney et al., 2019, 2020; Luu et al., 2022) and higher surface runoff (Folwell et al., 2022) when using CPMs as a driving dataset (relative to conventional parameterised models) would yield more realistic surface runoff patterns and greater non-local recharge. Furthermore, within such a basin/regional simulation, local infiltration in response to light rainfall is still more likely to be lost to evaporation (as this study has demonstrated), so were one to drive a regional dryland model with parameterised climate model output (compared to a CPM), there would be less surface runoff available for non-local recharge, and less localised recharge.

Hence, regardless of the spatial scale (point-based or regional-scale), it is critical that any dryland hydrological study uses rainfall data that correctly captures both rainfall totals and characteristics; discounting either risks misrepresenting societally relevant aspects of the hydrological cycle. For example, it is quite evident that any dataset that underestimates the tails of the rainfall distribution will result in lower estimates of flood risk (Ascott et al., 2023; Archer et al., 2024), but biases in rainfall characteristics could also have implications for crop health and groundwater recharge projections. Our results showing that using CP4A increases the volume of water penetrating deeper into the soil profile and higher transpiration rates which continue longer into the dry season (Folwell et al., 2022) suggests forcing crop models using parameterised climate model rainfall could result in an underestimation of crop yields and an overestimation of the risk of failure. Although equally, taking P25 as an example, lower rainfall intensities (reduced crop flooding) and shorter dry periods (lower water stress) could lead to overly optimistic projections of crop health relative to CP4A. Ei-

ther way, it is clear that for dryland regions of sub-Saharan Africa, where livelihoods are heavily dependent on subsistence agriculture and pasture (Davenport et al., 2017, 2018; Hoffmann et al., 2022), ensuring agricultural impact assessments use data that represents dryland rainfall characteristics is critical for providing more realistic projections of future change.

Driving hydrological models with datasets that correctly represent dryland rainfall characteristics could also be used to produce more realistic projections of future groundwater resources, the only source of perennial freshwater in drylands (MacDonald et al., 2012). Because as discussed above, groundwater recharge is highly sensitive to rainfall characteristics (Taylor et al., 2013; Batalha et al., 2018; Boas and Mallants, 2022) and does not linearly trend with higher seasonal rainfall totals. For example, in the context of the HOA, despite a decline in seasonal rainfall totals, groundwater storage is increasing, driven by a positive trend in extreme rainfall intensity (Adloff et al., 2022). In the HOA and drylands more broadly, future water resources will not simply follow changes in mean seasonal rainfall (even at point-scale where we exclude critical non-local processes, our results show enhanced drainage from the bottom of the soil profile when forcing Hydrus with CP4A despite lower seasonal rainfall), so it is critical any future assessments of groundwater resources utilise driving datasets that capture dryland rainfall characteristics.

Although while CPMs may provide valuable insights into projected changes in future rainfall characteristics (~~Bethou~~ Berthou et al., 2019; Kendon et al., 2019; Finney et al., 2019, 2020) they are not currently a panacea for the wider uncertainty that limits our understanding of future climate change impacts on water resources in the HOA. As while explicitly resolving convection can influence regional circulation patterns (Finney et al., 2020), CPMs still inherit the underlying uncertainties in the driving GCM, meaning it is unlikely they will resolve the failure of climate models to reproduce the observed drying trend in MAM rainfall over the last 30 years (named ~~as~~ the "East Africa Climate Paradox") (Lyon and Vigaud, 2017; Wainwright et al., 2019; Schwarzwald and Seager, 2024) and their inability to capture important modes of variability and wider large-scale processes (Schwarzwald et al., 2023). So, while robustly capturing rainfall characteristics is critical, the uncertainty and the computational costs associated with CPM simulations means utilising stochastic rainfall generators (Singer et al., 2018; Rios Gaona et al., 2024) and process-based scaling approaches consistent with CPM behaviour may be a better approach for producing a range of plausible time-evolving futures in the HOA (Klein et al., 2021). Taking such a "storyline" approach (Shepherd et al., 2018) could provide more valuable insights than simply forcing hydrological simulations using RCMs or GCMs that struggle to capture the mean climate state as well the dryland rainfall characteristics, as doing so risks producing misrepresentative projections of metrics such as soil mois-

ture, transpiration, and groundwater recharge, which could contribute to sub-optimal decision making around long-term land use or water supply policy.

## 5 Conclusions

In this study, we find that explicitly resolving convection improves the ability of climate models to capture dryland rainfall characteristics compared with models that parameterise the average effects of convection, CP4A dramatically reduces the systemic "drizzle" bias seen in P25, and better represents dry spell length, the magnitude of extremes, and the contribution of heavy rainfall events to seasonal totals. Despite using similar model physics and simulating comparable seasonal totals, the impact of convective representation on rainfall characteristics results in different hydrological outcomes when rainfall is propagated through a simple one-dimensional vadose-zone hydrological model. In dryland locations, using CP4A to drive Hydrus 1-D produces higher soil moisture, transpiration, and bottom drainage, despite simulating lower total rainfall and infiltration compared to P25. The "drizzle" bias in P25 confines infiltration to the upper layers of the soil profile, where it is quickly returned to the atmosphere via evaporation. Our results also show that while PET can influence vadose-zone hydrological outcomes, dryland hydrology is more sensitive to the impact of convective representation on rainfall characteristics. These findings suggest that any impact assessments of dryland hydrological resources must carefully consider the ~~spatiotemporal~~ spatio-temporal resolution and characteristics of rainfall datasets (or climate model rainfall), or they risk misrepresenting societally relevant aspects of the water cycle.

## Appendix A

Appendix A provides additional detail around the calculation PET as discussed in Sect. 2.2.

As discussed, is Sect. 2.2 dew point temperature is not directly outputted from either climate model, it was calculated using near-surface air temperature and relative humidity using the Clausius–Clapeyron approximation (Eq. A3) (Alduchov and Eskridge, 1996). Relative humidity is also not directly outputted from the models, so to calculate relative humidity (RH) we used the following equation (Eq. A1): **TS4**

$$\text{RH} = \frac{\omega}{\epsilon + \omega} \frac{\epsilon + \omega_s}{\omega_s} \tag{A1}$$

Where $\omega$ is the mixing ratio, $\omega_s$ is the saturation mixing ratio, and $\epsilon$ is the molecular weight ratio of vapor to dry air.

For the above equation $\omega$ is calculated using Eq. (A2), where $q$ is specific humidity:

$$\omega = \frac{q}{(1 - q)} \tag{A2}$$

To calculate dew point temperature (we Td) use the following equation (Eq. A3):

$$\text{Td} = \frac{b \cdot \gamma\,(T, \text{RH})}{a - \gamma\,(T, \text{RH})} \tag{A3}$$

Where $T$ is air temperature (in °C), RH is the relative humidity computed in Eq. (A1), $a$ refers to the empirical constant controlling the slope of the temperature–vapour pressure relationship ($a = 17.625$), and $b$ is temperature intercept in the saturation vapour pressure curve ($b = 243.04\,°C$). In the above equation $\gamma\,(T, \text{RH})$ is computed using Eq. (A4).

$$\gamma\,(T, \text{RH}) = \frac{a \cdot T}{b + T} + \ln\left(\frac{\text{RH}}{100}\right) \tag{A4}$$

Equations (A2)–(A6) **TS5** provides details the equation used to calculate the saturation vapour pressure ($e_s$), actual vapour pressure ($e_a$), slope of saturation vapour pressure ($\Delta$), net radiation ($R_n$), and the soil heat flux ($G$) used to compute PET in Eq. (1) in the main body of text.

For use in Eq. (1), $e_s$ and $e_a$ were calculated using the Tetens equation (Tetens, 1930) using hourly air temperature ($T_a$) and dew point temperature ($T_{\text{dew}}$) as detailed below (calculations are in °C after converting from K):

$$e_s = 0.6108 \exp\left(\frac{17.27 \times T_a}{T_a + 237.3}\right) \tag{A5}$$

$$e_a = 0.6108 \exp\left(\frac{17.27 \times T_{\text{dew}}}{T_{\text{dew}} + 237.3}\right) \tag{A6}$$

The Slope of saturation vapour pressure ($\Delta$) and the psychrometric constant ($\gamma$) were calculated as follows:

$$\Delta = \frac{4098 e_s}{(T_a + 237.3)^2} \tag{A7}$$

$$\gamma = \frac{C_p \times P}{\varepsilon \times \lambda} \tag{A8}$$

Where $P$ is atmospheric pressure, $C_p$ is the air's specific heat at constant pressure based on the ideal gas law with a value of $1.013 \times 10^{-3}\,\text{MJ}\,\text{kg}^{-1}\,°\text{C}^{-1}$, $\varepsilon$ is the ratio of the molecular weight of water vapor to that of dry air (0.622), and $\lambda$ is the latent heat of vaporization ($2.45\,\text{MJ}\,\text{kg}^{-1}$).

Net radiation ($R_n$) is estimated using net solar ($R_s$) and thermal radiation ($R_t$) as (all values in $\text{MJ}\,\text{m}^{-2}$):

$$R_n = R_s - R_t \tag{A9}$$

Finally soil heat flux ($G$) is estimated as:

$$G = \begin{cases} G_{\text{day}} = 0.1 \times R_n \\ G_{\text{night}} = 0.5 \times R_n \end{cases} \tag{A10}$$

Where the soil heat flux ($G$) is estimated to be 10 % of net radiation ($R_n$) during the day and 50 % during the night (as the night-time heat flux is negative). At each pixel we use net solar radiation to define day and nighttime periods. Following the method used to calculate hPET (Singer et al., 2021), nighttime PET values have not been set to zero.

It is important to note that while other studies have used CP4A and other convection-permitting models to explore evapotranspiration (Folwell et al., 2022; Halladay et al., 2023; Lee and Hohenegger, 2024), they used internal model evapotranspiration (ET), rather than externally calculating PET from model-derived atmospheric variables. While studying internal ET is useful for exploring land-atmosphere interactions – particularly feedbacks between soil moisture and precipitation – it reflects the model's internal assumptions about land surface properties, such as soil type and vegetation. In the case of CP4A, for example, ET is calculated using a uniform sandy soil across the entire domain, which limits its applicability in spatially heterogeneous hydrological studies. By contrast, externally calculating PET from model-derived atmospheric variables provides a consistent measure of atmospheric evaporative demand, independent of land surface parameterizations. For hydrological purposes where more detailed or locally calibrated soil and vegetation data is available, it is preferable to use the potential atmospheric evaporative demand (PET) computed using atmospheric climate model outputs.

## Appendix B

Appendix B provides additional figures and tables relevant to the analysis of Hydrus simulations (Sect. 3.3). Figure B1 shows the mean monthly CP4A and P25 rainfall at each of our four sites. Both models broadly capture the observed seasonality in the region (Wainwright et al., 2019), simulating the bimodal rainfall regime at our dryland sites (Sites SA, A, and HA) and the unimodal regime in the humid Ethiopian Highlands (Site HU). However, it is worth noting that P25 tends to simulate rainfall in every month (including July and August) at our dryland sites (CP4A simulates negligible rainfall) and simulates substantially higher rainfall totals during June and July at Site HU.

Figure B2 shows mean monthly CP4A and P25 PET at each of our four study sites. At all sites both models simulate comparable seasonal cycles and totals, although P25 consistently simulates higher PET at Sites A and HA. It is also worth noting that P25 simulates substantially higher PET during the months June–August, matching the higher JJAS PET simulated by P25 across the entire arid region of the HOA.

Figure B3 shows the raw time series of rainfall at each study site, it provides a clear demonstration of the tendency for CP4A to simulate heavier extreme rainfall events, as well as more frequent intense rainfall events at our dryland study sites (Sites SA, A, and HA). Whereas at our humid site in the Ethiopian Highlands, the differences are far more muted and at times it is P25 that is simulating heavier rainfall events.

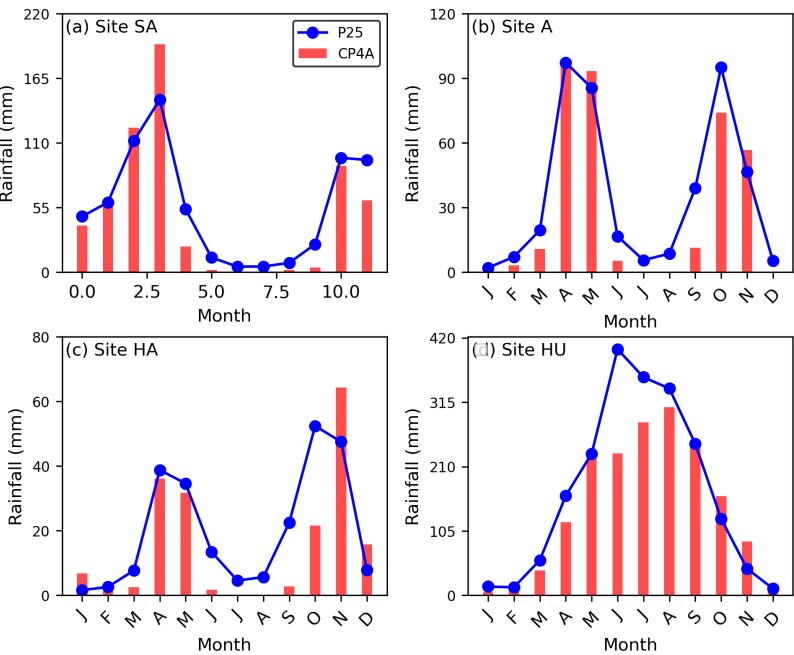

**Figure B1.** Mean monthly rainfall at each of our four hydrological study sites using the CP4A/P25 "historical" runs. These "historical" CP4A/P25 simulations are statistically consistent with the observed climate between March 1997 and February 2007. The simulated seasonal cycles show good agreement with the observed seasonality (Cocking et al., 2024). TS6

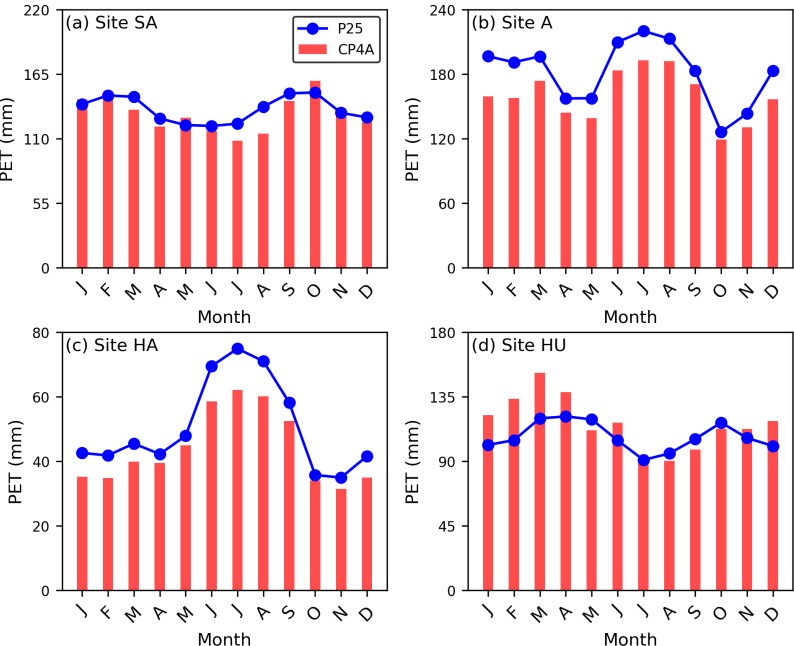

**Figure B2.** Mean monthly PET at each of our four hydrological study sites using the CP4A/P25 "historical" runs. These "historical" CP4A/P25 simulations were designed to be statistically consistent with the observed climate between March 1997 and February 2007. The simulated seasonal cycles of PET show good agreement with hPET (not shown).

Track changes document – Do not use for proofreading

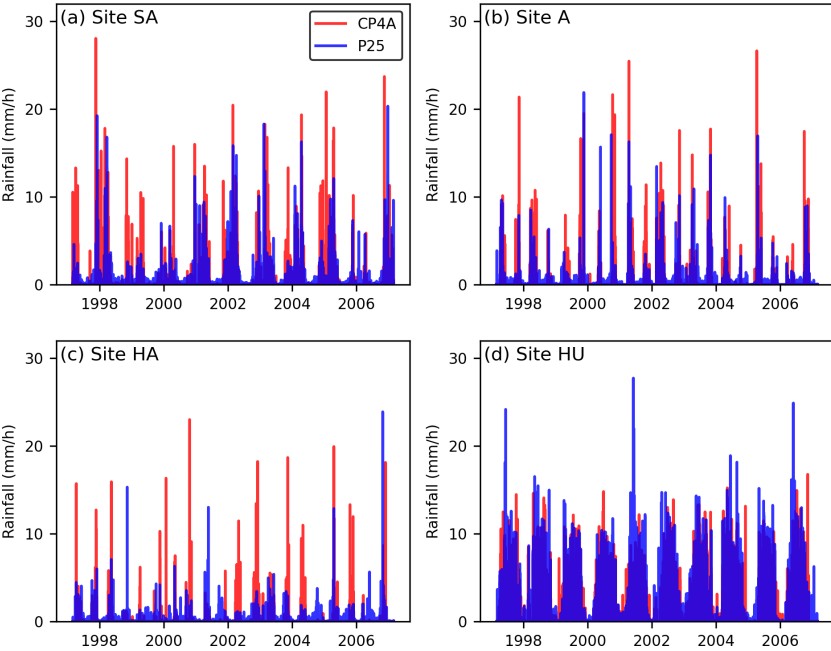

**Figure B3.** Raw time series of rainfall at each of our four hydrological study sites using the CP4A/P25 "historical" runs These "historical" CP4A/P25 simulations were designed to be statistically consistent with the observed climate between March 1997 and February 2007. They cannot capture individual observed rainfall events; hence we did not include the raw IMERG time series.

## Appendix C

Appendix C details the parameters that were altered between the different Hydrus simulations used at each site.

Table C1 shows the Feddes' parameters used to compute transpiration (root water uptake) within Hydrus (Feddes, 1978). Where P0 is pressure value at which roots start to extract water from the soil, POpt is the pressure head at which roots extract water at the maximum possible rate, P2H is the pressure head at which roots can no longer extract water at the maximum possible rate (assumes a potential transpiration rate of r2H), P2L is the same above but instead assumes a maximum possible transpiration rate of r2L, and P3 is the value of the pressure head at which roots can no uptake any water from the soil (wilting point). Vegetation type was taken from iSDA based on 2019 data (iSDA, 2024).

Hydrus has an internal database of Feddes' parameters for various crop types; the maize parameters were taken from this database and are based on Wesseling ~~et al. (1991~~and Brandyk (1985). However, shrubs are not in this database and there is very little published information on Feddes' parameters for shrubs. We were able to locate some thresholds for dryland shrubs: *Acacia Mearsii*, *Caragana korshinkii*, and *Sarcopoterium spinosum* (Xia and Shao, 2008, Sela et al., 2015, Watson, 2015). Sela et al. (2015) provided a range of Feddes' parameters used in their calibration process to correctly quantify transpiration rates of *Caragana korshinkii* in Hydrus 1-D, settling on an optimal parameter set (Table C1). However, we decided to use the upper range (here referred to as def) of the parameters given by Sela et al. (2015) as they better matched estimates given by Xia and Shoa (2008) and Watson (2015), particularly the wilting point. However, to ensure results any biases seen between CP4A and P25 Hydrus runs were consistent regardless of the Feddes' parameters chosen, we also used the optimal and lower values provided by Sela et al. (2015) at sites SA and A (where shrubs are the dominant land cover). However, unless stated otherwise all results reported in the main body of text were computed using the upper (def) Feddes' values – given in bold in Table C1.

At each site we used three different soil hydraulic parameters (Table C2): lowK, def, and highK. Where def (given in bold in Table C2) refers to the default soil parameters that are used for all simulations discussed in the main text, lowK refers to low hydraulic conductivity simulations, and highK is high hydraulic conductivity simulations. For example, at Site A hydraulic conductivity ($K_s$) ranges from $3.5\,\mathrm{mm\,h^{-1}}$ (lowK) to $21.8\,\mathrm{mm\,h^{-1}}$ (highK).

These soil parameters were estimated using Genuchten-Mualem (Van Genuchten, 1980) equations based on soil texture values taken from the Innovate Solutions for Decision Agriculture (iSDA) soil database (Hengl et al., 2021). Where Qr refers to residual soil water content, Qs is the saturated water content, Alpha is Parameter *a* in the soil water retention function [$\mathrm{L^{-1}}$], *n* is parameter *n* in the soil water retention function, $K_s$ is the saturated hydraulic conductivity [$\mathrm{L\,T^{-1}}$], and *I* is the tortuosity parameter in the conductivity function [–]. iSDA provides a lower and upper bound of sand, silt, and clay percentages (Hengl et al., 2021), our default ("def") parameter set was estimated using the mid-point of these lower and upper bounds for each soil texture.

While to create our low ("lowK") and high hydraulic conductivity ("highK") soil parameters, we used the lower and upper bound of the sand percentage respectively, and then proportionally adjusted the silt and clay percentage to ensure values equalled 100. So, for example, our "highK" scenarios have higher saturated hydraulic conductivity ($K_s$) as the relative percentage of sand is higher than in our "def" and "lowK" scenarios. We used the same labelling for each site as they follow the same methodology and are designed to be comparable across sites, although soil parameters will differ between sites based on the relative proportion of sand, silt, and clay at each site.

**Table C1.** Feddes'~ parameters controlling root water uptake (transpiration) for shrubs and maize used in Hydrus 1-D simulations. Values were for shrubs were taken from Sela et al. (2015) and maize from Wesseling ~~et al.~~ and Brandyk (~~1991~~1985). Bold: The upper parameter set is the default parameter set used in all Hydrus simulations unless explicitly stated otherwise. **TS7**

| Feddes' Parameter | P0 (mm) | POpt (mm) | P2H (mm) | P2L (mm) | P3 (mm) | r2H (mm h⁻¹) | r2L (mm h⁻¹) |
|---|---|---|---|---|---|---|---|
| **Shrubs Upper (def)** | **−150** | **−300** | **−5000** | **−15 000** | **−240 000** | **0.208** | **0.042** |
| Shrubs Mid | −58 | −224 | −326 | −6700 | −15 570 | 0.208 | 0.042 |
| Shrubs Lower | 0 | −150 | −300 | −5000 | −15 000 | 0.208 | 0.042 |
| Maize | −150 | −300 | −3250 | −6000 | −80 000 | 0.208 | 0.042 |

**Table C2.** Soil Hydraulic parameters used in all Hydrus simulation at our humid (HU), semi-arid (SA), arid (A), and hyper-arid (HA) sites across the HOA. Bold: The def parameters given in bold are the default soil hydraulic parameters used in all Hydrus simulations unless explicitly stated otherwise. TS8

| Site HU (Humid) | | | | | | | |
|---|---|---|---|---|---|---|---|
| Scenario | Depth | Qr | Qs | Alpha | $n$ | Ks $(\text{mm h}^{-1})$ | $I$ |
| LowK | 0–20 cm | 0.103 | 0.545 | 0.002 | 1.339 | 18.342 | 0.500 |
| LowK | 20–500 cm | 0.105 | 0.543 | 0.002 | 1.315 | 15.650 | 0.500 |
| **Def** | **0–20 cm** | **0.096** | **0.525** | **0.002** | **1.384** | **19.117** | **0.500** |
| **Def** | **20–500 cm** | **0.100** | **0.529** | **0.002** | **1.349** | **17.833** | **0.500** |
| HighK | 0–20 cm | 0.084 | 0.510 | 0.002 | 1.405 | 22.588 | 0.500 |
| HighK | 20–500 cm | 0.090 | 0.512 | 0.002 | 1.384 | 19.996 | 0.500 |

| Site SA (Semi-Arid) | | | | | | | |
|---|---|---|---|---|---|---|---|
| Scenario | Depth | Qr | Qs | Alpha | $n$ | Ks $(\text{mm h}^{-1})$ | $I$ |
| LowK | 0–20 cm | 0.076 | 0.416 | 0.001 | 1.405 | 3.221 | 0.500 |
| LowK | 20–500 cm | 0.081 | 0.423 | 0.002 | 1.351 | 3.204 | 0.500 |
| **Def** | **0–20 cm** | **0.068** | **0.409** | **0.002** | **1.400** | **5.658** | **0.500** |
| **Def** | **20–500 cm** | **0.076** | **0.418** | **0.002** | **1.354** | **4.175** | **0.500** |
| HighK | 0–20 cm | 0.066 | 0.408 | 0.002 | 1.401 | 6.663 | 0.500 |
| HighK | 20–500 cm | 0.072 | 0.417 | 0.002 | 1.367 | 6.100 | 0.500 |

| Site A (Arid) | | | | | | | |
|---|---|---|---|---|---|---|---|
| Scenario | Depth | Qr | Qs | Alpha | $n$ | Ks $(\text{mm h}^{-1})$ | $I$ |
| LowK | 0–20 cm | 0.081 | 0.431 | 0.001 | 1.426 | 3.542 | 0.500 |
| LowK | 20–500 cm | 0.085 | 0.436 | 0.001 | 1.404 | 3.338 | 0.500 |
| **Def** | **0–20 cm** | **0.067** | **0.418** | **0.002** | **1.415** | **7.558** | **0.500** |
| **Def** | **20–500 cm** | **0.069** | **0.419** | **0.002** | **1.404** | **6.363** | **0.500** |
| HighK | 0–20 cm | 0.055 | 0.414 | 0.003 | 1.475 | 21.817 | 0.500 |
| HighK | 20–500 cm | 0.058 | 0.413 | 0.002 | 1.439 | 16.004 | 0.500 |

| Site HA (Hyper-Arid) | | | | | | | |
|---|---|---|---|---|---|---|---|
| Scenario | Depth | Qr | Qs | Alpha | $n$ | Ks $(\text{mm h}^{-1})$ | $I$ |
| LowK | 0–20 cm | 0.070 | 0.417 | 0.002 | 1.400 | 5.779 | 0.500 |
| LowK | 20–500 cm | 0.069 | 0.417 | 0.002 | 1.394 | 6.783 | 0.500 |
| **Def** | **0–20 cm** | **0.067** | **0.417** | **0.002** | **1.406** | **7.633** | **0.500** |
| **Def** | **20–500 cm** | **0.069** | **0.417** | **0.002** | **1.394** | **6.783** | **0.500** |
| HighK | 0–20 cm | 0.056 | 0.413 | 0.003 | 1.480 | 21.646 | 0.500 |
| HighK | 20–500 cm | 0.056 | 0.411 | 0.003 | 1.466 | 19.329 | 0.500 |

## Appendix D

Appendix D provides additional figures and materials on the analyses of CP4A/P25 rainfall and PET simulations (Sect. 3.1 and 3.2). Figure D1 is a replication of Fig. 3 in the main body of text, plotting the distribution of rainfall intensities for all rainfall hours (based on a threshold of $0.1\,\mathrm{mm\,h^{-1}}$) for humid (AI $\geq 0.65$) and dryland regions (AI $< 0.50$) in each season across the HOA. The plots highlight the "drizzle" effect simulated by P25 (shown by the large frequency peaks in rain hours between $10^{-1}$ and $10^{1}\,\mathrm{mm\,h^{-1}}$) is consistent across seasons, with P25 overestimating the number of wet hours in both wet (MAM & OND) and dry (JF & JJAS) seasons.

To understand the drivers of PET differences between CP4A and P25 we conducted a multiple-linear regression between PET and the seven atmospheric variables described in Sect. 2.2. Table D1 shows the results of this multi-linear regression between the daily PET climatology and the daily climatology of the seven atmospheric variables used to compute PET in humid and arid regions of the Horn of Africa (discussed in Sect. 3.2).

Table D1 suggests meridional wind speed is relatively important in arid regions, Fig. D2 shows the daily meridional wind speed climatology in arid regions of the Horn of Africa. It appears that the positive significant relationship between meridional wind speed and PET (Table D1) is the driving factor behind higher PET in P25 during the months of June to September (see Fig. 3g), as Fig. D2 closely matches Fig. 3g.

**Table D1.** Results of the multi-linear regression analysis on the drivers of PET in CP4A and P25 in humid (top) (AI $\geq 0.65$) and arid (AI $< 0.2$) regions of the HOA. Bold values refer to CP4A, while non-bold values refer to P25. Values are based on the daily climatology of PET simulated across the "historical" CP4A/P25 simulations. These "historical" CP4A/P25 simulations were designed to be statistically consistent with the observed climate between March 1997 and February 2007.

| Humid Regions | | | | | |
|---|---|---|---|---|---|
| Variable | Coefficient | Std Error | $P > \lvert t \rvert$ | 0.025 CI | 0.975 CI |
| Temp | 0.127 (**0.438**) | 0.028 (**0.046**) | 0 (**0.000**) | 0.072 (**0.349**) | 0.182 (**0.529**) |
| Dew Point | 0.152 (**−0.129**) | 0.021 (**0.031**) | 0 (**0.000**) | 0.112 (**−0.191**) | 0.193 (**−0.068**) |
| Surface Pressure | 0.0001 (**−0.002**) | 0.000 (**0.000**) | 0.58 (**0.000**) | 0.000 (**−0.002**) | 0.000 (**−0.001**) |
| Short Wave Radiation | $7.2 \times 10^{-7}$ (**$7.9 \times 10^{-7}$**) | $1.6 \times 10^{-7}$ (**$2.2 \times 10^{-7}$**) | 0.000 (**0.001**) | $4.1 \times 10^{-7}$ (**$3.4 \times 10^{-7}$**) | $1.0 \times 10^{-6}$ (**$1.2 \times 10^{-6}$**) |
| Long Wave Radiation | $-7.9 \times 10^{-6}$ (**$2.6 \times 10^{-6}$**) | $1.4 \times 10^{-6}$ (**$1.8 \times 10^{-6}$**) | 0.000 (**0.149**) | $-1.1 \times 10^{-5}$ (**$-9.5 \times 10^{-7}$**) | $-5.1 \times 10^{-6}$ (**$6.2 \times 10^{-6}$**) |
| Meridional Wind | 0.041 (**−0.108**) | 0.024 (**0.034**) | 0.081 (**0.002**) | -0.005 (**−0.174**) | 0.088 (**−0.041**) |
| Zonal Wind | −0.181 (**−0.178**) | 0.027 (**0.039**) | 0 (**0.000**) | −0.235 (**−0.255**) | −0.127 (**−0.100**) |
| Arid Regions | | | | | |
| Variable | Coefficient | Std Error | $P > \lvert t \rvert$ | 0.025 CI | 0.975 CI |
| Temp | 0.285 (**0.232**) | 0.033 (**0.027**) | 0.000 (**0.000**) | 0.220 (**0.179**) | 0.349 (**0.285**) |
| Dew Point | −0.364 (**−0.391**) | 0.026 (**0.018**) | 0.000 (**0.000**) | −0.401 (**−0.426**) | −0.327 (**−0.356**) |
| Surface Pressure | 0.0009 (**−0.001**) | 0.000 (**0.000**) | 0.58 (**0.000**) | −0.001 (**−0.001**) | −0.001 (**−0.000**) |
| Short Wave Radiation | $2.0 \times 10^{-06}$ (**$7.0 \times 10^{-6}$**) | $2.1 \times 10^{-07}$ (**$4.5 \times 10^{-7}$**) | 0.000 (**0.001**) | $1.6 \times 10^{-6}$ (**$6.1 \times 10^{-6}$**) | $2.4 \times 10^{-6}$ (**$7.8 \times 10^{-6}$**) |
| Long Wave Radiation | $9.6 \times 10^{-6}$ (**$1.2 \times 10^{-5}$**) | $1.4 \times 10^{-6}$ (**$1.2 \times 10^{-6}$**) | 0.000 (**0.149**) | $6.8 \times 10^{-6}$ (**$1.0 \times 10^{-5}$**) | $1.2 \times 10^{-5}$ (**$1.5 \times 10^{-5}$**) |
| Meridional Wind | 0.178 (**0.103**) | 0.025 (**0.019**) | 0.000 (**0.002**) | 0.130 (**0.066**) | 0.226 (**0.139**) |
| Zonal Wind | −0.095 (**0.016**) | 0.026 (**0.019**) | 0.000 (**0.000**) | −0.146 (**−0.022**) | −0.044 (**0.053**) |

https://doi.org/10.5194/

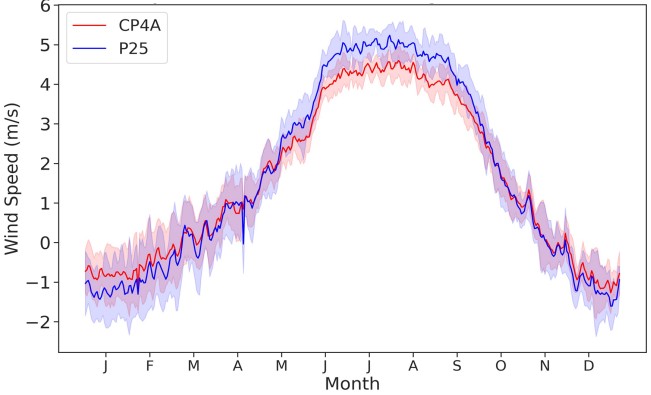

**Figure D1.** Rainfall KDE Plots. Kernel density estimate (kde) plots of CP4A, P25, and IMERG hourly rainfall in humid (AI $\geq$ 0.65) and dryland (AI < 0.5) regions of the Horn of Africa for JF **(a–b)**, MAM **(c–d)**, JJAS **(e–f)**, and OND **(g–h)**. Plots exclude dry hours by dropping any hours that receive < 0.1 mm h$^{-1}$ of rainfall. Plots cover the "historical" CP4A/P25 simulations. These "historical" CP4A/P25 simulations were designed to be statistically consistent with the observed climate between March 1997 and February 2007.

**Figure D2.** Daily climatology of meridional wind speed in arid (AI < 0.2) regions of the HOA. Plot covers the CP4A/P25 "historical" simulations, which were designed to be statistically consistent with the observed climate between March 1997 and February 2007.

## Appendix E

The following tables will provide additional detail on the soil moisture results for each site, building and complimenting what was discussed in the main body of text.

Table E1 shows that differences in soil moisture distributions are more pronounced (based on the Kolmogorov-Smirnov Test statistic) at the dryland sites versus Site HU. In all cases there are statistically significant differences between the CP4A and P25 runs ($p < 0.05$).

Table E2 shows that forcing Hydrus with CP4A rainfall also yields higher soil moisture values regardless of the Feddes' parameters (Table C1) used to estimate the root water uptake (transpiration) of dryland shrubs.

Table E3 shows that forcing Hydrus with CP4A rainfall yields higher soil moisture values regardless of the soil hydraulic parameters used (Table C2).

Running Hydrus with CP4A rainfall and hPET PET data (rather than CP4A PET) also demonstrates that differences in soil moisture between the CP4A and P25 Hydrus runs are primarily driven by rainfall (Table E4). As while the KS test statistic is lower (suggesting smaller differences in soil moisture distributions) compared to simulations driven by CP4A rainfall and PET (Table E1), there are still statistically significant differences between CP4A and P25 at all sites ($p < 0.05$) and at all depths. The tendency for the KS statistic to be higher in drylands also remains (Table E4).

The following tables provide additional detail around the cumulative water balance results discussed in Sect. 3.3.

Table E5 shows the same results as Table 6 in the main body of text, but for the Hydrus simulations using CP4A rainfall and hPET PET (rather than CP4A PET). It shows that while the raw values differ, the pattern where P25 loses more infiltration to evaporation and CP4A simulates higher transpiration and bottom drainage remains.

Table E6 shows the cumulative water balance values at sites SA and A under our different Feddes' parameters (Table C1), it shows that whether we use the most or least water resilient set of parameters, the results remain consistent (P25 – higher evaporative losses, CP4A – higher transpiration, runoff, and bottom drainage).

Table E7 shows the same cumulative water balance components at each site under our low hydraulic conductivity (LK), high hydraulic conductivity (HK) scenarios. In line with the soil moisture results, altering soil hydraulic parameters does not alter the biases seen between CP4A and P25, where P25 loses more infiltration to evaporation and CP4A simulates higher transpiration, surface runoff, and bottom drainage.

**Table E1.** Kolmogorov–Smirnov (KS) test statistics comparing soil moisture distributions at our hydrological four sites across different depths. Higher KS values indicate greater differences between distributions. All results are statistically significant ($p < 0.05$). Site HA consistently shows the largest differences, while Site HU exhibits the smallest KS values, suggesting the least divergence. All results refer to Hydrus simulations driven by CP4A rainfall and PET using default soil and Feddes' parameters. Values are based on hourly Hydrus simulated soil moisture using the "historical" CP4A/P25 runs. These "historical" CP4A/P25 runs were designed to be statistically consistent with the observed climate between March 1997 and February 2007.

| Depth (m below ground level) | Site SA | Site A | Site HA | Site HU |
|---|---|---|---|---|
| 0.2 m b.g.l. | 0.16 | 0.14 | 0.74 | 0.07 |
| 0.6 m b.g.l. | 0.22 | 0.27 | 0.89 | 0.07 |
| 0.9 m b.g.l. | 0.22 | 0.44 | 0.93 | 0.08 |
| 1.2 m b.g.l. | 0.23 | 0.40 | 0.94 | 0.08 |
| 1.5 m b.g.l. | 0.35 | 0.49 | 0.98 | 0.08 |
| 1.8 m b.g.l. | 0.46 | 0.15 | 0.99 | 0.09 |
| 2.1 m b.g.l. | 0.41 | 0.00 | 1.00 | 0.09 |
| 2.4 m b.g.l. | 0.49 | 0.00 | 1.00 | 0.09 |
| 2.7 m b.g.l. | 0.38 | 0.00 | 1.00 | 0.09 |
| 3.0 m b.g.l. | 0.38 | 0.00 | 0.85 | 0.08 |

**Table E2.** Relative percentage difference in median depth integrated soil moisture between CP4A and P25 Hydrus runs. Default/Upper (given in bold) refers to the default Feddes' parameters that are used for all simulations discussed in the main text, low refers to simulations using the low (least water resilient) Feddes' parameters, and mid refers to simulations using the mid-range Feddes' parameter set. The low, mid, and upper Feddes' parameters are taken from Sela et al. (2015). See Table C1 for more details on each simulation. For all values reported, it is a positive percentage difference between CP4A and P25 soil moisture. Eg CP4A simulates higher soil moisture in all cases. Values are based on hourly Hydrus simulated soil moisture across the "historical" CP4A/P25 runs. These "historical" CP4A/P25 runs were designed to be statistically consistent with the observed climate between March 1997 and February 2007.

| Hydrus Run (depth meters below ground level) | Site SA (Semi-Arid) | Site A (Arid) |
|---|---|---|
| Low (least water resilient) Feddes' Parameters (1.2 m b.g.l.) | 6.2 | 3.6 |
| Low (least water resilient) Feddes' Parameters (3.0 m b.g.l.) | 10.7 | 2.4 |
| Mid Feddes' Parameters (1.2 m b.g.l.) | 6.4 | 8.5 |
| Mid Feddes' Parameters (3.0 m b.g.l.) | 10.9 | 4.7 |
| **Default/Upper (most water resilient) Feddes' Parameters (1.2 m b.g.l.)** | **10.3** | **15.5** |
| **Default/Upper (most water resilient) Feddes' Parameters (3.0 m b.g.l.)** | **20.7** | **9.1** |

**Table E3.** Relative percentage difference in median depth integrated soil moisture between CP4A and P25 Hydrus runs. Default (given in bold) refers to the default soil parameters that are used for all simulations discussed in the main body of text, lowK refers to low hydraulic conductivity simulations, and highK is high hydraulic conductivity simulations. See Table C2 for more details on each simulation. For all values reported, it is a positive percentage difference between CP4A and P25 soil moisture. E.g. CP4A simulates higher soil moisture in all cases. Values are based on hourly Hydrus simulated soil moisture across the "historical" CP4A/P25 runs. These "historical" CP4A/P25 runs were designed to be statistically consistent with the observed climate between March 1997 and February 2007.

| Hydrus Run (depth – meters below ground level) | Site SA | Site A | Site HA | Site HU |
|---|---|---|---|---|
| Low Hydraulic Conductivity (1.2 m b.g.l.) | 10.6 | 10.5 | 17.0 | 5.2 |
| Low Hydraulic Conductivity (3.0 m b.g.l.) | 14.4 | 5.1 | 14.7 | 2.5 |
| **Default (1.2 m b.g.l.)** | **10.3** | **15.5** | **21.4** | **2.5** |
| **Default (3.0 m b.g.l.)** | **20.7** | **9.1** | **19.4** | **5.2** |
| High Hydraulic Conductivity (1.2 m b.g.l.) | 10.4 | 32.0 | 22.3 | 5.7 |
| High Hydraulic Conductivity (3.0 m b.g.l.) | 22.6 | 21.9 | 25.5 | 2.6 |

**Table E4.** Kolmogorov–Smirnov (KS) test statistics comparing soil moisture distributions at our hydrological four sites across different depths. Higher KS values indicate greater differences between distributions. All results are statistically significant ($p < 0.05$). Compared to Hydrus simulations driven by CP4A rainfall and PET, those driven by CP4A rainfall and hPET exhibit marginally lower KS values. However, all results remain statistically significant ($p < 0.05$), the KS test statistic at Site HU still exhibit the smallest values, suggesting the least divergence. Values are based on hourly Hydrus simulated soil moisture across the "historical" CP4A/P25 runs (rainfall only). These "historical" CP4A/P25 runs were designed to be statistically consistent with the observed climate between March 1997 and February 2007.

| CP4A + hPET Hydrus Runs | Depth | Site SA | Site A | Site HA | Site HU |
|---|---|---|---|---|---|
| | 0.2 m b.g.l. | 0.12 | 0.13 | 0.67 | 0.09 |
| | 0.6 m b.g.l. | 0.16 | 0.28 | 0.84 | 0.10 |
| | 0.9 m b.g.l. | 0.14 | 0.37 | 0.86 | 0.10 |
| | 1.2 m b.g.l. | 0.19 | 0.29 | 0.86 | 0.10 |
| | 1.5 m b.g.l. | 0.30 | 0.19 | 0.85 | 0.10 |
| | 1.8 m b.g.l. | 0.33 | 0.12 | 0.84 | 0.10 |
| | 2.1 m b.g.l. | 0.35 | 0.00 | 0.85 | 0.10 |
| | 2.4 m b.g.l. | 0.27 | 0.00 | 0.83 | 0.10 |
| | 2.7 m b.g.l. | 0.34 | 0.00 | 0.80 | 0.10 |
| | 3.0 m b.g.l. | 0.20 | 0.00 | 0.79 | 0.10 |

**Table E5.** Cumulative rainfall, potential evapotranspiration (PET), runoff, infiltration, evapotranspiration (ET), evaporation, transpiration, and drainage from the bottom of the soil profile. All values are given in mm and are the totals over the entire ten-year Hydrus simulations. All results are taken from the default (soil and Feddes' parameters) Hydrus runs forced with CP4A/P25 rainfall and hPET, rather than climate model PET. Those values given in brackets are the P25 rainfall Hydrus runs, all others are forced using CP4A rainfall. Values are the cumulative Hydrus fluxes recorded using the "historical" CP4A/P25 runs (rainfall only). These "historical" CP4A/P25 simulations were designed to be statistically consistent with the observed climate between March 1997 and February 2007. n/a = not applicable.

| CP4A/hPET Hydrus Run | Rainfall (mm) | PET (mm) | Runoff (mm) | Infiltration (mm) | ET (mm) | Evaporation (mm) | Transpiration (mm) | Drainage (mm) |
|---|---|---|---|---|---|---|---|---|
| Site SA | 5952 (6669) | 15 243 | 566 (246) | 5450 (6405) | 6052 (7007) | 3936 (5382) | 2117 (1625) | 162 (22) |
| Site A | 3521 (4289) | 20 223 | 158 (10) | 3330 (4297) | 2637 (3789) | 2637 (3789) | 693 (509) | 0 (0) |
| Site HA | 1849 (2394) | 18 972 | 94 (51) | 1750 (2325) | n/a | 1665 (2291) | n/a | 36 (1) |
| Site HU | 17 333 (20 003) | 13 086 | 0 (29) | 17 171 (19810) | 10 926 (11 237) | 4188 (4620) | 6737 (6617) | 6601 (8634) |

**Table E6.** Cumulative rainfall, potential evapotranspiration (PET), runoff, infiltration, evapotranspiration (ET), evaporation, transpiration, and drainage from the bottom of the soil profile. All values are given in mm and are the totals over the entire ten-year Hydrus simulations. The lower Feddes' refers to simulations using the low (least water resilient) Feddes' parameters, mid refers to simulations using the mid-range Feddes' parameter set, and the Def run is our default run which uses the most water resilient parameter set (see Table C1). Def is the run reported in the main body of text. The low, mid, and upper (def) Feddes' parameters are taken from Sela et al. (2015). See Appendix C for more details on each simulation. Those values given in brackets are the P25 Hydrus runs, all others are forced by CP4A. Values are the cumulative Hydrus fluxes recorded using the "historical" CP4A/P25 runs. These "historical" CP4A/P25 simulations were designed to be statistically consistent with the observed climate between March 1997 and February 2007. Bold: The def parameters given in bold are the default Hydrus runs are refer to the results reported in the main body of text. TS9

| | Runoff | Infiltration | ET | Evaporation | Transpiration | Drainage |
|---|---|---|---|---|---|---|
| Lower Feddes' Run | | | | | | |
| Site SA | 611 (254) | 5418 (6394) | 5807 (6859) | 3707 (5433) | 2099 (1426) | 394 (35) |
| Site A | 176 (23) | 3344 (4253) | 3314 (4271) | 2657 (3734) | 657 (537) | 4 (3) |
| Mid Feddes' Run | | | | | | |
| Site SA | 610 (253) | 5418 (6394) | 5819 (6859) | 3697 (5424) | 2121 (1435) | 382 (31) |
| Site A | 226 (11) | 3292 (4261) | 3266 (4286) | 2514 (3813) | 752 (474) | 4 (3) |
| Def Run Type | | | | | | |
| **Site SA** | **605 (250)** | **5423 (6398)** | **6003 (7044)** | **3611 (5320)** | **2392 (1724)** | **286 (23)** |
| **Site A** | **223 (22)** | **3297 (4254)** | **3255 (4283)** | **2402 (3598)** | **853 (694)** | **0 (0)** |

**Table E7.** Cumulative rainfall, potential evapotranspiration (PET), runoff, infiltration, evapotranspiration (ET), evaporation, transpiration, and drainage from the bottom of the soil profile. All values are given in mm and are the totals over the entire ten-year Hydrus simulations. lowK refers to low hydraulic conductivity simulations, and highK is high hydraulic conductivity simulations. Those values given in brackets are the P25 Hydrus runs, all others are forced by CP4A. Values are the cumulative Hydrus fluxes recorded using the "historical" CP4A/P25 runs. These "historical" CP4A/P25 simulations were designed to be statistically consistent with the observed climate between March 1997 and February 2007. n/a = not applicable.

| | Runoff (mm) | Infiltration (mm) | ET (mm) | Evaporation (mm) | Transpiration (mm) | Drainage (mm) |
|---|---|---|---|---|---|---|
| **LK Run Type** | | | | | | |
| Site SA | 983 (391) | 5043 (6256) | 5660 (6928) | 3573 (5319) | 2087 (1608) | 145 (23) |
| Site A | 326 (34) | 3192 (4239) | 3159 (4273) | 2477 (3681) | 681 (592) | 0 (0) |
| Site HA | 147 (66) | 1700 (2314) | n/a | 1590 (2273) | n/a | 31 (2) |
| Site HU | 4 (43) | 17 300 (19 849) | 10 485 (11 424) | 4045 (4709) | 6440 (6715) | 7145 (8620) |
| **HK Run Type** | | | | | | |
| Site SA | 467 (197) | 5564 (6450) | 6116 (7075) | 3590 (5294) | 2525 (1781) | 376 (24) |
| Site A | 7 (0) | 3512 (4273) | 3458 (4308) | 2357 (3599) | 1100 (709) | 0 (0) |
| Site HA | 2 (4) | 1846 (2377) | n/a | 1641 (2271) | n/a | 97 (5) |
| Site HU | 0 (12) | 17 299 (19 880) | 10 437 (11 384) | 3999 (4667) | 6438 (6717) | 7189 (8685) |

*Code and data availability.* The code used to extract precipitation and all variables needed to compute climate model PET (as well as the code to compute PET) can be found at: https://doi.org/10.6084/m9.figshare.28187072.v2 (Blake, 2025). All other code used to analyse CP4A rainfall/PET data and Hydrus simulations can be provided upon request.

CP4A and P25 data is publicly available at the Centre for Environmental Data Analysis Archive under the IMPALA: Improving model processes for African climate, datasets can be accessed from the CEDA Archive at https://data.ceda.ac.uk/badc/impala/data/ (last access: TS10). The CP4A/P25 PET datasets for the Horn of Africa can be downloaded separately from https://doi.org/10.6084/m9.figshare.28187072.v2 (Blake, 2025). Or you can use the provided code to compute PET for another domain (Blake, 2025). IMERG is publicly available at the NASA Global Precipitation Measurement Mission Data Directory (https://doi.org/10.5067/GPM/IMERG/3B-HH/07 TS11, Huffman et al., 2023), and the code needed to extract hPET data for any region of interest can be found at: https://github.com/Dagmawi-TA/hPET (last access: TS12; DOI: https://doi.org/10.5523/bris.qb8ujazzda0s2aykkv0oq0ctp TS13, Singer et al., 2020).

*Author contributions.* GB, KM, EK, MC, and MBS designed this study. GB performed all analysis of climate model data, PET computation, and all Hydrus simulations. EK facilitated access to CP4A data via JASMIN. KM and EK assisted GB in analysis of results. MC assisted in Hydrus parameterisation and model set up. All authors contributed to writing and revising the manuscript.

*Competing interests.* The contact author has declared that none of the authors has any competing interests.

ther geographical representation in this paper. While Copernicus Publications makes every effort to include appropriate place names, the final responsibility lies with the authors. Views expressed in the text are those of the authors and do not necessarily reflect the views of the publisher.

*Acknowledgements.* This work was funded by the UKRI Natural Research Environment Council under reference NE/S007504/1 with assistance from those funded under the Horizon 2020 DOWN2EARTH project (grant no. 869550). We would also like to acknowledge the UK Met Office, which provided additional funding and facilitated access to CP4A via JASMIN. Katerina Michaelides acknowledges support from a University of Bristol Research Fellowship and Leverhulme Research Fellowship (grant no. RF-2023-591\4).

Aspects of the schematic of our Hydrus simulations provided in Fig. 2 were created with the use of Artificial Intelligence (DALL·E 3).

*Financial support.* This research has been supported by the UK Research and Innovation (grant no. NE/S007504/1). TS14

*Review statement.* This paper was edited by Nadia Ursino and reviewed by Bo Huang, Federico Gómez-Delgado, and one anonymous referee.

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

Please note the remarks at the end of the manuscript.

Track changes document – Do not use for proofreading

**Remarks from the typesetter**

TS1    Should "potential evapotranspiration" be added here?

TS2    Please confirm added citation.

TS3    A5–A14 or A1–A10?

TS4    Please check; according to our standards, the Appendix equations should start with (A1).

TS5    Equations (A5)–(A10) or Equations (A5) and (A10)?

TS6    Please give an explanation of why this figure needs to be changed. We have to ask the handling editor for approval. Thanks.

TS7    Please confirm this has been inserted correctly.

TS8    Please confirm this has been inserted correctly.

TS9    Please confirm this has been inserted correctly.

TS10    Please provide date of last access.

TS11    Please confirm change to DOI and added citation.

TS12    Please provide date of last access.

TS13    Please confirm added DOI and citation.

TS14    The Financial support section is mandatory, and the funding information in both Acknowledgements and Financial support sections have to be identical. I would kindly ask you to check this and amend this section. Thank you.

TS15    Last access dates are required for all URLs according to our standards, because links may expire.

TS16    Please provide URL + last access date.

TS17    Please confirm reference list entry.

TS18    The publication year is correct according to the citation information found in the DOI. Please advise where this citation should be added in the text.

TS19    Please provide date of last access.

TS20    Please confirm addition.

TS21    "[data set]" has not been added because this is a reference for a journal article.

TS22    "[data set]" has not been added because this is a reference for a journal article.

TS23    Please confirm addition.

TS24    Should this reference list entry be removed?

TS25    Please provide date of last access.

TS26    Please provide date of last access.

TS27    Please provide date of last access.