# Peer review of "The Impact of Convection-Permitting Model Rainfall on the Dryland Water Balance"

_EGUsphere, 2025_

## Referee Comment (RC2)

**Federico Gómez-Delgado**
**Referee comment on the article:**

**The Impact of Convection-Permitting Rainfall on the Dryland Water Balance**

By George Blake, Katerina Michaelides, Elizabeth Kendon, Mark Cuthbert, and Michael Bliss Singer

**General comments**

This paper examines how climate model representation of convection affects vadose-zone hydrological simulations across an aridity gradient in the Horn of Africa. Using four sites – humid (Ethiopian Highlands), semi-arid (southern Kenya), arid (eastern Ethiopia), and hyper-arid (northern Somalia) – the authors compare the convection-permitting model CP4A with the parameterized model P25, benchmarked against satellite rainfall (IMERG) and potential evapotranspiration (hPET) datasets. One-dimensional hydrological responses are simulated using Hydrus 1-D.

I believe this study contributes to the advancement of hydrological modelling, through an Earth System approach, by demonstrating how effects such as the 'drizzle' bias introduced by some climate models when parameterizing the average effects of convection on precipitation can be effectively corrected (especially for use in drylands), by adopting a convection-permitting approach. This improves vadose-zone hydrology projections in drylands, which is an important step toward generating global change scenarios relevant to protecting lives and livelihoods. These scenarios, within a framework of hydrological modelling at the watershed scale (ideally validated with field observations), would allow for more realistic recommendations for long-term socioeconomic planning.

In my assessment, the climatological and soil physics analyses are methodologically valid, well-supported bibliographically, and adequately discussed in the paper. Furthermore, the article is very well written, with high-quality tables and figures, and solid statistical calculations and reporting to support interesting comparisons and inferences.

I also find it interesting that through the analysis of water movement in unsaturated soils and the quantitative estimation of the main vertical flows (infiltration, evaporation, transpiration and drainage), it is possible to verify the premise that the moisture lost to evaporation is greater if climate models that parameterize convection are used (producing lower intensity rainfall distributed more evenly over time), while a convection-sensitive model simulates greater penetration and retention of water in the root zone, producing greater transpiration, in a sustained manner over time.

Notwithstanding the above, noting that neither the contrast between convection-permitting and convection-blind climate models, nor the employed modelling strategy, are strictly novel (since the use of climate models to provide inputs to 1-D partial hydrological modelling can be found in different studies around the world, and noting also that this paper does not employ primary field/observational data), I consider that the manuscript should provide a more precise narrative

regarding its scientific scope and relevance within the broader domain of hydrological sciences, highlighting not only the findings reached (as acknowledged above) but also the limits of the proposed modelling strategy, given that only some of the "containers" and larger-scale processes of the hydrological cycle are analysed here, leaving out others such as watershed-scale surface hydrology and hydrogeological processes. Such a simplification of the hydrological system could only be justified if the hypothesis that such processes are negligible in this region of the world could be demonstrated by extensive bibliographic documentation. This would lead to the strong assumption that vertical transport dominates the hydrological system, and that this condition has low spatial (horizontal) variability in the study area. Otherwise, point modelling of vertical water flows in a few sites would limit the ability to make a sufficiently comprehensive inference about the availability/scarcity of water resources for human and ecosystem use.

In summary, this study offers a well-executed comparison of climate models coupled with a robust soil physics analysis, which is however tested through discrete modelling of a vertical profile at only four sites along a long subcontinental aridity gradient. It focuses on routing water dynamics in the unsaturated zone of the soil (modelled assuming a constant uptake layer of 3 meters along the entire transect, and a simplified routine for estimating surface runoff), without considering horizontal movements of water over the soil in the form of Hortonian or saturation-excess overland flow, return flow, and surface channel flow (surface hydrology), nor shallow subsurface movement as inter- or preferential flow (subsurface hydrology), nor multidirectional interaction or exchange with the saturated zone and aquifer systems (groundwater hydrology), which, even in arid to semi-arid climates, can provide baseflow in perennial or quasi-perennial rivers.

Considering the above, I would suggest the following:

1. Reformulate some statements in the analysis of results and conclusions (in this review, I propose some wording for your consideration)

2. To expand the interesting discussion in lines 563-575 by adding more references (without this becoming an exhaustive meta-analysis) that delve deeper, through observational and/or modelling studies, into the possible effect on water resource distribution caused by watershed-based surface hydrological processes in this vast study region (ephemeral and permanent fluvial hydrology – especially in the presence of perennial or quasi-perennial rivers, as well as flood and flash flood events) and hydrogeological processes (aquifer recharge and baseflow through groundwater return in the form of gravity, fault-controlled, or capillary rise/artesian flow, among others), elaborating on how the assumptions of the approach used here can be improved in order to more comprehensively generalize the modelling results and conclusions to the landscape scale, and making recommendations for future research. This may also contribute to a more multidisciplinary approach and balance (in favour of hydrology) the significant analytical weight the article places on topics of climatology, soil physics, and agronomy, as seen in the discussions and the solid bibliographical support provided, for example, in lines 525-553, 559-561, and 576-590. I believe the exercise proposed here may also enrich not only the title and abstract but especially the conclusions of the article.

**Specific comments**

In **L.14-15** you state "However, rainfall datasets used in hydrological modelling and assessments of water resources are typically derived from climate models." I suggest removing the word "typically," considering that many hydrologic modelling applications, not only in research but also for operational purposes (as part of early warning systems, for example), rely on inputs of observed precipitation from weather stations, numerical weather prediction (NWP) models, radar or satellite estimates, or others. Another option might be to say "However, in the absence of precipitation estimates based on observations, rainfall datasets used in hydrological modelling and assessments of water resources are typically derived from climate models."

**L.28, 30 & 199:** "bottom drainage" is not a universal term in hydrology and is mostly linked to the conceptualization of the modelling process, so to start with, you may want to elaborate a little more on this, for example, by phrasing it here as you did in L.207: "drainage below the soil profile".

**L.30-31:** when you say "…means surface runoff is up to ten times higher and bottom drainage up to 25 times higher…" are you talking in terms of flow rate or in terms of total depth/volume?

**L.31:** I would rather say: "…We conclude that dryland vadose zone hydrology is highly sensitive to climate model representation of convection…"

**L.32-33:** when you say "…forcing hydrological model projections with convectional climate models that parameterise the average effects of convection risks underestimating future crop health…" But viewed from another perspective, a convection-permitting model would simulate longer dry periods (increasing water stress) and more intense rainfall events (risk of crop damage or flooding), which could imply worse (but more realistic) crop health compared to the output of the conventional model. If so, wouldn't conventional climate models be mistakenly "more optimistic" and thus overestimate future crop health?

**L.38:** I would say "…by limited  rainfall that varies greatly in time and space, where high temperature…"

**L.39:** proposed amendment: "…exceeds the available moisture supply stored in the soil…"

**L.51:** proposed amendment: "…drylands cover ~45% of the Earth's land surface…" (as we know, water covers ~71% of the Earth's total surface)

**L.70-71:** in this statement: "…with temporal offsets between potential evapotranspiration (PET) and rainfall capable of directly influencing impacting soil moisture…", knowing that PET is a theoretical concept of evaporative demand potential, and that although it experiences temporal variations, it is of a continuous nature, what does a "temporal offset between PET and rainfall" mean? Can you explain this a little more in detail?

**L.79-80:** regarding your statement: "...when runoff is significant enough to generate flow in dry channels, leading to localised transmission losses…", I want to note that under the traditional concept of a hydrologic system model, runoff over land or discharge in rivers or canals are considered either variables or outputs. While the analysis and conclusions of this study are not

explicitly posed in terms of such a hydrologic system model, they are at least framed at a landscape scale. Therefore, it might be more appropriate to conceptualize runoff or streamflow as variables subject to system transformation functions, rather than as a "transmission losses" which is why I suggest reviewing the terminology used here. So, we could rather say that when runoff is significant enough to generate flow in dry channels, "it runs off at localized points", or something along those lines.

**L.198-199:** What about infiltration in this list of modelled processes? "We used Hydrus 1-D v4.17 (Šimůnek et al., 2012) to simulate dynamic changes in surface runoff, evaporation, transpiration, soil moisture, and bottom drainage when forced with each climate model rainfall and PET…"

**L.105-106:** In relation to this statement: " Furthermore, no studies to date have assessed how model representation of convection can impact the atmospheric variables that control PET. ", I did a very quick search for possible studies addressing this topic, and I found some references that might be relevant (in fact, the first recommended reference includes as first author one of the co-authors of this paper). Such references, along with others worth exploring, could be included here as part of a more detailed bibliographic review:

- Kendon, E. J., Stratton, R. A., Tucker, S. O., Marsham, J. H., Berthou, S., Rowell, D. P., Roberts, N. M., and Finney, D. L.: Convection-permitting climate simulations for South America with the Met Office Unified Model: model evaluation and climate change impacts, Clim. Dynam., 61, 3517–3539, https://doi.org/10.1007/s00382-023-06853-0, 2023.
- Hohenegger, C., Dirmeyer, P. A., D'Andrea, F., and Pritchard, M. S.: Weaker land–atmosphere coupling in global storm-resolving simulation, Proc. Natl. Acad. Sci. USA, 121, e2314265121, https://doi.org/10.1073/pnas.2314265121, 2024.
- Skinner, C. B., Poulsen, C. J., and Eltahir, E. A. B.: How does the explicit treatment of convection alter the precipitation–soil hydrology interaction in the mid-Holocene African Humid Period?, Clim. Past, 19, 637–652, https://doi.org/10.5194/cp-19-637-2023, 2023.
- Omotosho, J. B., and Abiodun, B. J.: Sensitivity of dynamical downscaling seasonal precipitation forecasts to convection and land surface parameterization in a high-resolution regional climate model, Adv. Meteorol., 2019, 6010674, https://doi.org/10.1155/2019/6010674, 2019.

**L.134-136:** could you review the paragraph: "However, it is important to note that CP4A uses a uniform soil map that assumes all soils to be sandy, which risks poor representation of soil moisture – precipitation feedbacks that are critical …". I don't think it's sufficiently clear, as it discusses two ideas (soil type/precipitation feedback) without sufficiently establishing the relationship or causality between them.

**L.141-143:** two datasets (IMERG, Huffman et al., 2012; and hPET, Singer et al., 2021) are used in this study as references for rainfall and hourly PET. Verifying the high quality of these products, which also have extensive coverage and are openly accessible, makes me wonder about the utility/gain of using models like P25 or CP4A for any water resources application. Would it be possible to delve deeper into this?

**L.143-147:** While recognizing the very high quality of the IMERG product, the fact that very good quality meteorological station records could be available at certain sites makes me believe that it would be more prudent to slightly reword this statement to read: "IMERG utilises space-based radar, passive microwave, infrared, and rain gauge data from the Global Monthly Precipitation Climatology Centre (Huffman et al., 2012), Its high spatial (30') and temporal resolution (half-hourly)  the most appropriate for evaluating dryland rainfall metrics (Ageet et al., 2022) in the absence of good quality local weather station records in the immediate vicinity where an analysis will be run. However, IMERG is only available from June 2000, so we can only compare CP4A/P25 to 6.5 years of rainfall data…"

**L.202-205:** Most of the studies cited in these lines adopt a simulation strategy similar to the one presented here, in which processes such as surface runoff, horizontal subsurface flow, aquifer recharge and return flows, flash floods and flooding, etc., are considered negligible (although in some cases, observations of groundwater levels at specific sites are used to validate the models). For example, Boas and Mallants (2022) assume runoff and hysteresis are negligible in the context of their study in arid zone environments of central Australia. I believe it would be important to understand the arguments and assumptions employed by these authors when providing a justification for (or stating the limitation of) neglecting all these processes in the present study.

**L.212-213:** in the sentence: "…includes a sink term to account for root water uptake…", it would be worth checking whether it is possible to homogenize the terms root water uptake with transpiration.

**L.218:** "Hence, we ran four 1-D vadose-zone hydrological simulations along…"

**L.227:** "…To ensure our one-dimensional vadose-zone hydrological simulations isolate…"

**L.229:** in the statement: "…mean annual rainfall and PET was broadly comparable…" what do you mean by "broadly comparable"?

**L.230-232:** in the statement: "…this ensures that if fluxes such as soil moisture or bottom drainage are higher when forcing Hydrus with CP4A rainfall, it is reflective of differences in rainfall characteristics rather than simply higher annual totals.", if I understood the exercise correctly, this would only hold as long as the model parameters are kept constant.

**L.234:** the title of Table 1 indicates that the rainfall and PET simulated by CP4A are in bold, but it doesn't indicate where the non-bolded figures come from. Are the non-bolded from P25?

**L239:** the title of Fig. 1 indicates that the site identifier for the Ethiopian Highlands wetlands is "Site HU," however, the figure itself labels it as "Site H." Can you verify the consistency of the use of "HU" and "H" for this site throughout the document?

**L.242-244:** I would rather say: "Our experimental one-dimensional Hydrus simulations examine how climate model representation of convection can control how moisture propagates vertically through the vadose zone of a particular site, rather than aiming to reproduce 'realistic' hydrological simulations."

**L.247-249:** please note that the following assumption: "All Hydrus simulations utilised a three-meter soil profile (preliminary simulations suggested minimal water fluxes below this depth at some locations) with a free draining bottom boundary (no interactions between water Table and soil profile above)." is very strong, especially in semi-arid but especially in humid hydrological systems. Again, this should prevent us from overtly generalizing these results to the scale of basin or landscape hydrologic systems.

**L.259:** I would reintroduce the reference here, this time in the main text body of the manuscript: "…from the iSDAsoil database (iSDA, 2024), which applies…"

**L.302-303:** how could you prove the statement: "CP4A does not simulate the same 'drizzle effect' in drylands and offers a clear improvement in the frequency of dryland rainfall…"

**L.304-305:** the statement: "Using the Kolmogorov-Smirnov (KS) test shows that while there is still a statistically significant difference in the distribution on rainfall relative to IMERG …" is not clear to me. Please elaborate a little more on how the test was used and what the hypotheses were (the difference between which distributions is being tested?)

**L.319-320:** in the statement: "While both climate models replicate the spatial pattern of CDD observed in IMERG (CDD is higher in drylands), the relative biases of P25/CP4A compared to IMERG are opposing…", please explain better what I have underlined (maybe you could give some examples to better illustrate what you are saying)

**L.352:** in "…the median value is just 36% higher in CP4A vs P25…", what do you mean by just?

**L.354:** in "…dryland regions (7.1 mm vs 5.8 mm), although this may be related to the use of wet rather than all hours when computing percentiles…" is this comparison referring to CP4A vs P25, or wet vs dryland? What do you mean with "the use of wet"?

**L.363-364:** since you indicate that "…Given we have used the IMERG 95th percentile as our threshold, we are more focused on comparing CP4A and P25 to each other rather than IMERG.", you should perhaps exclude Fig.5 (d) from the mosaic: if it is not directly comparable with (e) and (f) this could cause confusion

**L.370:** in "…values are 21.5% and 7.8% respectively…", is this comparison referring to CP4A vs P25 or to Ethiopia vs Somalia?

**L.376-378:** in "…Bottom Panel - Percentage of mean annual rainfall that falls during 'heavy' rainfall events, in this context we are defining a 'heavy' rainfall event as the 95th percentile of IMERG rainfall (wet hours).", does this description of the Bottom Panel apply to Fig. 5 (d)?

**L.386:** in "…CP4A simulates PET that exceeds 2000 mm a-1 in just 18% of cells…" do you mean that PET exceeds 2000 mm yr-1?

**L.421-422:** in "…and distributions (Kolmogorov-Smirnov) at all sites, the differences are more pronounced in drylands (KS statistics…", when you say "all sites" does this also include Site

HU? When presenting KS statistics, are you reporting the Test Statistic (D), the P values, or something else?

**L.425-426:** your statement "…differences at site A are more pronounced if we consider depth-integrated θs at 1.2 mbgl, as below this depth there are minimal fluxes…" is, again, an extremely strong assumption.

**L.507-509:** is the statement "…this metric is not indicative of groundwater recharge, as in reality it is unlikely the water table would be so shallow, and moisture could still be lost to transpiration through deep rooted shrubs (Stone and Kalisz, 1991; Maeght et al., 2013; Shadwell and February 2017)." soundly supported by literature for your study transect, or is it otherwise a risky assumption?

**L.551:** the statement "…while most hydrological studies tend not to distinguish between soil evaporation and transpiration…" may be irrelevant, considering that although they are not the majority, there are numerous examples of hydrological model applications (including basin-scale models) in which such a distinction is made.

**L.592:** again, the range of relevant hydrological processes can be expanded here: "…as this risks producing misrepresentative projections of metrics such soil moisture, transpiration, overland flow, floods and flash floods, baseflow (groundwater return) and groundwater recharge, which could contribute to sub-optimal decision making around long-term land use or water supply policy…."

**L.605-606:** Again, considering that several leading processes that can produce large-scale flow transfers are not part of the modelling strategy here, in order to avoid overreaching in this conclusion, I would rather say: "…Our results also  suggest that while PET can influence hydrological vadose-zone outcomes, dryland hydrology  appears to be more sensitive to the impact of climate model representation of convection on rainfall…."

**Technical corrections**

Below I recommend technical and typographical corrections to this manuscript, and some typing suggestions.

**L.26**: "…where at each of our four sites  Hydrus…"

**L.68:** "…the dryland water balance is also sensitive to how synchronicity between rainfall and evaporative demand impacts…"

**L.97-98**: "…parameterised climate models, and critically if  dryland water partitioning sensitive to climate model representation of convection (via its impact on rainfall and PET)…."

**L.124:** "…(Stratton et al., 2018; Kendon et al., 2019)…"

**L.136-137:** "…There are also clear limitations with results based upon…"

**L.144:** "…(hPET, Singer et al. 2021)…"

**L.146:** "…its high spatial (30') and temporal resolution (half-hourly) means it is the most appropriate…". Please note that the single prime (') is the SI-accepted symbol for the unit of the minute of arc.

**L.152:** "… it is available at a high spatial (0.1 degree) and…"

**L.158:** "2. 10 m meridional (v) wind speed (m s-1)"

**L.166:** "… Several studies have demonstrated…"

**L.172:** "2. Maximum precipitation dry spell length" Missing units!

**L.173:** Missing units!

**L.178:** "…Furthermore, precipitation dry spell length…"

**L.187:** "…(hour with rainfall >= 0.1 mm)…"

**L.260:** "…range of fine- and coarse-scale…"

**L.284:** "…PET as well as rainfall, in order to assess whether the impact…"

**L.289-290:** title of Fig. 2 is divided (one part before and the rest after the figure)

**L.290:** "…to note that  the  figure above represents both shrubs, maize, and bare soil,  whereas only one vegetation type can be modelled."

**L.303:** "…of dryland rainfall. In both humid and dryland regions CP4A still simulates…"

**L.307:** "…Kernel density estimate (KDE) plots of CP4A…"

**L.333:** the legend of the horizontal axes of the three upper figures should be better positioned

**L.337-344:** All this information could be better presented in a table.

**L.347:** "…climate models underestimate the magnitude of wet extreme events relative to IMERG…"

**L.355:** "…computing percentiles, as P25  significantly overestimates the frequency of rainfall (most notably in drylands)."

**L.373:** considering the title of Fig. 5 (b), shouldn't the title of Fig. 5 (c) be "CP4A - IMERG"?

**L.374-378:** please check the clarity of the title of Figure 5 and, in general, of the titles of all figures in the manuscript

**L.393:** "…diurnal cycle (Fig. 6d&f) and replicate the hPET seasonal cycle (Fig. 6e&g)…"

**L.397-399:** All this information could be better presented in a table.

**L.407:** "…At each hydrological study site, CP4A and P25 correctly simulate the seasonal cycle of rainfall and tend to produce broadly comparable seasonal totals (Appendix C - Figure C1 ), although on average P25 deliver higher annual rainfall (Table 2). Both models also

produce comparable seasonal PET totals and simulate the same seasonal cycle, although P25 simulates substantially higher PET during JJAS at Site HA (Appendix C – Figure C2 2C) …"

**L.422-425:** All this information could be better presented in a table. Also, why are the sites listed in this order: SA, A, HA, and HU? Wouldn't it be easier to interpret the results if they were sorted by aridity level? Same applies for Figure 7 (L.435)

**L.445-446:** "…vegetation health modelling. For example, while differences in median depth integrated θs at Site A is are less than three percentage points, shallower wetting fronts…"

**L.459:** "…higher soil moisture in Hydrus CP4A simulations was were a function of…"

**L.465:** "…for longer using CP4A (41% vs 24%), compared to P25. The reduction is especially…"

**L.471:** Please note that Figure 9 is never mentioned in the manuscript!

**L.472:** in "Figure 10 details how water is partitioned between surface runoff, evaporation, transpiration, and bottom drainage…" this procedure is done for which of the study sites?

**L.475-476:** Note that what is indicated in this text "Figs. 10a - f shows substantially higher transpiration at Sites SA (2392 mm vs 1724 mm) & A (893 mm vs 694) when using CP4A rainfall…", is something that cannot be seen anywhere in Fig. 10 (the modelling sites are not indicated here)

**L.485-486:** please note that the green dashed line (a), and the red dashed line (b) cannot be clearly seen in the current version of Fig. 9 (there seems to be a colouring problem in the figure)

**L.492:** "…despite mean annual lower PET being is lower…"

**L.498:** Figure 10 presents the results from which of the study sites?

**L.505:** "…In dryland locations, between 6% and 10% of rainfall is lost to runs off when…"

**L.520-522:** "…fundamentally opposing manners (light/frequent vs heavy/infrequent), resulting in differing hydrological 1D modelling outcomes when their output is propagated through a vadose-zone hydrological model. This study also verifies that while dryland vadose-zone hydrology is more sensitive to PET than humid regions, differing hydrological outcomes water flows are primarily driven by rainfall…"

**L.534:** "…Our modelling demonstrates that the vertical water partitioning in drylands is sensitive to…"

**L.538:** "…produce such differing hydrological vadose-zone flow outcomes when propagated through a simple 1-D model highlights the importance of carefully selecting driving datasets in vadose-zone, or more comprehensive hydrological studies…"

**L.544:** "…when forcing hydrological models with CPM rainfall (Ascott et al., 2023; Archer et al., 2024),. And while no flood…"

**L.590:** please add the references in "…Taking a 'storyline' approach (add reference) built around stochastic scenarios…"

---

## Author Comment (AC1)

**The Impact of Convection-Permitting Rainfall on the Dryland Water Balance – Reviewer Comments**

We would like to thank Dr Bo Huang for his helpful comments on our manuscript. Below please find responses to each comment, which we hope help clarify some of the points raised. Here we will provide responses to points 1-3 raised by Dr Huang, as the internal HESS response platform does not allow for easy presentation of equations needed to address the issues raised. Responses to points 4-7 raised by Dr Huang are provided in main reply.

1. **The authors use the FAO Penman-Monteith method (Section 2.2) to calculate potential evapotranspiration (PET) and list seven atmospheric variables. However, the equation itself and the role of these variables in its application are not explicitly provided. Additionally, Penman-Monteith method uses wind speed at 2m height but this manuscript does not clarify how wind speed measured at 10m height is adjusted to 2m. Please include the equation, explain the variables, and describe the methodology for converting 10m wind speed to 2m (e.g., logarithmic wind profile adjustment or FAO-recommended constants).**

Thank you for spotting this. We actually did convert wind speed from 10 to 2 meters, but we omitted to clarify this in the methods. The manuscript will be updated to include this step. We converted the original climate model output which is for 10 meters above the land surface to the required 2 metre value using the logarithmic velocity profile above a short grass surface:

$$u_2 = u_z \left( \frac{4.87}{\ln (67.8z - 5.42)} \right)$$

Where $u_z$ is the wind speed at height $z$ above the land surface (10 meters in this case) computed as $u_z = \sqrt{u^2 + v^2}$. It is worth noting that where the shape of the velocity profile does not follow this form, errors may arise.

For brevity we didn't include the Penman-Monteith equation itself as it is heavily documented elsewhere, but we appreciate that clarity would be improved if we explicitly outlined its use in this study. We follow the methodology outlined by Singer et al (2021).

We used the Pen-Montieth equation for reference crop evapotranspiration as described in Allen et al (1998) to compute PET at an hourly resolution (*t*) at each pixel (*x*) in our domain:

$$hPET_{x,t} = \frac{0.408\Delta(R_n - G + \gamma(\frac{37}{T_a + 273})u_2(e_s - e_a)}{\Delta + \gamma(1 + 0.34u_2)}$$

Where $R_n$ is hourly net radiation (MJ m$^{-2}$), $G$ is the soil heat flux (MJ m$^{-2}$), $\gamma$ is the psychometric constant (kPA °C$^{-1}$), $\Delta$ is the slope of saturation vapour pressure (kPA °C$^{-1}$), $T_a$ is hourly air temperature (°C), $e_s$ is saturation vapour pressure (kPa), $e_a$ is the actual vapour pressure, and $u_2$ is the converted (from 10 m above the land surface) wind speed (m s$^{-1}$) at 2 m above the land surface.

For use in the above equation, $e_s$ and $e_a$ are calculated using the Tetens equation (Tetens, 1930) using hourly air temperature ($T_a$) and dew point temperature ($T_{dew}$) as detailed below (calculations are in in °C after converting from K):

$$e_s = 0.6108 \exp\left(\frac{17.27 * T_a}{T_a + 237.3}\right)$$

$$e_a = 0.6108 \exp\left(\frac{17.27 * T_{dew}}{T_{dew} + 237.3}\right)$$

Slope of saturation vapour pressure (Δ) and the psychrometric constant are calculated as follows:

$$\Delta = \frac{4098 e_s}{(T_a + 237.3)^2}$$

$$\gamma = \frac{C_p * P}{\varepsilon * \lambda}$$

Where $P$ is atmospheric pressure, $C_p$ is the air's specific heat at constant pressure based on the ideal gas law with a value of $1.013 \times 10^{-3}$ MJ kg$^{-1}$ per °C, ε is the ratio of the molecular weight of water vapor to that of dry air (0.622), and $\lambda$ is the latent heat of vaporization, (2.45 MJ kg$^{-1}$).

Net radiation ($R_n$) is estimated using net solar ($R_s$) and thermal radiation ($R_t$) as (all values in MJ m$^{-2}$):

$$R_n = R_s - R_t$$

Finally soil heat flux (G) is estimated as:

$$G = \begin{cases} G_{day} = 0.1 * R_n \\ G_{night} = 0.5 * R_n \end{cases}$$

Where the soil heat flux (G) is estimated to be 10% of net radiation ($R_n$) during the day and 50% during the night (as the night-time heat flux is negative). At each pixel we use net solar radiation to define day and nighttime periods. Unlike many other PET datasets, nighttime PET values have not been automatically set to zero.

2. **The climatological analysis appears to aggregate data across the entire study period. Are there any differences between wet and dry seasons? Given the region's likely seasonal contrasts (Figure C1), will the results/signal change in wet and dry seasons?**

Thank you for this comment which makes a good point. Yes, we did also analyse at the seasonality in the rainfall metrics. For brevity and to reduce the number of figures we decided to only report the annual results. However, we can add the below figure to supplementary material, which shows the 'drizzle' effect in P25 is evident in all the seasons including the MAM and OND rainy seasons and the dry season(s) JF and JJAS.

[Figure]

*Figure 1. Rainfall KDE Plots. Kernel density estimate (kde) plots of CP4A, P25, and IMERG hourly rainfall in humid (AI >= 0.65) and dryland (AI < 0.5) regions of the Horn of Africa for JF (a-b), MAM (c-d), JJAS (e-f), and OND (g-h). Plots exclude dry hours by dropping any hours that receive < 0.1 mm/hr of rainfall.*

3. **The aridity index (AI) is referenced repeatedly (Lines 37, 226, 527), but its definition (e.g., ratio of precipitation to PET or another formula) is not provided. Please clarify the specific equation or source used for calculating AI to ensure reproducibility and reader comprehension. Or directly cite the reference to indicate the four regions other than use value of aridity index.**

We use Aridity Index (AI) as P/PET (Zomer et al, 2007). Under section 2.4.2 we will add the following text and table (which will be labelled Table 1 in the main body of text): Given the sensitivity of dryland hydrology to rainfall characteristics, we wanted to establish whether relative differences in hydrological outcomes between Hydrus simulations (when forced with CP4A and P25 rainfall/PET) varied with aridity. Hence, we ran four 1-D hydrological simulations along an aridity gradient across the Horn of Africa (HOA), ranging from humid to hyper-arid (Fig. 1). Here we classify aridity based on aridity index (AI = P/PET) values taken from the CGIAR-CSI (Consortium of International Agricultural Research Centres' Consortium for Spatial Information) (Zomer et al., 2007) using the classification of Mirzabaev et al (2019).

The Aridity Index (AI) is a numerical indicator of climatic aridity based on long-term precipitation deficits relative to atmospheric water demand:

$$AI\ (Aridity\ Index) = MAP\ /\ MAE$$

Where MAP is mean annual precipitation and MAE is mean annual potential evaporation, CGIAR-CSI calculate both MAP and MAE using data obtained from WorldClim Global Climate Data (Hijmans et al, 2005). CGIAR-CSI outputs AI values at 1 km resolution, which can be used to define the climate type based on the climate classification of Mirzabaev et al, 2019, shown in Table 1.

| Climate Type | Aridity Index |
|---|---|
| Hyper-Arid | AI < 0.05 |
| Arid | 0.05 <= AI < 0.2 |
| Semi-Arid | 0.2 <= AI < 0.5 |
| Dry Sub-Humid | 0.5 <= AI < 0.65 |
| Humid | AI >= 0.65 |

*Table 1 – Climate classifications based on aridity index thresholds taken from (Mirzabaev et al, 2019).*

**Hijmans, R.J., Cameron, S.E., Parra, J.L., Jones, P.G. and Jarvis, A., 2005. Very high resolution interpolated climate surfaces for global land areas. *International Journal of Climatology: A Journal of the Royal Meteorological Society*, *25*(15), pp.1965-1978.**

---

## Author Comment (AC2)

**CPM Paper Reviewer 2 Response**

We would like to thank Dr Gómez-Delgado for his very detailed and thoughtful review. His comments are very helpful and we are glad to have the opportunity to elaborate on some of the issues raised. Below we respond to each of the comments (in red).

**General comments**

"the manuscript should provide a more precise narrative regarding its scientific scope and relevance within the broader domain of hydrological sciences" and the "limits of the proposed modelling strategy" should be explicitly outlined.

We acknowledge that one-dimensional point-based hydrological modelling is a simplification of the hydrological system in the Horn of Africa and that it does not capture watershed-scale surface hydrological pathways and hydrogeological processes. However, the focus on 1D processes was a deliberate decision that would most effectively isolate the impact of rainfall characteristics – specifically the representation of convection  on rainfall intensity-duration – on vertical vadose zone hydrological partitioning, without the complexity that would be introduced by lateral and non-local processes if modelling was carried out at a basin or regional scale. Runoff generation in drylands is predominantly via the infiltration-excess overland flow mechanism (Hortonian overland flow) which is also a function of rainfall intensity and infiltration rate. However, we exclude the consideration of runoff generation and subsurface lateral flows as they tend to add more water to downslope locations which may then infiltrate or evaporate. We wanted to strip away these added water sources to simply understand the balance between evapotranspiration, soil moisture and deeper drainage. This balance alone, provides valuable insights into water available for plants and groundwater recharge vs that lost back to the atmosphere, purely based on rainfall representation. We can add more text in the revised manuscript (in Section 2.4) to justify the approach and to acknowledge the hydrological processes excluded. We will also caveat our approach by expanding the discussion in lines 563–575 to include literature on surface and subsurface hydrological processes in drylands and their response to rainfall characteristics.

Critically for a dryland context, we will emphasise literature demonstrating the importance of surface flows (runoff) in their contribution to focused recharge in drylands, and the ephemeral flow or pooling that leads to transmission losses is only initiated by high-intensity rainfall events. Given that our results show that CP4A better represents the upper tails of the rainfall distribution (i.e., reduced underestimation of 99th percentiles relative to IMERG), it is reasonable to infer that using CP4A to drive a basin/regional-scale hydrological model could lead to greater non-local recharge and more realistic runoff patterns than P25. We will also strengthen our argument by noting that CP4A-driven Hydrus simulations show higher surface runoff, which at a landscape scale could contribute to lateral redistribution and recharge processes.

We appreciate the reviewer's emphasis on the importance of scaling these insights to broader hydrological systems and believe that strengthening the discussion with additional literature can provide more context around the implications of these results at a landscape scale. But a larger scale spatial analysis is beyond the scope of the current study and would be better achieved in a separate paper.

We hope that these clarifications and our planned revisions address the reviewer's points and reinforce the value of our study as a focused, process-oriented investigation into the hydrological impacts of convective representation in climate models.

Below are our responses to Dr Gómez-Delgado's specific comments :

**Specific Comments**

In **L.14-15** you state "However, rainfall datasets used in hydrological modelling and assessments of water resources are typically derived from climate models." I suggest removing the word "typically," considering that many hydrologic modelling applications, not only in research but also for operational purposes (as part of early warning systems, for example), rely on inputs of observed precipitation from weather stations, numerical weather prediction (NWP) models, radar or satellite estimates, or others. Another option might be to say

"However, in the absence of precipitation estimates based on observations, rainfall datasets used in hydrological modelling and assessments of water resources are typically derived from climate models."

We will rewrite the above sentence to clarify that we meant that climate model output is typically used in the context of modelling future projections of water resources.

**L.28, 30 & 199:** "bottom drainage" is not a universal term in hydrology and is mostly linked to the conceptualization of the modelling process, so to start with, you may want to elaborate a little more on this, for example, by phrasing it here as you did in L.207: "drainage below the soil profile".

For brevity and to maximise understanding to a wide audience, in the abstract we can simply add a bracket to provide more context, so that L.28 reads: "bottom drainage (indicative of potential recharge)". We will update L199 to read: "We used Hydrus 1-D v4.17 (Šimůnek et al., 2012) to simulate dynamic changes in ….. bottom drainage when forced with each climate model rainfall and PET, where in this context bottom drainage refers to any drainage out of the 1D soil profile (note - while this water could contribute to groundwater recharge it could also still be lost to transpiration)."

**L.30-31:** when you say "…means surface runoff is up to ten times higher and bottom drainage up to 25 times higher…" are you talking in terms of flow rate or in terms of total depth/volume?

Here we are referring to total cumulative volume over the ten-year simulation. We will update the above to make this clear.

**L.31:** I would rather say: "…We conclude that dryland vadose zone hydrology is highly sensitive to climate model representation of convection…"

Yes, we agree that this clarification is useful.

**L.32-33:** when you say "…forcing hydrological model projections with convectional climate models that parameterise the average effects of convection risks underestimating future crop health…" But viewed from another perspective, a convection-permitting model would simulate longer dry periods (increasing water stress) and more intense rainfall events (risk of crop damage or flooding), which could imply worse (but more realistic) crop health compared to the output of the conventional model. If so, wouldn't conventional climate models be mistakenly "more optimistic" and thus overestimate future crop health?

Thank you, this is a really good point that we hadn't fully considered. Here we have made this assertion based on lower soil moisture and longer periods where acacia shrubs are below the wilting point. But we would agree that crop health is dependent on other factors such as heat stress and potential flooding. To make such a statement we need to conduct specific crop modelling, so we will rewrite the above sentence to: "…forcing hydrological model projections with convectional climate models that parameterise the average effects of convection risks underestimating the soil moisture critical for crops …". In the discussion where we also consider crop health, we will incorporate your comments provided above.

**L.38:** I would say "…by limited and highly variable rainfall that varies greatly in time and space, where high temperature…"

We can change the sentence as suggested.

**L.39:** proposed amendment: "…exceeds the available moisture supply stored in the soil…"

Yes, we are happy to accept this amendment.

**L.51:** proposed amendment: "…drylands cover ~45% of the Earth's land surface…" (as we know, water covers ~71% of the Earth's total surface)

We will update to "…drylands cover ~45% of the Earth's land surface…"

**L.70-71:** in this statement: "…with temporal offsets between potential evapotranspiration (PET) and rainfall capable of directly influencing impacting soil moisture…", knowing that PET is a theoretical concept of

evaporative demand potential, and that although it experiences temporal variations, it is of a continuous nature, what does a "temporal offset between PET and rainfall" mean? Can you explain this a little more in detail?

While PET does evolve seasonally and temporally its variability is far more limited compared to rainfall. The point we were trying to emphasise here is that PET alone is capable of impacting hydrological metrics relevant to human well-being. For example, Kimutai et al (2025) suggest that PET was a key driver of the recent 2020-2023 multi-season drought. We can simply alter the above statement to read: "In addition to rainfall characteristics, the dryland water balance is also sensitive to atmospheric evaporative demand (PET or potential evapotranspiration), both in how high PET impacts antecedent soil moisture conditions (soils quickly dry out between rainfall events) (Zhang and Shilling, 2006; Nazarieh et al., 2018; Cuthbert et al., 2019; Schoener and Stone, 2019; Schoener, 2021; Boas and Mallants, 2022) and its direct impact on agricultural yields and drought severity (Porporato et al., 2002; Lobell et al., 2011; Vicente-Serrano et al., 2018; Tugwell-Wootton et al., 2020; Kimutai et al., 2023)."

**L.79-80:** regarding your statement: "...when runoff is significant enough to generate flow in dry channels, leading to localised transmission losses…", I want to note that under the traditional concept of a hydrologic system model, runoff over land or discharge in rivers or canals are considered either variables or outputs. While the analysis and conclusions of this study are not explicitly posed in terms of such a hydrologic system model, they are at least framed at a landscape scale. Therefore, it might be more appropriate to conceptualize runoff or streamflow as variables subject to system transformation functions, rather than as a "transmission losses" which is why I suggest reviewing the terminology used here. So, we could rather say that when runoff is significant enough to generate flow in dry channels, "it runs off at localized points", or something along those lines.

We can rewrite the above to make it clear that "...when rainfall is heavy enough to generate surface runoff, a certain proportion can enter dry river channels and generate locally substantial flows that can lead to localised transmission losses…",

**L.198-199:** What about infiltration in this list of modelled processes? "We used Hydrus 1-D v4.17 (Šimůnek et al., 2012) to simulate dynamic changes in surface runoff, evaporation, transpiration, soil moisture, and bottom drainage when forced with each climate model rainfall and PET…"

Yes, thank you for spotting this oversight. Will add infiltration to the above.

**L.105-106:** In relation to this statement: " Furthermore, no studies to date have assessed how model representation of convection can impact the atmospheric variables that control PET. ", I did a very quick search for possible studies addressing this topic, and I found some references that might be relevant (in fact, the first recommended reference includes as first author one of the co-authors of this paper). Such references, along with others worth exploring, could be included here as part of a more detailed bibliographic review:

•        Kendon, E. J., Stratton, R. A., Tucker, S. O., Marsham, J. H., Berthou, S., Rowell, D. P., Roberts, N. M., and Finney, D. L.: Convection-permitting climate simulations for South America with the Met Office Unified Model: model evaluation and climate change impacts, Clim. Dynam., 61, 3517–3539, https://doi.org/10.1007/s00382-023-06853-0, 2023.

•        Hohenegger, C., Dirmeyer, P. A., D'Andrea, F., and Pritchard, M. S.: Weaker land–atmosphere coupling in global storm-resolving simulation, Proc. Natl. Acad. Sci. USA, 121, e2314265121, https://doi.org/10.1073/pnas.2314265121, 2024.

•        Skinner, C. B., Poulsen, C. J., and Eltahir, E. A. B.: How does the explicit treatment of convection alter the precipitation–soil hydrology interaction in the mid-Holocene African Humid Period?, Clim. Past, 19, 637–652, https://doi.org/10.5194/cp-19-637-2023, 2023.

• Omotosho, J. B., and Abiodun, B. J.: Sensitivity of dynamical downscaling seasonal precipitation forecasts to convection and land surface parameterization in a high-resolution regional climate model, Adv. Meteorol., 2019, 6010674, https://doi.org/10.1155/2019/6010674, 2019.

We will clarify what we mean. These studies look at internally simulated evapotranspiration (ET) from the climate model. We computed PET using the Penman-Monteith equation from model outputs of the atmospheric variables needed (2 m zonal (u) wind speed (m s-1), 2 m meridional (v) wind speed, 2 m dew point temperature (K), 2 m air temperature (K), surface net solar radiation (J m-2), surface net thermal radiation (J m-2), atmospheric surface pressure (Pa)) . While internal ET is useful for exploring land-atmosphere interactions — particularly feedbacks between soil moisture and precipitation — it reflects the model's internal assumptions about land surface properties, such as soil type and vegetation. In the case of CP4A, for example, ET is calculated using a uniform sandy soil across the entire domain, which limits its applicability in spatially heterogeneous hydrological studies.

By contrast, externally calculating PET from model-derived atmospheric variables provides a consistent measure of atmospheric evaporative demand, independent of land surface parameterizations. For hydrological purposes where more detailed or locally calibrated soil and vegetation data is available, it is important to use the potential atmospheric evaporative demand computed using atmospheric model outputs. We will revise the manuscript to clarify this distinction and better justify the use of externally computed PET in the context of hydrological modelling.

L.134-136: could you review the paragraph: "However, it is important to note that CP4A uses a uniform soil map that assumes all soils to be sandy, which risks poor representation of soil moisture – precipitation feedbacks that are critical …". I don't think it's sufficiently clear, as it discusses two ideas (soil type/precipitation feedback) without sufficiently establishing the relationship or causality between them.

We will update the above section to clarify the point, so that it reads: "However, it is important to acknowledge the uncertainties associated with CP4A, one being that the use of a uniform soil map (assumes all soils to be sandy) could impart biases in the spatial pattern of rainfall across the Horn of Africa. As in water limited regions such as the Horn of Africa, soil moisture (partly a function of soil properties) directly regulates evapotranspiration and can contribute to precipitation via moisture recycling (Seneviratne et al, 2010), so a uniform soil map risks poor representation of the soil moisture – precipitation feedbacks that are critical to inducing convective rainfall and a realistic spatial pattern of rainfall (Taylor et al., 2011, 2012; Hsu et al., 2017; Zhou et al., 2021).

L.141-143: two datasets (IMERG, Huffman et al., 2012; and hPET, Singer et al., 2021) are used in this study as references for rainfall and hourly PET. Verifying the high quality of these products, which also have extensive coverage and are openly accessible, makes me wonder about the utility/gain of using models like P25 or CP4A for any water resources application. Would it be possible to delve deeper into this?

While IMERG and hPET are indeed high-quality products that can be used for a wide range of water resources applications, they cannot be used to model future water resources.  Studies that consider the impact of future climate change on water resources typically use climate model output. Hence, we want to assess how model choice may impact future projections (solely in terms of rainfall characteristics and PET dynamics). I think a potential issue throughout the manuscript is that we have not made it explicitly clear enough that we are focussing on CP4A/P25 as climate model output is the key resource when considering future climate change impacts on water resources (but comparing them to historical model output as we have observational data to compare it to).

L.143-147: While recognizing the very high quality of the IMERG product, the fact that very good quality meteorological station records could be available at certain sites makes me believe that it would be more prudent to slightly reword this statement to read: "IMERG utilises space-based radar, passive microwave, infrared, and rain gauge data from the Global Monthly Precipitation Climatology Centre (Huffman et al., 2012)., iIts high spatial (30' mins) and temporal resolution (half-hourly) means it is the most appropriate for evaluating dryland rainfall metrics (Ageet et al., 2022) in the absence of good quality local weather station records in the

immediate vicinity where an analysis will be run. However, IMERG is only available from June 2000, so we can only compare CP4A/P25 to 6.5 years of rainfall data…"

We can include this caveat. While there are some (very limited) weather stations available in the region, they are not available at hourly resolution and often have frequent gaps. We also wanted to run the rainfall/PET comparison across the entire Horn of Africa drylands rather than at specific sites.

**L.202-205:** Most of the studies cited in these lines adopt a simulation strategy similar to the one presented here, in which processes such as surface runoff, horizontal subsurface flow, aquifer recharge and return flows, flash floods and flooding, etc., are considered negligible (although in some cases, observations of groundwater levels at specific sites are used to validate the models). For example, Boas and Mallants (2022) assume runoff and hysteresis are negligible in the context of their study in arid zone environments of central Australia. I believe it would be important to understand the arguments and assumptions employed by these authors when providing a justification for (or stating the limitation of) neglecting all these processes in the present study.

As we mention above, we are not saying that these lateral flows are not important in arid regions. We are deliberately adopting a simple 1D approach to demonstrate the direct impact of rainfall characteristics on the water balance by stripping away the complexities. This comment will be addressed by the general edits to the paper, where we explicitly acknowledge the rationale for the approach and the processes neglected.

**L.212-213:** in the sentence: "…includes a sink term to account for root water uptake…", it would be worth checking whether it is possible to homogenize the terms root water uptake with transpiration.

This is how transpiration is referred to within Hydrus documentation (root water uptake).I will alter the above to read: "…includes a sink term to account for root water uptake (hereby referred to as transpiration) …". We will then only use transpiration after this point.

**L.218:** "Hence, we ran four 1-D vadose-zone hydrological simulations along…"

Accepted

**L.227:** "…To ensure our one-dimensional vadose-zone hydrological simulations isolate…"

Accepted

**L.229:** in the statement: "…mean annual rainfall and PET was broadly comparable…" what do you mean by "broadly comparable"?

We agree that broadly is too vague without further clarification. We choose our four study sites in grid cells where CP4A/P25 simulated rainfall and PET that was within +/- 35% of each other, as it is very challenging to find grid cells where total rainfall and PET are exactly identical. For example, P25 simulated 15%, 12%, 23%, and 33% higher total annual rainfall at sites HU, SA, A, and HA respectively. And P25 simulated 15%, 3%, 13%, 15% higher total annual PET at sites HU, SA, A, and HA respectively. We can add this additional information to increase clarity on what we mean by 'broadly' in this case.

**L.230-232:** in the statement: "…this ensures that if fluxes such as soil moisture or bottom drainage are higher when forcing Hydrus with CP4A rainfall, it is reflective of differences in rainfall characteristics rather than simply higher annual totals.", if I understood the exercise correctly, this would only hold as long as the model parameters are kept constant.

If the reviewer means Hydrus parameters, then yes, these are kept constant between the CP4A and P25 runs. What we wanted to emphasize here is that one may expect fluxes such as soil moisture and bottom drainage to broadly follow annual rainfall totals. e.g. if P25 simulates higher total rainfall we might expect higher soil moisture and bottom drainage. Whereas given CP4A simulates lower total rainfall, if driving Hydrus with CP4A results in higher soil moisture and bottom drainage (versus P25) it cannot be argued that this is simply a function of it simulating higher total rainfall. We can alter the above lines to make this clearer.

**L.234:** the title of Table 1 indicates that the rainfall and PET simulated by CP4A are in bold, but it doesn't indicate where the non-bolded figures come from. Are the non-bolded from P25?

Yes, the non-bolded figures are from P25, we will update the caption and figure to make this clear.

**L239:** the title of Fig. 1 indicates that the site identifier for the Ethiopian Highlands wetlands is "Site HU," however, the figure itself labels it as "Site H." Can you verify the consistency of the use of "HU" and "H" for this site throughout the document?

Thank you for spotting this oversight. In earlier version our humid site was labelled as H rather than HU. I will ensure all labelling is consistent. HU – Humid, SA – Semi-Arid, A – Arid, and HA – Hyper-Arid.

**L.242-244:** I would rather say: "Our experimental one-dimensional Hydrus simulations examine how climate model representation of convection can control how moisture propagates vertically through the vadose zone of a particular site hydrological system, rather than aiming to reproduce 'realistic' hydrological simulations."

Yes, we are happy with this rewording and will alter the text accordingly.

**L.247-249:** please note that the following assumption: "All Hydrus simulations utilised a three-meter soil profile (preliminary simulations suggested minimal water fluxes below this depth at some locations) with a free draining bottom boundary (no interactions between water Table and soil profile above)." is very strong, especially in semi-arid but especially in humid hydrological systems. Again, this should prevent us from overtly generalizing these results to the scale of basin or landscape hydrologic systems.

I believe here you are noting that interactions between the water table and unsaturated zone is important in semi-arid to humid regions and that we should make it explicitly clear that this is why our results should not be overtly generalised to a basin/landscape scale. While we agree with this, we feel that with the additional manuscript revisions detailed above that we will have sufficiently enhanced clarity about the scope of this work and its limitations. But we can provide evidence from the literature that water table depth is generally deep in drylands, including specifically in the Horn of Africa (Bonsor and MacDonald, 2011; Fan et al, 2013).

**L.259:** I would reintroduce the reference here, this time in the main text body of the manuscript: "…from the iSDAsoil database (iSDA, 2024), which applies…"

Accepted

**L.302-303:** how could you prove the statement: "CP4A does not simulate the same 'drizzle effect' in drylands and offers a clear improvement in the frequency of dryland rainfall…"

We believe this is demonstrated in Figures 3 and 4a-c. The drizzle effect simulated by P25 can be clearly seen in Figure 3 with the peak in rainfall frequency of intensities < 1 mm/hr. We can add an additional graphic/circle/shape and labelling to emphasise this is the drizzle effect. It can also be seen in Figure 4b where P25 simulates most rainfall being delivered via low-intensity rainfall (< 1 mm/hr). Whereas CP4A does not replicate this peak in low-intensity rainfall in Figure 3 and better agrees with IMERG in terms of how much annual rainfall is delivered as drizzle in Figure 4.

**L.304-305:** the statement: "Using the Kolmogorov-Smirnov (KS) test shows that while there is still a statistically significant difference in the distribution on rainfall relative to IMERG …" is not clear to me. Please elaborate a little more on how the test was used and what the hypotheses were (the difference between which distributions is being tested?)

Thank you for pointing this out. We agree that the original sentence lacked clarity regarding the application of the Kolmogorov–Smirnov (KS) test. We have revised the sentence to explicitly state which distributions are being compared and the nature of the statistical test.

"To quantitatively assess differences in rainfall frequency distributions, we used the Kolmogorov–Smirnov (KS) test with a null hypothesis for each comparison that the modelled distribution of hourly rainfall intensities (based on all hours with rainfall > 0.1 mm/h) is drawn from the same distribution as IMERG. Both P25 and CP4A

show statistically significant differences from IMERG (p < 0.05), but the KS statistic is markedly lower for CP4A (0.03) than for P25 (0.24), indicating that CP4A more closely reproduces the observed rainfall intensity distribution."

**L.319-320:** in the statement: "While both climate models replicate the spatial pattern of CDD observed in IMERG (CDD is higher in drylands), the relative biases of P25/CP4A compared to IMERG are opposing…", please explain better what I have underlined (maybe you could give some examples to better illustrate what you are saying)

We were aiming to emphasise that while both models simulate higher CDD in dryland areas to the east, but their bias relative to IMERG differs. Meaning that while P25 underestimates the number of CDD relative to IMERG, CP4A overestimates the number of CDD. We will update the above to read: "While both climate models replicate the spatial pattern of CDD observed in IMERG, where CDD is higher in drylands (compared to the Ethiopian Highlands), P25 underestimates CDD length compared to IMERG and CP4A overestimates CDD length….",

**L.352:** in "…the median value is just 36% higher in CP4A vs P25…", what do you mean by just?

Here we used 'just' as we were comparing 36% to >110% - "Differences between CP4A and P25 are more muted in humid regions, where the median value is just 36% higher in CP4A vs P25 (vs > 110% in drylands), and IQR ranges overlap …." But we agree that using the word 'just' is misleading and will be removed.

**L.354:** in "…dryland regions (7.1 mm vs 5.8 mm), although this may be related to the use of wet rather than all hours when computing percentiles…" is this comparison referring to CP4A vs P25, or wet vs dryland? What do you mean with "the use of wet"?

We clarify that the comparison in this sentence (7.1 mm vs 5.8 mm) refers to the 99th percentile of rainfall in humid versus dryland regions, respectively, using P25. By "the use of wet," we mean that percentiles were calculated based on all hours with rainfall ≥ 0.1 mm/h, rather than including all hours (both wet and dry). This approach focuses on the distribution of rainfall intensities during actual rain events, rather than across the full time series. We acknowledge that this method can influence the percentile values, especially in models like P25 that overestimate the number of wet hours (particularly in dryland regions, as shown by the drizzle bias in Figure 3). As a result, the 99th percentile may appear lower in drylands — not necessarily because rainfall extremes are weaker, but because the large number of light rain events dilutes the distribution of wet-hour rainfall. We will revise the manuscript to make this distinction clearer.

**L.363-364:** since you indicate that "…Given we have used the IMERG 95th percentile as our threshold, we are more focused on comparing CP4A and P25 to each other rather than IMERG.", you should perhaps exclude Fig.5 (d) from the mosaic: if it is not directly comparable with (e) and (f) this could cause confusion

Yes, we agree with you. We will swap Figure 5d with Figure B1 which shows the magnitude of the IMERG 95[th] percentile. This will allow readers to understand the spatial variability in the threshold magnitude we are using for Figures 5e and 5f. We will ensure Figure 5d uses a different colour scheme to 5e and 5f.

**L.370:** in "…values are 21.5% and 7.8% respectively…", is this comparison referring to CP4A vs P25 or to Ethiopia vs Somalia?

This is comparing CP4A and P25. The values refer to median contributions in arid regions of the Horn of Africa – which is primarily in eastern Ethiopia and Somalia. We can rewrite the above to: "In arid regions (eastern Ethiopia and Somalia) this difference becomes more pronounced in comparison to humid regions, with the median contributions of heavy rainfall being 21.5% and 7.8% in CP4A and P25 respectively.

**L.376-378:** in "…Bottom Panel - Percentage of mean annual rainfall that falls during 'heavy' rainfall events, in this context we are defining a 'heavy' rainfall event as the 95th percentile of IMERG rainfall (wet hours).", does this description of the Bottom Panel apply to Fig. 5 (d)?

Yes correct. We are defining heavy rainfall as the IMERG 95[th] percentile. It does apply to Figure 5d. But as we are swapping Figure 5d for Figure B1 this caption will be updated.

**L.386:** in "…CP4A simulates PET that exceeds 2000 mm a-1 in just 18% of cells…" do you mean that PET exceeds 2000 mm yr-1?

Yes, we do mean yr$^{-1}$. We can update it from a$^{-1}$ to yr$^{-1}$. We will ensure this labelling is consistently used throughout the manuscript.

**L.421-422:** in "…and distributions (Kolmogorov-Smirnov) at all sites, the differences are more pronounced in drylands (KS statistics…", when you say "all sites" does this also include Site HU? When presenting KS statistics, are you reporting the Test Statistic (D), the P values, or something else?

Yes, when stating all sites we are including Site HU. When we are not including Site HU we will use "all dryland sites". And we are only reporting the Test Statistic in the main body of the text, as all P values are statistically significant. We will make it clear throughout the manuscript where KS is used, we are only reporting the Test Statistic.

**L.425-426:** your statement "…differences at site A are more pronounced if we consider depth-integrated θs at 1.2 mbgl, as below this depth there are minimal fluxes…" is, again, an extremely strong assumption.

This isn't an assumption, it is based on our simulations and is shown in Figure 8c/d. We report depth-integrated θs values up to 1.2 mbgl as below this there are minimal fluxes (very little moisture reaches any deeper). So, reporting the relative differences in depth-integrated θs down to 3.0 mbgl will be lower as both CP4A and P25 simulate minimal moisture fluxes below this depth, so θs below 1.2 mbgl is mainly a function of the initial conditions used.

**L.507-509:** is the statement "…this metric is not indicative of groundwater recharge, as in reality it is unlikely the water table would be so shallow, and moisture could still be lost to transpiration through deep rooted shrubs (Stone and Kalisz, 1991; Maeght et al., 2013; Shadwell and February 2017)." soundly supported by literature for your study transect, or is it otherwise a risky assumption?

While the water table is shallow enough to be influenced by seasonal rainfall across much of the Horn of Africa, studies that have explicitly estimated water table depth estimate it to generally be far deeper than 3 meters below ground level. Bonsor and MacDonald (2011) model that water table depth in the Horn of Africa exceeds 50 meters across much of the region, with no areas seeing a water table shallower than 7 meters below ground level (Bonsor and MacDonald, 2011). While Fan et al (2013) generally estimates water table depths of 5 to 80 plus meters below ground level, with a small fraction shallower than 2.5 meters below ground level along the Kenyan coast. This is typical of many dryland regions.

There is also robust evidence that dryland acacia shrubs are particularly deep rooted and are capable of extracting water at depths water deeper than 3 meters, they can extract water directly from the water table (Stone and Kalisz, 1991; Maeght et al., 2013; Shadwell and February 2017).

Either way, as we are not explicitly trying to model representative fluxes, even if such an assumption was not fully supported by the literature it would not undermine the purpose of this study or the reporting of bottom drainage.

**L.551:** the statement "…while most hydrological studies tend not to distinguish between soil evaporation and transpiration…" may be irrelevant, considering that although they are not the majority, there are numerous examples of hydrological model applications (including basin-scale models) in which such a distinction is made.

We will edit the sentence to remove that part and retain "our results show moisture lost to evaporation is far higher when forcing Hydrus with P25, while using CP4A increases the volume of water available for transpiration within the root zone, meaning transpiration is higher and continues longer into the dry season (Folwell et al., 2022)."

**L.592:** again, the range of relevant hydrological processes can be expanded here: "…as this risks producing misrepresentative projections of metrics such soil moisture, transpiration, overland flow, floods and flash floods, baseflow (groundwater return) and groundwater recharge, which could contribute to sub-optimal decision making around long-term land use or water supply policy…."

We apologise but we are unclear on what the reviewer is noting here, with further clarification we will be more than happy to respond.

**L.605-606:** Again, considering that several leading processes that can produce large-scale flow transfers are not part of the modelling strategy here, in order to avoid overreaching in this conclusion, I would rather say: "…Our results also show suggest that while PET can influence hydrological vadose-zone outcomes, dryland hydrology appears to be more sensitive to the impact of climate model representation of convection on rainfall…."

Yes, we are happy to add 'appears' to these conclusions.

**Technical corrections**

Below I recommend technical and typographical corrections to this manuscript, and some typing suggestions.

**L.26:** "…where at each of our four sites sties Hydrus…"

Thank you for spotting this – will we update the spelling of sites.

**L.68:** "…the dryland water balance is also sensitive to how synchronicity between rainfall and evaporative demand impacts…"

Thank you for spotting the missing to. This sentence will be also updated based on a previous comment.

**L.97-98:** "…parameterised climate models, and critically if s dryland water partitioning sensitive to climate model representation of convection (via its impact on rainfall and PET)…."

Will change is to if.

**L.124:** "…(Stratton et al., 2018; Kendon et al., 2019)…"

Noted, thank you.

**L.136-137:** "…There are also clearly limitations with results based upon…"

Apologies but we are not sure what the change is here.

**L.144:** "…(hPET, Singer et al. 2021)…"

Noted, thank you.

**L.146:** "…its high spatial (30' mins) and temporal resolution (half-hourly) means it is the most appropriate…". Please note that the single prime (') is the SI-accepted symbol for the unit of the minute of arc.

This is an error, it should read its high spatial (0.1 degrees) and temporal resolution (half-hourly).

**L.152:** "… it is available at a high spatial (0.1 degrees) and…"

Apologies but we are not sure what the change is here.

**L.158:** "2. 10 m meridional (v) wind speed (m s-1)"

Noted, thank you.

**L.166:** "…While other Several studies have demonstrated…"

Noted, this will be updated to: "Several studies …"

**L.172:** "2. Maximum precipitation dry spell length" Missing units!

Noted, will add units (days)

**L.173:** Missing units!

Noted, will add units (mm/hr)

**L.178:** "…Furthermore, precipitation dry spell length…"

Noted, will add precipitation.

**L.187:** "…(hour with rainfall >= 0.1 mm of rainfall)…"

Will alter to "(any hour with >= 0.1 mm of rainfall)

**L.260:** "…range of fine- and coarse-scale…"

Noted, will alter to simply "fine and coarse-scale"

**L.284:** "…PET as well as rainfall, so we also in order to assess whether the impact…"

We will alter to "…PET, to ensure we are only capturing the influence of rainfall characteristics on water partitioning, we also forced Hydrus with climate model rainfall but replaced climate model PET with gridded hPET values (see Section 2.2)."

**L.289-290:** title of Fig. 2 is divided (one part before and the rest after the figure)

Noted, will change. Thank you.

**L.290:** "…to note that in the above figure above represents both shrubs, maize, and bare soil, represented, whereas only one vegetation type can be modelled."

Will alter to "It is important to note that in the above figure both shrubs, maize, and bare soil are represented, whereas only one vegetation type can be modelled in any one simulation."

**L.303:** "…of dryland rainfall., iIn both humid and dryland regions CP4A still simulates…"

Will alter to "CP4A does not simulate the same 'drizzle effect' in drylands and offers a clear improvement in the frequency of rainfall, although (in both humid and dryland regions) CP4A simulates fewer rainfall events with an intensity of > 10 mm/hr compared to IMERG.

**L.307:** "…Kernel density estimate (KDEkde) plots of CP4A…"

Thank you, will alter.

**L.333:** the legend of the horizontal axes of the three upper figures should be better positioned

Apologies, but we are not sure what needs to be altered here.

**L.337-344:** All this information could be better presented in a table.

Yes, we agree. Will alter.

**L.347:** "…climate models underestimate the magnitude of wet extremes events relative to IMERG…"

Yes agree, will alter.

**L.355:** "…computing percentiles, as P25 dramatically significantly overestimates the frequency of rainfall (most notably in drylands)."

Yes agree, will alter.

**L.373:** considering the title of Fig. 5 (b), shouldn't the title of Fig. 5 (c) be "CP4A - IMERG"?

Yes, thank you for catching this oversight. It should read CP4A – IMERG.

**L.374-378:** please check the clarity of the title of Figure 5 and, in general, of the titles of all figures in the manuscript

Noted.

**L.393:** "…diurnal cycle (Fig. 6d-&f) and replicate the hPET seasonal cycle (Fig. 6e-&g)…"

Noted, will alter.

**L.397-399:** All this information could be better presented in a table.

Yes, we agree, will make this change.

**L.407:** "…At each hydrological study site, CP4A and P25 correctly simulated the seasonal cycle of rainfall and tended to produce broadly comparable seasonal totals (Appendix C - Figure C1 1C), although on average P25 delivered higher annual rainfall (Table 2). Both models also produce comparable seasonal PET totals and simulate the same seasonal cycle, although P25 simulates substantially higher PET during JJAS at Site HA (Appendix C – Figure C2 2C) …"

Thank you, will update the tenses.

**L.422-425:** All this information could be better presented in a table. Also, why are the sites listed in this order: SA, A, HA, and HU? Wouldn't it be easier to interpret the results if they were sorted by aridity level? Same applies for Figure 7 (L.435)

Yes, we would agree with both points raised here. We will make these changes.

**L.445-446:** "…vegetation health modelling. For example, while differences in median depth integrated θs at Site A is are less than three percentage points, shallower wetting fronts…"

Noted, will alter.

**L.459:** "…higher soil moisture in Hydrus CP4A simulations was were a function of…"

Noted, will alter.

**L.465:** "…for longer using CP4A (41% vs 24%), compared to P25. The reduction is especially…"

Noted, will add compared to P25.

**L.471:** Please note that Figure 9 is never mentioned in the manuscript!

Thank you for capturing this oversight – in lines 445 – 449 where we are referring to Figure 8, we are meant to be referring to Figure 9. This will be updated.

**L.472:** in "Figure 10 details how water is partitioned between surface runoff, evaporation, transpiration, and bottom drainage…" this procedure is done for which of the study sites?

This is done for all sites but in Figure 10 only the three dryland sites are presented.

**L.475-476:** Note that what is indicated in this text "Figs. 10a - f shows substantially higher transpiration at Sites SA (2392 mm vs 1724 mm) & A (893 mm vs 694) when using CP4A rainfall…", is something that cannot be seen anywhere in Fig. 10 (the modelling sites are not indicated here)

This can be seen in Figure 10f. The dark blue is the semi-arid site and green is the arid site, with the dashed line being the P25 simulation and solid line being the CP4A simulations. We can update the figure capture and/or the legend to make this clearer.

**L.485-486:** please note that the green dashed line (a), and the red dashed line (b) cannot be clearly seen in the current version of Fig. 9 (there seems to be a colouring problem in the figure)

Both subplots have two orange dashed lines – the figure caption was not updated and is referring to an old version of the plot. The caption will be updated and colours altered to increase clarity.

**L.492:** "…despite mean annual lower PET being is lower…"

Noted, will alter.

**L.498:** Figure 10 presents the results from which of the study sites?

The three dryland sites. This will be made clearer both in the caption and main body of text.

**L.505:** "…In dryland locations, between 6% and 10% of rainfall is lost to runs off when…"

Noted, will alter.

**L.520-522:** "…fundamentally opposing manners (light/frequent vs heavy/infrequent), resulting in differing hydrological 1D modelling outcomes when their output is propagated through a vadose-zone hydrological model. This study also verifies that while dryland vadose-zone hydrology is more sensitive to PET than humid regions, differing hydrological outcomes water flows are primarily driven by rainfall…"

Noted, will alter.

**L.534:** "…Our modelling demonstrates that the vertical water partitioning in drylands is sensitive to…"

Noted, will alter.

**L.538:** "…produce such differing hydrological vadose-zone flow outcomes when propagated through a simple 1-D model highlights the importance of carefully selecting driving datasets in vadose-zone, or more comprehensive hydrological studies…"

Noted, will alter.

**L.544:** "…when forcing hydrological models with CPM rainfall (Ascott et al., 2023; Archer et al., 2024),. And while no flood…"

This will be updated to: "Although even in humid regions, studies have found differences in hydrological outcomes between CPMs and parameterised climate models, such as increased surface runoff (Folwell et al., 2022) and higher flood risk when forcing hydrological models with CPM rainfall (Ascott et al., 2023; Archer et al., 2024). No flood risk assessments using CP4A has yet been conducted across the HOA at time of writing, however, it is reasonable to assume that greater sub-daily rainfall extremes (Bethou et al., 2019; Kendon et al., 2019; Finney et al., 2019, 2020) and surface runoff will result in greater flood hazard potential in the HOA when forcing hydrological models with CPM rainfall."

**L.590:** please add the references in "…Taking a 'storyline' approach (add reference) built around stochastic scenarios…"

Thank you for noticing this absence, we will add the following reference: Shepherd, T.G., Boyd, E., Calel, R.A., Chapman, S.C., Dessai, S., Dima-West, I.M., Fowler, H.J., James, R., Maraun, D., Martius, O. and Senior, C.A., 2018. Storylines: an alternative approach to representing uncertainty in physical aspects of climate change. *Climatic change*, *151*, pp.555-571.

**References (not included in the original manuscript)**

Bonsor, H.C. and MacDonald, A.M., 2011. An initial estimate of depth to groundwater across Africa.

Fan, Y., Li, H. and Miguez-Macho, G., 2013. Global patterns of groundwater table depth. *Science*, *339*(6122), pp.940-943.

Seneviratne, S.I., Corti, T., Davin, E.L., Hirschi, M., Jaeger, E.B., Lehner, I., Orlowsky, B. and Teuling, A.J., 2010. Investigating soil moisture–climate interactions in a changing climate: A review. *Earth-Science Reviews*, *99*(3-4), pp.125-161.

Shepherd, T.G., Boyd, E., Calel, R.A., Chapman, S.C., Dessai, S., Dima-West, I.M., Fowler, H.J., James, R., Maraun, D., Martius, O. and Senior, C.A., 2018. Storylines: an alternative approach to representing uncertainty in physical aspects of climate change. *Climatic change*, *151*, pp.555-571.

---

## Author Response (AR2)

**The Impact of Convection-Permitting Model Rainfall on the Dryland Water Balance – Reviewer Response (Review #3)**

We thank the reviewer for their insightful comments and for taking the time to read the original and revised manuscript. We hope the following response will sufficiently address their comments, if not, we welcome providing additional clarification.

Main Reviewer Remark:

In this work, projections from two periods (a past one, and a future one), are used. I was expecting that the availability of these two periods would be exploited to assess the impact of the use of CP-RCM data to evaluate the evolution of hydrological indicators. However, it seems not to be the case. Actually, in most figures (except Fig 3 where all data are aggregated, and Figure 10 where we see that the past period is used), results are presented on unknown periods. This poses two issues : the first one, is that we simply don't know exactly what is shown. The second one is that, if the two periods are mixed, it involves that two different distributions of climate variables are mixed, troubling the conclusions we can draw. This issue needs to be tackled.

We thank the reviewer for this helpful comment and can address it without any major alterations to the manuscript. There appears to have been some confusion regarding the time slices used in this study. While CP4A is available for both a present-day (1997–2007) and a future (2095–2105) time slice, we only used the present-day simulation. This is already stated in the manuscript:

**Section 2.2 Climate Data**

"To establish whether CP4A can better capture dryland rainfall characteristics and PET dynamics (relative to P25), we compared both climate models (historical run) to the gridded Integrated Multi-satellitE Retrievals for GPM (IMERG, Huffman et al., 2012) rainfall product and an hourly potential evapotranspiration dataset (hPET, Singer et al., 2021)."

However, we agree that this needs to be made clearer. We will revise Section 2.2 to explicitly state that only the present-day (historical) simulations were used and to include the rationale for this choice.

We considered only the historical period because our analysis required direct comparison between model output (CP4A and P25) and observed rainfall and PET data. It is also worth noting that, the CP4A present-day time slice does not reproduce the actual timeseries of the climate variables between 1997-2007 – it is purely statistically consistent with historical observations. It simulates a plausible sequence of weather consistent with the observed large-scale climate of 1997-2007 but it does not reproduce the actual day-to-day or event-level sequence of observed weather over this period. CP4A realistically captures annual totals, rainfall seasonality, and the distribution of rainfall characteristics, but the precise timing, duration, spatial pattern, and intensity of individual storms do not correspond to those seen in IMERG.

We agree that exploring the future CP4A time slice to assess the evolution of hydrological indicators would be a valuable extension, but as discussed in the manuscript (Section 4.0), we have limited confidence in regional climate projections for East Africa. Including the future period would divert the focus from our central aim — understanding the influence of rainfall characteristics on hydrological partitioning. We believe that assessing future hydrological changes would be best addressed in a dedicated follow-up study. For analyses of projected rainfall changes in CP4A, we refer the reviewer to Kendon et al (2019).

We will update the manuscript accordingly.

Miscellaneous Comments:

Line 70: Is it really sensitive to PET? Because, as the authors state before, the evaporative demand is higher than the amount of water available. So, when PET increases, it might not really have an impact on actual ET, as it will rapidly be limited by the amount of available water.

- We agree with this comment that in water-limited environments hydrology is relatively insensitive to changes in PET. However, while its impact is limited compared to rainfall, PET can still exert an influence especially between rainfall events or during the rainy season. We will remove the following lines:

  "The dryland water balance is also sensitive to atmospheric evaporative demand (PET or potential evapotranspiration), both in how high PET impacts antecedent soil moisture conditions (soils quickly dry out between rainfall events) (Zhang and Shilling, 2006; Nazarieh et al., 2018; Cuthbert et al., 2019; Schoener and Stone, 2019; Schoener, 2021; Boas and Mallants, 2022) and its direct impact on agricultural yields and drought severity (Porporato et al., 2002; Lobell et al., 2011; Vicente-Serrano et al., 2018; Tugwell-Wootton et al., 2020; Kimutai et al., 2025)."

  And will update it to the following (and move it later in the introduction):

  In drylands, actual evapotranspiration (AET) is primarily constrained by soil moisture rather than atmospheric demand (i.e., water- rather than energy-limited), meaning that the direct hydrological effects of PET are typically small relative to those of precipitation (Vicente-Serrano et al., 2019). However, PET can still exert an important influence on vegetation and soil moisture dynamics, and plays a key role in land–atmosphere feedbacks (Seneviratne et al., 2010). Moreover, the strong temporal variability of rainfall means that drylands are not always water-limited—for example, during or following high-rainfall periods—when PET can shape antecedent soil moisture conditions before the next rainfall event, as soils dry rapidly between events (Zhang and Shilling, 2006; Nazarieh et al., 2018; Cuthbert et al., 2019; Schoener and Stone, 2019; Schoener, 2021; Boas and Mallants, 2022).

Line 148: which agreements ?

- There is consistent agreement amongst convection-permitting models in being able to better represent rainfall characteristics compared to non-convection permitting models. So, our findings that CP4A can better capture rainfall characteristics

(particularly in drylands) is in line with other studies that have used different CPMs in other regions.

Line 155: How does IMERG propose 30-min data if rain gauges data are daily? Does it come from the remote sensing data temporal resolution? Even if it is "the most appropriate" product in the area due to its temporal resolution, what is the quality of this product in this area?

- IMERG uses daily rain gauge data to bias-correct daily totals while keeping the temporal resolution at 30 mins. In Huffman et al., they state "that even monthly gauge analyses produce significant improvements, at least for some regions in some seasons. Recent work at CPC shows substantial improvements in the bias correction using daily gauge analysis for regions in which there is a sufficient number of gauges." For a more detailed methodology please refer to Huffman et al (2020).
- For a review of the quality of IMERG data in Africa (including the Horn of Africa) please refer to Dezfuli et al (2017). IMERG compares well to a gauge data (where available) and other satellite-derived rainfall products, but critical for this study is that is available at hourly resolution which is critical for evaluating dryland rainfall characteristics.

Line 157: Several grammatical imperfections are present in the text, such as this sentence that misses the main sentence part, there are also sentences beginning with "But", etc.

The line in question has been altered and any sentences that begin with "But" have been altered.

Line 166: What is the quality of wind, radiation and air humidity data in these two models? Usually it is of poor quality in climate models. In addition, are these models considering evolutive aerosols? If not, radiation should not be used and should be replaced with a proxy based on temperature. See Boé J., Somot S, Corre L, Nabat P (2020) : Large discrepancies in summer climate change over Europe as projected by global and regional climate models: causes and consequences. Clim Dyn. https://doi.org/10.1007/s00382-020-05153-1

- A robust analysis of wind, radiation, and humidity data is beyond the scope of this paper; however, we feel we do not need to change our methodology as our CP4A/P25 simulations do a reasonable job of replicating hPET data and any bias in our model PET does not undermine our key finding (dryland hydrological partitioning is highly sensitive to rainfall characteristics). Nonetheless, we thank the reviewer for flagging the above publication and we acknowledge that deficiencies in model representation of these variables could impart biases into our PET simulations (for example potentially the early peak in diurnal PET cycle in both models).
- Also, if by "evolutive aerosols" the reviewer is referring to whether aerosol concentrations vary in the present-day simulations and/or if there is a change between present and future time-slice, we can confirm that CP4A uses fixed aerosols. For the future time slice, only sea-surface temperature and well-mixed greenhouse gas concentrations are altered. This decision was made to ensure clear attribution of the most fundamental climate change (Senior et al., 2020). As we did not consider the future time slice in this analysis it is not a major concern.

Line 169 : Please verify all references to Appendixes, as it should rather be here Appendix A.

- Thanks for spotting this, we have updated and doubled checked all other references to Appendices.

Line 207 : It is rather strange to compare rainfall among periods of different durations (2 to 4 months)

- These periods are defined based on what the dominant rainy seasons are in the region. Rather than computing 99$^{th}$ percentiles using the entire annual rainfall record, we computed the 99$^{th}$ percentile of wet season rainfall only. We cannot choose one rainy season for the entire region as there is variability in seasonality. In bimodal regions across the eastern drylands the dominant rainy season is either MAM or OND, whereas for the more humid Ethiopian Highlands the dominant rainy season is JJAS.

- We will also update the manuscript to clarify that it is only for the 99$^{th}$ percentile (extreme) rainfall that we use these seasons. All other data is analysed at an annual scale (consecutive dry days, drizzle, light vs heavy contributions).

Line 209: The numbering is right, but here it should be noted Appendix.
In addition, Appendix B should be cited before Appendix D, I suggest to reorder them.

- Thank you for spotting this, we have altered it to Appendix. We also agree that the ordering of Appendices should be altered. We have moved the rainfall and PET figures from Appendix D (Fig. 1D – Fig. 3D) to Appendix B, we have not moved the Hydrus results from this Appendix. Meaning the original Appendix B is now Appendix C, Appendix C is now Appendix D, and Appendix D has now become Appendix E. These changes are reflected in the tracked manuscript, but for clarity I will summarise each Appendix here:

  Appendix A – Provides additional detail on the method of computing PET using the Penman-Monteith equation.

  Appendix B – Mean monthly rainfall/PET and raw hourly rainfall time-series at each of our four Hydrus locations – compliments the summary statistics provided in Table 2.

  Appendix C – Soil hydraulic and Feddes' parameters used for all Hydrus runs, including for results not shown in the main body of text.

  Appendix D – Figures that compliment Section 3.1 & Section 3.2 (rainfall and PET results).

  Appendix E – Additional results from Hydrus simulations – includes more detailed analysis of the default results (those primarily discussed in the main body of text) as well as additional low/high hydraulic conductivity/Feddes' parameter runs.

Line 214: Doesn't it seriously smooth the precipitation patterns?

- Regridding does smooth/dampen precipitation intensity, but if we are to compare climate model rainfall (in terms of characteristics) fairly we need to use a common grid. Else it would be challenging to discern whether differences in rainfall intensity (for

example between CP4A and P25) are due to model physics or simply the grid resolution of model output. As peak rainfall intensity is always likely to be higher if outputted on a smaller grid as it is averaged over a smaller spatial area. So, it is important to note that if we compared CP4A and P25 at their native grids (4.5 km vs 25 km) the differences in rainfall intensity would be even more pronounced.

Line 220: Did you use LAI from climate models? Is this LAI evolving (due to increased CO2 for instance)?

- LAI data was taken from Vermote (2019). It is a NOAA remote sensing product that captures the temporal evolution of LAI in the region. Please refer to Vermote (2019) for more details on the methodology.

  The source is stated in the manuscript:

  "To calculate transpiration Hydrus needs land cover (Table 2) and leaf area index (LAI) data, which were taken from iSDA (as of 2019) and the National Centers for Environmental Information AVHRR LAI dataset (Vemote, 2019) respectively."

Table 2: It is common practice, especially in hydrology, to bias correct climate projections. Here, it would have had even more sense, as the rationale of the study is that hydrology of drylands is impacted by the distribution of rainfall rather than the total amount, so having the same average amount would have allowed a fair comparison. I do see that the P/PET ratio is higher with P25, but the message could be altered.

- We agree that this would have been a potential option to ensure any hydrological differences only reflected differences in the rainfall distribution, but we feel that our choice to not bias-correct but select locations where P25 simulates higher rainfall (and higher P/PET ratio) is equally robust. As this means our findings that forcing Hydrus with CP4A rainfall results in higher soil moisture, transpiration, and bottom drainage is despite total rainfall being lower in CP4A compared to P25. So, we can confidently say that these higher values of the above variables are not simply driven by higher rainfall totals, but rather reflect how rainfall is delivered (light/frequent vs heavy/infrequent).

Line 315: « This mean » -> « This means »

- Updated

Line 319: does that mean that LAI is fixed for all years? Is it also the case for the future?

- LAI is time-evolving and reflects crop/vegetation growth from 1997 to 2007, see Vermote (2019) for more details. We did not consider the future CP4A time-slice.

Line 390: « IN » -> « in »

- Updated

Line 391: this is not very precise: for the IQR bounds? For the mean?

- These values refer to the median 99$^{th}$ percentiles, the manuscript will be updated to reflect this and a reference to see Table 3 for the IQR.

Figure 5: I agree that heavy rainfall is more frequent in CPA4 than in P25. However, as it is based on the 95th percentile of IMERG, shouldn't it be ideally 5% in CPA4? Here it seems that the percentage is too high.

- The 95$^{th}$ percentile of IMERG rainfall is used to define a 'heavy' rainfall threshold at every grid cell across our domain. It is not based on the 95$^{th}$ percentile of annual rainfall totals. This means that while 'heavy' rainfall events will be infrequent, they can contribute far more than just 5% of the annual rainfall totals. This is particularly true in drylands, where a large proportion of rainfall falls during very intense but short-lived events.

Figure 7: The periods that are studied are missing on most figures

- All results refer to analysis from 1997-2007, where it is appropriate to include the periods in the x-axis (time-series) we have done so. We have updated the manuscript in section 2.1 to make this clear.

References: Several DOIs are missing, Leterne et al. Is listed as a discussion paper whereas it has been accepted 13 years ago. Same goes for Quichimbo, accepted 4 years ago.

- Will update the references listed above and ensure all DOIs are listed where available.

**References (not cited in original manuscript).**

Dezfuli, A.K., Ichoku, C.M., Huffman, G.J., Mohr, K.I., Selker, J.S., van de Giesen, N., Hochreutener, R. and Annor, F.O., 2017. Validation of IMERG precipitation in Africa. *Journal of Hydrometeorology*, *18*(10), pp.2817-2825.

Senior, C., Finney, D., Owiti, Z., Rowell, D., Marsham, J., Jackson, L., Berthou, S., Kendon, E., Misiani, H. (2020). Technical guidelines for using CP4-Africa simulations data. Future Climate for Africa. https://doi.org/10.5281/zenodo.4316466.

---

## Author Response (AR3)

We thank the editor, Dr Ursino for her work in editing and reviewing our paper.

We assume that the editor's comment: "Please, address the main remark raised by reviewer #2 concerning the graphical representation of the data." refers to a point by the Anonymous referee #3 report dated 2$^{nd}$ October (labelled Report #2) which commented:

In this work, projections from two periods (a past one, and a future one), are used. I was expecting that the availability of these two periods would be exploited to assess the impact of the use of CP-RCM data to evaluate the evolution of hydrological indicators. However, it seems not to be the case. Actually, in most figures (except Fig 3 where all data are aggregated, and Figure 10 where we see that the past period is used), results are presented on unknown periods. This poses two issues: the first one, is that we simply don't know exactly what is shown. The second one is that, if the two periods are mixed, it involves that two different distributions of climate variables are mixed, troubling the conclusions we can draw. This issue needs to be tackled.

In our response to the review, we clarified that our study does not consider projections from two-periods (historical and future) in CP4A. We only consider the 'historical' CP4A runs which are statistically consistent with the large-scale climate between 1997-2007 but do not reproduce annual climate (such that in CP4A the rainfall in 2000 is does not reproduce the observed rainfall in 2000 – but that the overall statistics of the CP4A rainfall between 1997-2007 are consistent with observed statistics for the same period). We have strengthened this clarification in the manuscript and have made it clear that "all results discussed refer to the 'historical' CP4A/P25 runs, we do not use the 'future' runs at any point in this study." (Line ~386). Along with this statement we have also added additional clarification statements throughout the methods section. Finally, we have also made it clear in every figure caption that the results refer to the CP4A/P25 historical runs.

For example, the Fig. 4 caption has these lines added:

"**Plots cover rainfall recorded/simulated between June 2000 and February 2007, which is the period where CP4A/P25 and IMERG overlap. These 'historical' CP4A/P25 simulations are statistically consistent with the observed climate between March 1997 and February 2007.**"

Where there is a time-evolving x-axis, the following lines have been added to figure captions (such as Fig. 10):

**It is worth noting that while the x-axis covers 1997-2007, the 'historical' CP4A/P25 simulations are only designed to be statistically consistent with the observed climate rainfall between March 1997 and February 2007. They cannot replicate individual rainfall events observed over this period.**

We hope the updated manuscript addresses the point raised, if not we are more than happy to provide additional alterations.